# Safe Exploration via Policy Priors

**Manuel Wendl, Yarden As**[*]
ETH Zurich

**Manish Prajapat**
ETH Zurich

**Anton Pollak**
ETH Zurich

**Stelian Coros**
ETH Zurich

**Andreas Krause**
ETH Zurich

## ABSTRACT

Safe exploration is a key requirement for reinforcement learning (RL) agents to learn and adapt online, beyond controlled (e.g. simulated) environments. In this work, we tackle this challenge by utilizing suboptimal yet conservative policies (e.g., obtained from offline data or simulators) as priors. Our approach, SOOPER, uses probabilistic dynamics models to optimistically explore, yet pessimistically fall back to the conservative policy prior if needed. We prove that SOOPER guarantees safety throughout learning, and establish convergence to an optimal policy by bounding its cumulative regret. Extensive experiments on key safe RL benchmarks and real-world hardware demonstrate that SOOPER is scalable, outperforms the state-of-the-art and validate our theoretical guarantees in practice.

## 1 INTRODUCTION

A defining feature of intelligence is the ability to utilize online streams of information to learn and adapt over time (Silver & Sutton, 2025). Reinforcement learning (RL) provides a framework for online learning without supervision, driving several notable real-world applications (Silver et al., 2017; Ouyang et al., 2022). Safety is key to unlocking RL in the physical world—RL agents simply cannot risk taking actions that lead to catastrophic outcomes (Dalrymple et al., 2024), even *while* learning, when they lack complete knowledge about their environments (Legg, 2023). Such agents must *explore safely* (Amodei et al., 2016), gradually discovering new safe behaviors to solve their tasks.

Safe exploration is a major challenge in practice. Methods with theoretical safety guarantees often struggle to scale to complex, general-purpose tasks, whereas scalable approaches typically fail to ensure safety during learning (cf. Section 2). A key reason lies in how they incorporate prior knowledge about the environment—a crucial component for anticipating danger without learning through harmful trial-and-error. Such prior knowledge can be instantiated via *policies*. Beyond primarily maintaining safety, as done in much of prior work, we argue that prior policies can also *guide* exploration toward promising regions of the environment, enabling agents to provably learn *near-optimal* policies.

Motivated by this insight, we propose **S**afe **O**nline **O**ptimism for **P**essimistic **E**xpansion in **R**L— SOOPER—a model-based RL algorithm for safe exploration in constrained Markov decision processes (CMDP, Altman, 1999). SOOPER uses prior policies that can be derived from scarce offline data or simulation under distribution shifts. Such policies are invoked *pessimistically* during online rollouts to maintain safety. These rollouts are collected *optimistically* so as to maximize information about a world model of the environment. Using this model, SOOPER constructs a simulated RL environment, used for planning and exploration (Amodei et al., 2016), whose trajectories *terminate* once the prior policy is invoked. The key idea is that early terminations incentivize the agent to avoid the prior policy when trajectories with higher returns can be obtained safely. This design allows SOOPER to leverage standard RL methods, bypassing the complexity of directly solving CMDPs. More fundamentally, this enables a novel bound on the agent's cumulative regret.

**Our contribution.**

- First, we propose SOOPER, a scalable model-based RL algorithm for safe exploration in continuous state-action spaces that leverages offline- and simulation-trained policies as safe priors.

---

[*]Equal contribution. Corresponding authors: {mwendl,yardas}@ethz.ch

- Next, under standard regularity assumptions, we prove that SOOPER guarantees constraint satisfaction with high probability throughout learning. We additionally show that SOOPER gradually expands an implicit set of safe policies until it converges to the feasible set defined by the true CMDP. This is accomplished by reformulating the problem as an unconstrained MDP, making the approach naturally compatible with standard deep RL methods, while avoiding min-max formulations typical for CMDPs. Using this formulation, we further establish a novel upper bound on the cumulative regret, improving over prior works that provide optimality guarantees only at the end of training with arbitrarily poor performance during exploration (Wagener et al., 2021; As et al., 2025b).

- Finally, we conduct extensive empirical evaluation of SOOPER, comparing it with other state-of-the-art safe exploration algorithms when initialized from offline- and simulation-trained policies in RWRL benchmark (Dulac-Arnold et al., 2021) and SafetyGym (Ray et al., 2019). In all these experiments, SOOPER substantially outperforms the baselines in terms of safety and performance. We further validate SOOPER for safe online learning on *hardware*, giving empirical evidence that our theory translates to the real world.

## 2 RELATED WORKS

Prior works on safe exploration fall broadly into two categories: theoretically principled methods with guarantees on safety and optimality and those prioritizing practicality and scalability.

**Provable safety.** A canonical line of work leverages safe Bayesian optimization (Sui et al., 2015), which has seen real-world success (Berkenkamp et al., 2016; Kirschner et al., 2019; Sukhija et al., 2023b), but typically ignores the Markovian structure of sequential decision problems and is limited to a few parameters. Berkenkamp et al. (2017) extend to continuous MDPs and provide safety and optimality guarantees via Lyapunov stability, though they rely on manual resets of the system at each timestep and state-space discretization for safe set computation. In contrast, Prajapat et al. (2025b) leverage model predictive control (MPC) to implicitly compute these sets and ensure that the agent can always return to a safe state. Other MPC-based methods ensure safety either through "backup" policies (Koller et al., 2018; Prajapat et al., 2024; 2025a) or via low-fidelity models with reachability analysis (Hewing et al., 2019) and safety filters (Wabersich & Zeilinger, 2021; Curi et al., 2022). However, these approaches lack optimality guarantees. A substantial body of work addresses safe exploration in *discrete* CMDPs (Moldovan & Abbeel, 2012; Turchetta et al., 2016; Wachi et al., 2018; Wachi & Sui, 2020; Ding et al., 2021; Simão et al., 2021; Prajapat et al., 2022; Bura et al., 2022). Akin to our work, Bura et al. (2022) proposes a model-based approach that uses optimism and pessimism to provide optimality and safety guarantees for small-scale tabular problems. Although the above methods guarantee safety throughout learning, scaling them to high-dimensional problems is computationally hard.

**Scalability.** The works of Dalal et al. (2018); Eysenbach et al. (2018); Srinivasan et al. (2020); Thananjeyan et al. (2021); Bharadhwaj et al. (2021); Thomas et al. (2021); Sun et al. (2021); Liu et al. (2022); As et al. (2022); Sootla et al. (2022); Huang et al. (2024) use scalable tools from deep RL for safe learning, with different levels of theoretical guarantees and effectiveness in maintaining safety throughout learning in practice. Safe exploration has also been studied from an *optimization* perspective, with methods aiming to guarantee feasibility of all optimization iterates. These techniques span trust regions (Achiam et al., 2017; Milosevic et al., 2024), Lyapunov stability (Chow et al., 2018), interior-point (Liu et al., 2020; Usmanova et al., 2024; Ni & Kamgarpour, 2025), and primal-dual optimization (Usmanova & Levy, 2025). While these methods scale well, they often overlook the exploration aspect of learning, namely how to actively sample data that maximizes information about the environment. As a result, they often lack optimality except under assumptions on the initial state distribution.

**Optimality.** As et al. (2025b) propose ActSafe, which guarantees safety and optimality for *simple* regret, relying on reward-free exploration, and therefore might perform poorly during exploration. Wachi et al. (2023) introduce MASE, which enforces safety under high-probability constraints on safe states but relies on a restrictive "emergency stop" action that can effectively 'stop time' at any state. Lastly, Wagener et al. (2021) propose SAILR, a state-of-the-art algorithm that like SOOPER, reformulates safety through terminations whenever their "backup" policy is invoked. However, its guarantee on simple regret hinges on the probability of such resets vanishing over time, yet no formal conditions are provided to ensure this occurs. In this work, we relax several of these assumptions and establish a novel bound on the cumulative regret, while also demonstrating improved empirical performance.

## 3 PROBLEM SETTING

**Constrained Markov decision processes.** Safety in RL has been conceptualized in several ways across the literature (García et al., 2015; Brunke et al., 2022; Gu et al., 2024). In this work, we focus on constrained Markov decision processes (CMDP, Altman, 1999), due to their flexibility and capacity to generalize several notable formulations of safety (cf. Wagener et al., 2021; Curi et al., 2022; Wachi et al., 2023). A discounted infinite-horizon CMDP is defined by the tuple $\mathcal{M}_c \coloneqq (\mathcal{S}, \mathcal{A}, p, r, c, \gamma, \rho_0)$, where $\mathcal{S} \subseteq \mathbb{R}^{d_{\mathcal{S}}}$ and $\mathcal{A} \subset \mathbb{R}^{d_{\mathcal{A}}}$ are the continuous state and compact action spaces, with the transition probability $p(s_{t+1}|s_t, a_t)$ of reaching $s_{t+1} \in \mathcal{S}$ by applying $a_t \in \mathcal{A}$ in $s_t \in \mathcal{S}$. We consider the transition probabilities $p$ to be governed by the *unknown* stochastic dynamics

$$s_{t+1} = f(s_t, a_t) + \omega_t, \quad (s_t, a_t) \in \mathcal{S} \times \mathcal{A}, \quad s_0 \sim \rho_0, \tag{1}$$

with the initial state distribution $\rho_0$ and additive noise $\omega_t \in \mathcal{W} \subseteq \mathbb{R}^{d_{\mathcal{S}}}$. Throughout our theoretical analysis, we consider the setting of additive zero-mean Gaussian noise.

**Assumption 1** (Gaussian noise (Kakade et al., 2020)). *The additive noise $\omega_t$ is independent and identically distributed (i.i.d.) zero-mean Gaussian noise with variance $\sigma^2$.*

We note that this assumption can be relaxed to a richer class of sub-Gaussian distributions. The reward and the cost are given by $r : \mathcal{S} \times \mathcal{A} \to [0, R_{\max}]$ and $c : \mathcal{S} \times \mathcal{A} \to [0, C_{\max}]$ with discount factor $\gamma \in (0, 1)$. We assume the following regularity conditions for the dynamics, reward, and cost.

**Assumption 2** (Lipschitz-continuity (Berkenkamp et al., 2017)). *The dynamics $f$, reward $r$, cost $c$ and epistemic uncertainty $\sigma_n, \forall n \geq 0$ are Lipschitz continuous with known constants $L_f, L_r, L_c, L_\sigma$.*

**Assumption 3** (Regularity (Berkenkamp et al., 2017)). *The unknown dynamics $f$ lie in an RKHS associated with a kernel $k$ and have a component-wise bounded norm $||f_i||_k \leq B$ with a known constant $B$. Moreover, we assume the kernel $k$ to be bounded by $k((s, a), (s, a)) \leq k_{max}, \forall (s, a) \in \mathcal{S} \times \mathcal{A}$.*

Intuitively, this assumption means that the complexity of $f$ is controlled by the kernel, which in turn enables generalization from limited data. Our analysis uses Assumption 3 to relate the statistical complexity of learning the dynamics to the number of online episodes required for convergence. In Section 5, we model it using neural networks that can learn features to represent $f$ accurately.

The goal in our setting is to find a stationary, stochastic policy $\pi_c^*(a_t|s_t)$ for the true dynamics $f$ that maximizes value function $V_r^\pi(s) \coloneqq \mathbb{E}_\pi[\sum_{t=0}^\infty \gamma^t r(s_t, a_t) \mid s_0 = s]$, while ensuring that the cost value function $V_c^\pi(s) \coloneqq \mathbb{E}_\pi[\sum_{t=0}^\infty \gamma^t c(s_t, a_t) \mid s_0 = s]$ remains below a safety budget $d \in \mathbb{R}_{\geq 0}$:

$$\pi_c^* \in \arg\max_\pi J_r(\pi, f) \coloneqq \mathbb{E}_{s_0 \sim \rho_0}[V_r^\pi(s_0)] \quad \text{s.t.} \quad J_c(\pi, f) \coloneqq \mathbb{E}_{s_0 \sim \rho_0}[V_c^\pi(s_0)] \leq d. \tag{2}$$

We hereafter consider in our theoretical analysis strict feasibility of $\pi_c^*$; namely that an optimal policy of the constrained problem does not lie on the boundary of the feasible set but may be arbitrarily close, such that $J_c(\pi_c^*, f) < d$. Equation (2) naturally decouples the objective from safety, so each can be solved independently. This explicit separation prevents agents from exploiting the reward function to achieve high performance at the expense of harmful behaviors, a problem commonly referred to as *reward hacking* in the AI safety literature (Amodei et al., 2016).

**Safe online learning.** Since knowledge of the dynamics is limited a priori, the agent must learn them through online interaction. We consider learning in an *episodic* setting, where in each episode $n = 1, \ldots N$, infinite-horizon trajectories are truncated after $T_n$ steps and the agent is reset to a new initial state (Puterman, 2014; Sutton & Barto, 2015). The agent deploys a policy $\pi_n$, collecting transitions $\mathcal{D}_n = \{(s_t, a_t, s_{t+1})_{t=1,\ldots T_n}\}$. The optimality gap due to incomplete knowledge over episodes is quantified by cumulative regret, namely, the difference in performance between $\pi_n$ and the optimal policy given full access to $\mathcal{M}_c$:

$$R(N) = \sum_{n=1}^N \left(J_r(\pi_c^*, f) - J_r(\pi_n, f)\right) \quad \text{s.t.} \quad J_c(\pi_n, f) \leq d, \forall n \in \{1, \ldots, N\}. \tag{3}$$

The crux of safe exploration is that unlike performance, where suboptimality during learning is tolerable, safety must be maintained throughout all $n$ episodes.

**Probabilistic world models.** Given full knowledge of the CMDP $\mathcal{M}_c$, the agent can solve Equation (3) with no regret. In practice however, the agent has only limited knowledge about $\mathcal{M}_c$. We capture this uncertainty by defining a *set of plausible models* of the dynamics $\mathcal{F}_n$ for each iteration $n$.

**Definition 1.** *We define the set of plausible models in episode $n'$ as $\mathcal{F}_{n'} := \{\tilde{f} \mid |\tilde{f} - \mu_{n'}| \leq \beta_{n'}\sigma_{n'}\}$, described by a nominal model $\mu_{n'} : \mathcal{S} \times \mathcal{A} \to \mathbb{R}^{d_\mathcal{S}}$ and the uncertainty $\sigma_{n'} : \mathcal{S} \times \mathcal{A} \to \mathbb{R}^{d_\mathcal{S}}$ given data $\mathcal{D}_{1:n'}$. This implies the model of $f$ is always well-calibrated (Curi et al., 2020), if there exist $\forall n' \leq n : \beta_{n'} \in \mathbb{R}_{>0}$ such that, with probability of at least $1 - \delta$ it holds $\forall (s,a) \in \mathcal{S} \times \mathcal{A}$ that $f(s,a) \in \mathcal{F}_n := \bigcap_{n'=0}^{n} \mathcal{F}_{n'}$.*

Definition 1 formalizes the agent's *epistemic* uncertainty about $\mathcal{M}_c$. At each iteration $n$, the agent maintains a confidence set that is statistically consistent with the observed data. This is done by learning a model with nominal dynamics $\mu_n : \mathcal{S} \times \mathcal{A} \to \mathbb{R}^{d_\mathcal{S}}$ and epistemic uncertainty $\sigma_n : \mathcal{S} \times \mathcal{A} \to \mathbb{R}^{d_\mathcal{S}}$ for every $n = 1, \ldots, N$, using transitions from all previously collected trajectories $\mathcal{D}_{1:n}$. This type of statistical models can be learned in practice via various (approximate) Bayesian inference methods (e.g., Rasmussen & Wiliams, 2006; Lakshminarayanan et al., 2017; Liu & Wang, 2016). Additionally, calibration can be achieved empirically with post-hoc recalibration techniques (Kuleshov et al., 2018).

**Pessimistic policy priors.** The set $\mathcal{F}_0$ has a unique interpretation: it formally represents the agent's prior knowledge about the environment. For instance, in fully data-driven settings, given an offline dataset $\mathcal{D}_0$, a model $\mathcal{F}_0$ can be learned from $\mathcal{D}_0$ and then used to construct a conservative MDP that penalizes the agent when it goes outside of the data distribution (Levine et al., 2020; Yu et al., 2020). Alternatively, in simulator-driven workflows, one has access to a nominal model $\mu_0 : \mathcal{S} \times \mathcal{A} \to \mathbb{R}^{d_\mathcal{S}}$, and a bounded 'sim-to-real gap' $u : \mathcal{S} \times \mathcal{A} \to \mathbb{R}^{d_\mathcal{S}}$, giving $f \in \mathcal{F}_0 = \{\tilde{f} \mid |\tilde{f} - \mu_0| \leq u\}$. Probabilistic models of this sort formalize the conservatism needed for safe prior policies.

**Assumption 4** (Safe policy prior). *We assume access to a pessimistic policy prior $\hat{\pi} : \mathcal{S} \to \mathcal{A}$ that satisfies the constraint in Equation (2) for all dynamics $f \in \mathcal{F}_0$ such that under the true dynamics we have $V_c^{\hat{\pi}}(s) \leq V_c^{\pi_c^*}(s)$ for all states with probability at least $1 - \delta$.*

Learning such "worst-case" policies builds on the principle of pessimism in the face of uncertainty and has been extensively studied in the robust RL literature (Kitamura et al., 2024; Zhang et al.; Shi et al.; Shi & Chi, 2024), offline safe RL (Jin et al., 2021; Liu et al., 2023; Zheng et al., 2024; Wachi et al., 2025), safe learning from demonstrations (Schlaginhaufen & Kamgarpour, 2023; Lindner et al., 2024) and when transferring from simulators (Queeney & Benosman, 2023; As et al., 2025a). In our experiments in Section 5, we leverage a few of these methods and demonstrate the effectiveness of combining them with safe online learning.

## 4 SOOPER: SAFE ONLINE OPTIMISM FOR PESSIMISTIC EXPANSION IN RL

In a nutshell, SOOPER can be explained by two modes of operation: **(i)** safe online data collection by pessimistically invoking a safe prior policy early enough so as to ensure constraint satisfaction; **(ii)** optimistic planning in "simulation" via model-based rollouts, driving exploration of the unknown dynamics through intrinsic rewards.

**Quantifying pessimism.** We define the pessimistic cost value function $\bar{V}_c^{\hat{\pi}}$ for the policy prior $\hat{\pi}$:

$$\bar{V}_c^{\hat{\pi}}(s) := \max_{\tilde{f} \in \mathcal{F}_n} \mathbb{E}_{\hat{\pi}}\left[\sum_{t=0}^{\infty} \gamma^t c(s_t, a_t) \,\middle|\, s_0 = s\right], \quad s_{t+1} = \tilde{f}(s_t, a_t) + \omega_t. \tag{4}$$

By definition, Equation (4) considers the worst-case model $\tilde{f}$ in the well-calibrated set $\mathcal{F}_n$ (per Definition 1), which implies that $\bar{V}_c^{\hat{\pi}}(s)$ upper-bounds $V_c^{\hat{\pi}}(s)$ for all $s \in \mathcal{S}$ with probability $1 - \delta$. We later formally show that, as the number of episodes $n$ increases, the agent refines its model $\mathcal{F}_n$ and tightens the pessimistic upper-confidence bound $\bar{V}_c^{\hat{\pi}}$ on the expected accumulated cost.

**A tractable upper bound.** Computing the pessimistic cost-value $\bar{V}_c^{\hat{\pi}}$ in Equation (4) is intractable due to the maximization over $\mathcal{F}_n$. Instead, we upper-bound $\bar{V}_c^{\hat{\pi}}$ by augmenting the cost function with

an uncertainty penalty such that

$$Q_{c,n}^{\hat{\pi}}(s,a) := \mathbb{E}_{\hat{\pi}} \left[ \sum_{t=0}^{\infty} \gamma^t (c(s_t,a_t) + \lambda_{\text{pessimism}} \|\sigma_n(s_t,a_t)\|) \, \Big| \, s_0 = s, a_0 = a \right],$$

$$\bar{V}_c^{\hat{\pi}}(s) \leq V_{c,n}^{\hat{\pi}}(s) = \mathbb{E}_{a \sim \hat{\pi}} \left[ Q_{c,n}^{\hat{\pi}}(s,a) \right], \quad s_{t+1} = \mu_n(s_t,a_t) + \omega_t. \tag{5}$$

Crucially, directly penalizing the cost function avoids finding the worst-case model within $\mathcal{F}_n$, and lends itself gracefully to existing temporal difference techniques for value function learning. The derivation of $\lambda_{\text{pessimism}}$ and the proof for this upper bound are deferred to Lemma 2 in Appendix A.

**Online cost tracking.** SOOPER operates in an online fashion, where for each of the trajectory rollouts, it uses $Q_{c,n}^{\hat{\pi}}$ to predict when $a_t \sim \pi_n(\cdot|s_t)$ is potentially unsafe. Specifically, this may occur if the sum of the accumulated cost until $t$, i.e., $c_{<t} = \sum_{\tau=0}^{t-1} \gamma^\tau c(s_\tau,a_\tau)$ and the pessimistic cost value $\gamma^t Q_{c,n}^{\hat{\pi}}(s_t,a_t)$ exceeds the safety budget $d$. We exploit this idea in our design of Algorithm 1. In Theorem 1 we show that this design guarantees safety throughout all episodes $n = 1, \ldots, N$.

---

**Algorithm 1** Safe Rollout via Online Cost Tracking

---

**Require:** Pessimistic cost value $Q_{c,n}^{\hat{\pi}}$, policy $\pi_n$, safe policy prior $\hat{\pi}$, and safety budget $d$
1: Initialize accumulated cost $c_{<0} = 0$ and experience buffer $\mathcal{D}_n = \emptyset$
2: **for** step $t = 1, \ldots, T_n$ **do**
3:     Execute $a_t \sim \bar{\pi}_n(a_t|s_t, c_{<t}, Q_{c,n}^{\hat{\pi}})$, observe $s_t, a_t, s_{t+1}$           ▷ Theorem 1
4:     Append the tuple $(s_t, a_t, s_{t+1})$ to $\mathcal{D}_n$
5:     Update accumulated cost: $c_{<t+1} = c_{<t} + \gamma^t c_t$
6: **end for**
7: **return** $\mathcal{D}_n$

---

**Theorem 1** (Safety guarantee). *Suppose Assumptions 1 to 4 hold and $\mathcal{F}_n$ is well-calibrated $\forall n = 1, \ldots, N$ according to Definition 1. If actions are executed for all timesteps $t$ according to*

$$\bar{\pi}_n(a_t|s_t, c_{<t}, Q_{c,n}^{\hat{\pi}}) := \begin{cases} \pi_n(\cdot|s_t) & \text{if } \Phi(s_t, a_t, c_{<t}, Q_{c,n}^{\hat{\pi}}) < d \\ \hat{\pi}(s_t) & \text{otherwise,} \end{cases}$$

*where $\Phi$ is the discounted sum of the realized accumulated cost $c_{<t} := \sum_{\tau=0}^{t-1} \gamma^\tau c(s_\tau, a_\tau)$ until $t - 1$ and the pessimistic cost value $Q_{c,n}^{\hat{\pi}}$ such that $\Phi(s_t, a_t, c_{<t}, Q_{c,n}^{\hat{\pi}}) := c_{<t} + \gamma^t Q_{c,n}^{\hat{\pi}}(s_t, a_t)$ with factor $\lambda_{pessimism} = \bar{C} \frac{\gamma}{1-\gamma} \frac{(1+\sqrt{d_S})\beta_N(\delta)}{\sigma}$, where $\bar{C} = \max\{C_{max}, k_{max}\}$. Then, Algorithm 1 satisfies the safety constraint with probability $1 - \delta$ on $f$ for every episode $n = 1, \ldots, N$.*

We refer to Appendix A for the formal proof. Monitoring costs online allows the agent to execute $\pi_n$ until $\hat{\pi}$ is required. This enables the agent to visit state-actions that $\hat{\pi}$ would avoid, since those would be deemed unsafe by $\bar{V}_c^{\hat{\pi}}(s_0)$.

**Simulated exploration.** While Theorem 1 guarantees safety during *online* rollouts, the policy $\pi_n$ may explore potentially unsafe behaviors by using simulated rollouts on the learned world model. This allows SOOPER to train $\pi_n$ on an *unconstrained* "planning MDP" $\widetilde{\mathcal{M}}$, that differs from the real CMDP $\mathcal{M}_c$ in that it reaches a terminal state $s_\dagger$ after those state-actions that would trigger $\hat{\pi}$ during online deployment:

$$\tilde{p}(s_{t+1} \mid s_t, a_t) := \begin{cases} \mathbf{1}\{s_{t+1} = s_\dagger\} & \text{if } \Phi(a_t, s_t, c_{<t}, Q_{c,n}^{\hat{\pi}}) \geq d \text{ or } s_t = s_\dagger, \\ p(s_{t+1} \mid s_t, a_t) & \text{otherwise.} \end{cases} \tag{6}$$

The resulting termination signal encourages the agent to find policies that outperform $\hat{\pi}$ if possible. This is done by assigning the terminal reward on $\widetilde{\mathcal{M}}$ to the pessimistic value $\underline{V}_r^{\hat{\pi}}$ of $\hat{\pi}$ upon termination:

$$\tilde{r}(s_t, a_t) := \begin{cases} \underline{V}_r^{\hat{\pi}}(s_t) & \text{if } \Phi(a_t, s_t, c_{<t}, Q_{c,n}^{\hat{\pi}}) \geq d, \\ 0 & \text{if } s_t = s_\dagger, \\ r(s_t, a_t) & \text{otherwise,} \end{cases} \tag{7}$$

whereby,

$$\underline{V}_r^{\hat{\pi}}(s) := \min_{\tilde{f} \in \mathcal{F}_n} \mathbb{E}_{\hat{\pi}} \left[ \sum_{t=0}^{\infty} \gamma^t r(s_t, a_t) \, \Big| \, s_0 = s \right], \quad s_{t+1} = \tilde{f}(s_t, a_t) + \omega_t. \tag{8}$$

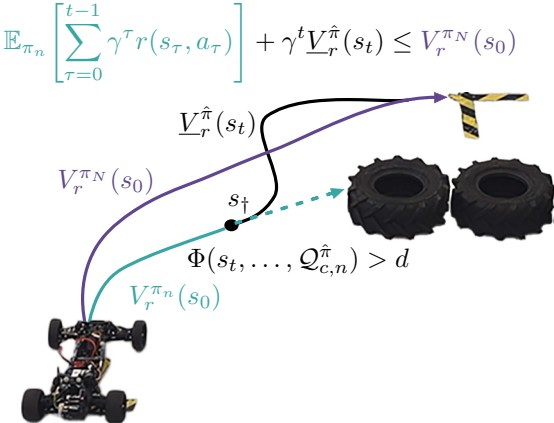

Figure 1: The agent's goal is to reach the cross marker while avoiding obstacles (tires). The agent deploys a policy $\pi_n$, however at time $t$, it switches to the prior policy $\hat{\pi}$, to maintain the safety criterion of Algorithm 1. The prior policy $\hat{\pi}$ ensures safety by following a conservative route, but sacrifices performance, resulting in a lower return $\underline{V}_r^{\hat{\pi}}(s_t)$. The trajectory of iteration $n$ is recorded to improve models of subsequent iterations. After $N$ iterations, as more data is gathered, the agent learns a more rewarding trajectory via model-generated rollouts in $\widetilde{\mathcal{M}}$. See provided video.

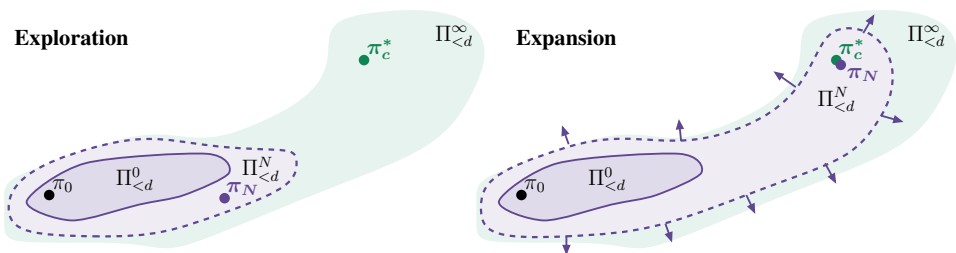

Figure 2: We denote the implicit set of safe policies in iteration $n$ by $\Pi_{<d}^n$, based on the learned model $\mathcal{F}_n$. **Left:** exploration-exploitation in constrained tasks may not find an optimal policy because search is limited to $\Pi_{<d}^n$. **Right:** expansion proactively enlarges the safe set and reaches the optimum.

This choice of the terminal reward keeps the reward structure consistent with the rewards encountered during online rollouts of $\hat{\pi}$ on $f$, unlike prior works that treat the terminal reward as a hyperparameter (Thomas et al., 2021; Wagener et al., 2021). Figure 1 illustrates a concrete example of how this design encourages the agent to learn to outperform the prior policy.

**Exploration-exploitation and expansion.** Probabilistic models have proven particularly effective in the unconstrained setting. Optimistic planning with these models enables a principled balance between exploration and exploitation, forming the basis of the celebrated UCRL algorithm (Auer & Ortner, 2006; Curi et al., 2020). However, when safety must hold at every iteration $n$, exploration-exploitation is restricted to the set of policies that are safe with high probability. In particular, this set consists only those policies that satisfy the constraint under the *worst-case* model in $\mathcal{F}_n$. Crucially, due to pessimism, an optimal policy $\pi_c^*$ may not lie in this set initially, as shown in Figure 2. Nonetheless, this set can be *expanded* by updating the model using the collected measurements from the environment, even in those regions where rewards are predicted to be low under the agent's model at iteration $n$. Prior works typically follow an expand-then-explore-exploit strategy (Sui et al., 2015; Berkenkamp et al., 2017; Bura et al., 2022; As et al., 2025b), which begins by learning the dynamics with reward-free trajectory sampling to expand the safe set, followed by exploration-exploitation to find an optimal and safe policy. This approach has two limitations: **(i)** it requires agents to first learn an auxiliary policy unrelated to the task, leading to wasted compute and exploration; **(ii)** it restricts the theoretical analysis to simple regret, allowing arbitrarily poor performance during exploration as long as the final policy is near-optimal.

**A unified objective.** In contrast, SOOPER addresses the exploration-exploitation-expansion dilemma through *intrinsic rewards* (Sukhija et al., 2025) with a single tractable objective:

$$\pi_n \in \arg\max_\pi \mathbb{E}_\pi \left[ \sum_{t=0}^{\infty} \gamma^t \tilde{r}(s_t, a_t) + (\gamma^t \lambda_{\text{explore}} + \sqrt{\gamma^t} \lambda_{\text{expand}}) \|\sigma_n(s_t, a_t)\| \right], \qquad (9)$$

$$s_{t+1} = \mu_n(s_t, a_t) + \omega_t, \quad a_t \sim \pi(\cdot|s_t), \quad s_0 \sim \rho_0(\cdot),$$

where $\lambda_{\text{explore}}$ and $\lambda_{\text{expand}}$ are weighting factors for the respective components of the intrinsic reward bonus, encouraging exploration and expansion. In Lemma 12 in Appendix B, we derive these quantities in closed form and prove that Equation (9) indeed enables expansion. Importantly, since

we only augment the reward with exploration bonuses, an approximate solution to Equation (9) can be computed efficiently.

**Converging to the optimum.** By iteratively updating the model $\mathcal{F}_n$ and planning according to Equation (9) in each episode, SOOPER achieves *sublinear cumulative regret* and therefore converges for a sufficient number of episodes $N$ in Algorithm 2 according to Theorem 2.

---

**Algorithm 2** Safe Online Optimism for Pessimistic Expansion in RL (SOOPER)

---

**Require:** Initial dynamics set $\mathcal{F}_0$, pessimistic cost value $Q_{c,0}^{\hat{\pi}}$, policy prior $\hat{\pi}$, safety budget $d$
1: Initialize experience buffer $\mathcal{D}$
2: **for** episode $n = 1, \ldots, N$ **do**
3:     Execute Algorithm 1 to collect $\mathcal{D}_n$                                 ▷ On the real environment
4:     Update $\mathcal{F}_n \leftarrow \mathcal{D}_{1:n}$                                            ▷ Refine model
5:     Obtain $\pi_n, Q_{c,n}^{\hat{\pi}}$ via Equations (5) and (9)      ▷ Model-based exploration and planning
6: **end for**
7: **return** $\pi_n$

---

**Theorem 2** (Sublinear cumulative regret). *Suppose Assumptions 1 to 4 hold and the model $\mathcal{F}_n$ is well calibrated according to Definition 1. Then, Algorithm 2 guarantees with probability $1 - \delta$*

$$R(N) = \sum_{n=1}^{N} \left( J_r(\pi_c^*, f) - J_r(\bar{\pi}_n, f) \right) \leq \mathcal{O}\left( \Gamma_{N \log(N)}^{7/2} \sqrt{N} \right),$$

$$\text{and } J_c(\bar{\pi}_n, f) \leq d, \; \forall n \in \{1, \ldots, N\},$$

*that is, the cumulative regret grows sub-linearly in $N$, dependent on the maximal information gain $\Gamma_{N \log(N)}$, while satisfying the constraint throughout learning by Theorem 1.*

**Proof sketch.** Previous works explicitly expand the safe set of policies via pure exploration. This restricts them to a simple regret analysis, since performance can be evaluated *only after* the reward-free expansion phase. SOOPER optimizes a single objective, thus expansion occurs *implicitly, during learning the task*. This allows us to extend the analysis to cumulative regret, providing guarantees on the agent's performance throughout learning. We bound the cumulative regret in Equation (3) by decomposing the performance gap in each iteration $n$ into two terms (Lemma 4). **(i)** The first term corresponds to the regret on the planning MDP $\widetilde{\mathcal{M}}$ (cf. Equations (6) and (7)). Specifically, we compare the performance difference between an optimal policy of $\widetilde{\mathcal{M}}$ and the learned policy $\pi_n$. Since this setting is purely unconstrained, we bound this term following the regret analysis for unconstrained MDPs of Kakade et al. (2020). **(ii)** Next, we analyze the performance gap due to invoking the prior policy $\hat{\pi}$ (Algorithm 1). In particular, we study the performance of executing $\pi_c^*$, an optimal policy of the true CMDP $\mathcal{M}_c$, however under the safety criterion of Algorithm 1. Intuitively, due to the pessimism of $Q_{c,n}^{\hat{\pi}}$, instead of performing optimally with $\pi_c^*$, Algorithm 1 invokes $\hat{\pi}$, thus leading to suboptimal performance. The gap in performance between executing $\pi_c^*$ freely and under Algorithm 1 is bounded by the model uncertainty at iteration $n$ (Lemmas 6 and 8). The key intuition is that since $\pi_c^*$ is in fact safe, as model uncertainty shrinks, Algorithm 1 needs to invoke $\hat{\pi}$ less, hence converging to optimal performance. Finally, we show that model uncertainty shrinks as $n$ grows under Assumption 3 due to Chowdhury & Gopalan (2017) and that the sum of the above two terms is upper-bounded by the objective in Equation (9) in each episode $n$. The complete proof is detailed in Appendix B.

**Practical implementation.** Our practical implementation follows closely Algorithm 2. We instantiate a "deep" version of Algorithm 2 that scales to high-dimensional continuous control tasks, following a model-based actor-critic architecture similar to MBPO (Janner et al., 2019a). Line 4 is implemented via a probabilistic ensemble of neural-networks (Chua et al., 2018). In our practical implementation we additionally learn the reward and cost functions (see Appendix C). The standard deviation across ensemble predictions is used to estimate the epistemic uncertainty $\sigma_n$ (Depeweg et al., 2018). We found this approach to deliver reliable uncertainty estimates in practice (cf. Section 5 and Appendix D). In Line 5, we solve Equations (5) and (9) using model-generated rollouts and by performing a fixed number of actor-critic updates as described by Janner et al. (2019a). Adapting MBPO to our setting is straightforward, as it only requires wrapping the model predictions within the MDP defined by $\widetilde{\mathcal{M}}$. This suggests that SOOPER is readily extensible to future advancements in deep RL.

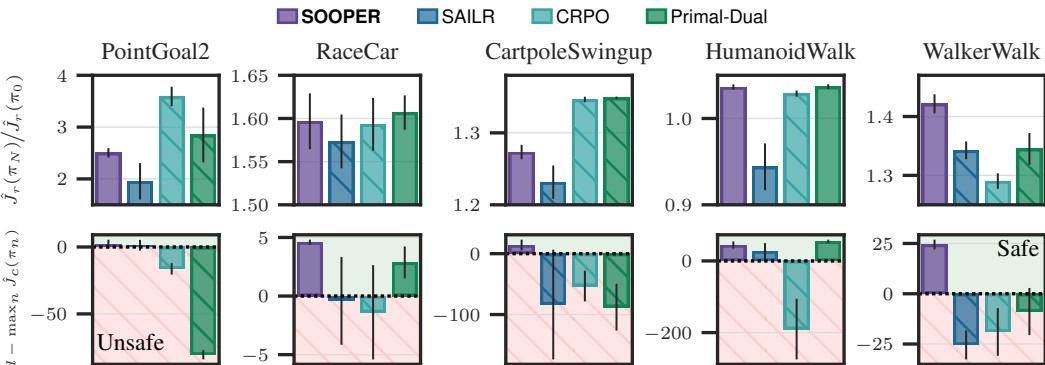

Figure 3: Performance improvement over the baseline policy and largest constraint recorded throughout learning. Among all methods, SOOPER remains safe in all tasks while consistently outperforming or being on par with the other baselines when they satisfy the constraints.

## 5 EXPERIMENTS

We now present our empirical results, demonstrating SOOPER's safety and performance guarantees in practice. In the Appendix D, we provide additional ablations, illustrating how each component of SOOPER plays a key role in attaining good performance while maintaining safety throughout learning.

**Setup.** In all experiments, we use the model-based architecture described in Section 4. We compare SOOPER with the following baselines: **(i)** SAILR (Wagener et al., 2021), a state-of-the-art RL algorithm for safe exploration. **(ii)** CRPO (Xu et al., 2020), a CMDP solver without safety guarantees during learning. **(iii)** Similarly, Primal-Dual (Bertsekas, 2016) is a constrained optimization algorithm, which is commonly used as a CMDP solver. CRPO and Primal-Dual serve as natural baselines, as they represent the standard approaches practitioners would typically adopt when fine-tuning policies initialized from simulation or offline training. Unless specified otherwise, we use a batch of 128 trajectories to obtain empirical estimates of the undiscounted cumulative rewards $\hat{J}_r(\pi)$ and costs $\hat{J}_c(\pi)$ and run each experiment for five random seeds. We slightly abuse notation, so that for SOOPER, $\pi$ denotes the policy induced by Algorithm 1 and for SAILR the policy induced by their "advantage-based intervention". All baselines are initialized with the same task-specific prior policy.

**Safe transfer under dynamics mismatch.** We study policies that are trained under a mismatch in the dynamics on five tasks from RWRL (Dulac-Arnold et al., 2021), SafetyGym (Ray et al., 2019) and RaceCar (Kabzan et al., 2020). We provide additional details in Appendices E to G about how mismatches in the dynamics are implemented for each task. We refer to Appendix D for additional details on how we train the pessimistic prior policies. We train prior policies under the shifted dynamics ($\mu_0$, cf. Section 3) and continue learning on the true dynamics. Figure 3 presents the largest cumulative cost recorded on the true dynamics throughout training, along with the relative performance improvement at the end of training compared to the policy prior before online learning. As shown, SOOPER satisfies the constraints throughout learning in all of the tasks. For each task, among all algorithms that satisfy the constraints, SOOPER is consistently on par or outperforms the baselines. This demonstrates SOOPER's ability to transfer safely under distribution shifts.

**Scaling to vision control.** Next, we demonstrate that SOOPER scales to control tasks with image-based observations. Specifically, we use the CartpoleSwingup task from Dulac-Arnold et al. (2021) and train policies that operate on three temporally-stacked 64×64 grayscale images as depicted in Figure 4. As in Figure 3, we introduce a distribution shift in the dynamics and pretrain a vision policy under the perturbed dynamics using DrQ (Yarats et al., 2021). We then train SOOPER by learning the dynamics model directly on the embeddings produced by the pretrained DrQ vision encoder. Figure 5b shows that even when trained on these embeddings, SOOPER satisfies the safety constraints and achieves near-optimal performance.

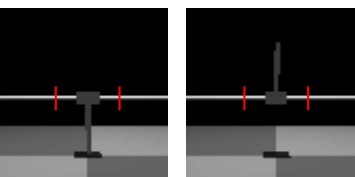

Figure 4: Image observations for the CartpoleSwingup task. The goal is to swing the pendulum to the upright position while avoiding the range outside the vertical red lines.

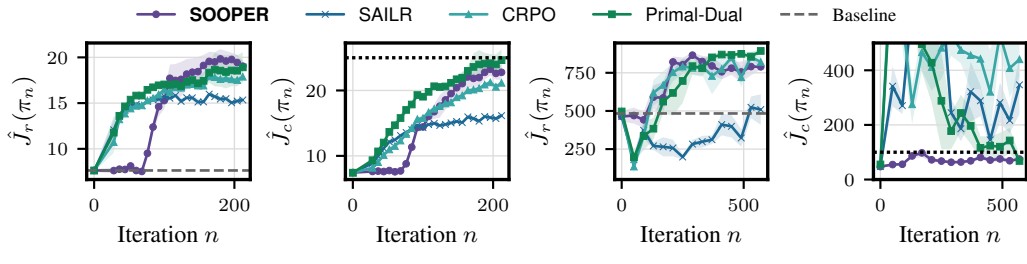

(a) Safe online learning from an offline-trained policy.          (b) Vision control task.

Figure 5: Learning curves of the objective and constraint when learning from offline and vision policies. SOOPER satisfies the constraints while significantly improving over the initial prior policy.

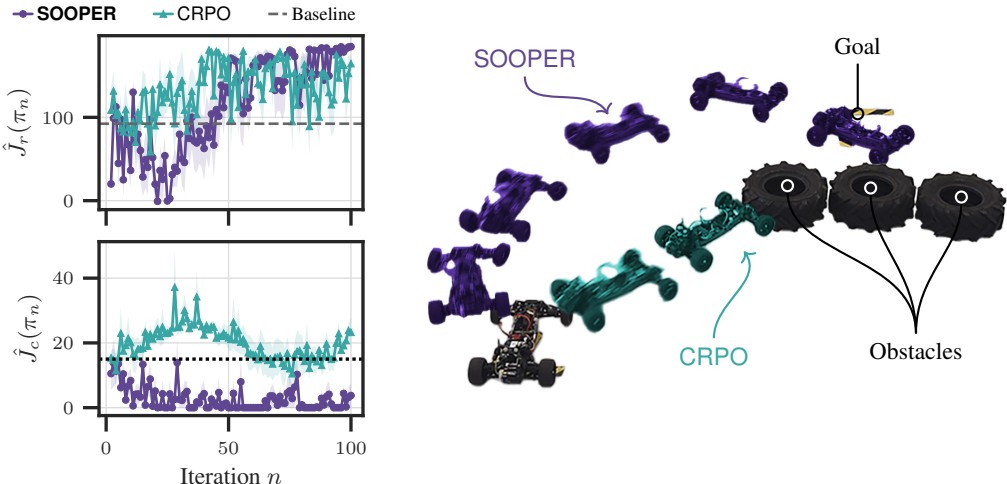

Figure 6: Safe exploration on real hardware with SOOPER. We report the mean and standard error across five seeds of the objective and constraint measured on the real system. SOOPER learns to improve over the prior policy while satisfying the constraints throughout learning. Video of training.

**Safe offline-to-online.** We evaluate SOOPER when initialized with a conservative policy using only offline data. To this end, we collect a fixed offline data of 2M transitions from the PointGoal1 task of SafetyGym. We train a pessimistic prior policy offline by following MOPO (Yu et al., 2020), a well-established model-based offline RL algorithm. We use a Primal-Dual optimizer to solve the constraint optimization problem with this offline data. We found this approach to yield policies that satisfy the constraint while still delivering nontrivial performance. We use these policies to initialize all the baselines for our experiments. Figure 5a presents the objective and constraint during online learning. Notably, SOOPER outperforms all the baselines while maintaining safety.

**Safe online learning on real hardware.** Finally, we evaluate SOOPER when trained online *directly on real-world robotic hardware*: a highly dynamic, remote-controlled race car operating at 60 Hz. The task is to reach a designated goal position located behind three obstacles (tires) that must be avoided (illustrated in Figure 6). This setting is particularly challenging due to stochasticity introduced by high-frequency control and delays in both actuation and motion-capture measurements. We initialize training with a prior policy derived from a first-principles simulator (Kabzan et al., 2020), and then fine-tune it on the physical system using trajectories collected in real time. We repeat the experiment with five random seeds and report cumulative rewards and costs after each iteration in Figure 6. As shown, after training, SOOPER is roughly twice as good as the prior policy in terms of rewards while satisfying the safety constraint throughout learning. In Appendix D, we extend this experiment to the offline setting, and report results when using a prior policy trained offline with data collected from the real system.

## 6 CONCLUSION

In this work, we address a critical challenge in systems that learn autonomously, namely, maintaining safety during learning. We present SOOPER, a novel algorithm that provably improves performance over policies trained in simulation or offline, while satisfying safety constraints throughout learning. SOOPER is simple to implement, achieves state-of-the-art performance and consistently maintains safety when deployed in practice. In addition, we establish a new theoretical result on the cumulative regret, enabling a guarantee of good performance during learning and not only at the last iteration. These contributions advance our understanding of safe exploration and make an important step towards deployable RL. Nonetheless, many important challenges remain. In terms of theory and methodology, extending our work to high-probability constraints over particular states and developing new methods for the non-episodic setting, where agents learn from a single trajectory without resets, are two challenging yet crucial problems. On the practical side, considering more complex tasks with more powerful models is a promising direction for future work.

### ACKNOWLEDGMENTS

We would like to thank Armin Lederer, Bruce D. Lee and Dongho Kang for insightful discussions during the development of this project. We thank the anonymous reviewers for their valuable comments and suggestions. This project has received funding from a grant of the Hasler foundation (grant no. 21039) and the ETH AI Center.

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

# A  SAFETY GUARANTEE

First, we derive in Lemma 1 that the pessimistic safety-condition with online cost tracking (Theorem 1) guarantees safety given the true pessimistic cost value function $\bar{V}_c^{\hat{\pi}}$ of Equation (4).

**Lemma 1.** *Suppose Assumptions 1 to 4 hold and $\mathcal{F}_n$ is well-calibrated $\forall n = 1, \ldots, N$ according to Definition 1. Given the safe prior $\hat{\pi}$, the rollout of the policy $\pi_n$ satisfies the safety constraint with probability $1 - \delta$ on $f$ for every episode $n = 1, \ldots, N$, if the actions are executed for all timesteps $t$ according to the pessimistic safety-condition:*

$$\tilde{\pi}_n(a_t|s_t, c_{<t}, \mathcal{F}_n) = \begin{cases} \pi_n(\cdot|s_t) & \text{if } \bar{\Phi}(s_t, a_t, c_{<t}, \mathcal{F}_n) < d \\ \hat{\pi}(s_t) & \text{otherwise,} \end{cases}$$

*where*

$$\bar{\Phi}(s_t, a_t, c_{<t}, \mathcal{F}_n) = c_{<t} + \gamma^t c(s_t, a_t) + \gamma^{t+1} \max_{\tilde{f} \in \mathcal{F}_n} \mathbb{E}_\omega \left[ \bar{V}_c^{\hat{\pi}}(\tilde{f}(s_t, a_t) + \omega_t) \right], \qquad (10)$$

*indicates the sum of the accumulated costs $c_{<t} = \sum_{\tau=0}^{t-1} \gamma^t c(s_t, a_t)$ until $t-1$, the immediate cost at $t$, and the worst-case pessimistic cost value $\bar{V}_c^{\hat{\pi}}$ defined in Equation (5).*

*Proof.* We prove the safe rollout of $\pi_n$ for the unknown $f$ by induction.

1. Base Case ($t = 0$): By definition of $\tilde{\pi}_n$, we only execute $a_0 \sim \pi_n(\cdot|s_0)$ if $\bar{\Phi}(s_0, a_0, c_{<0}, \mathcal{F}_n) < d$ with $c_{<0} = 0$ and otherwise the safe prior $\hat{\pi}$. By definition of $\bar{\Phi}$, either $a_0$ is safe to execute with

$$d > c(s_0, a_0) + \gamma \max_{\tilde{f} \in \mathcal{F}_n} \mathbb{E}_\omega \left[ V_c^{\hat{\pi}}(\tilde{f}(s_0, a_0) + \omega_0) \right], \qquad (11)$$

   and otherwise executing $a_0 = \hat{\pi}(s_0)$ is safe since $\bar{V}_c^{\hat{\pi}}(s_0) < d$ by Assumption 4.

2. Inductive Step: Given the agent is safe up to timestep $t - 1$, i.e.,

$$c_{<t-1} + \gamma^{t-1} c(s_{t-1}, a_{t-1}) + \gamma^t \max_{\tilde{f} \in \mathcal{F}_n} \mathbb{E}_\omega \left[ V_c^{\hat{\pi}}(\tilde{f}(s_{t-1}, a_{t-1}) + \omega_{t-1}) \right] < d, \qquad (12)$$

   we show constraint satisfaction for time $t$, satisfying

$$d > c_{<t} + \gamma^t c(s_t, a_t) + \gamma^{t+1} \mathbb{E}_\omega \left[ V_c^{\hat{\pi}}(f(s_t, a_t) + \omega_t) \right]. \qquad (13)$$

   Using $\tilde{\pi}_n$, we either execute $a_t$ if Equation (13) is satisfied and therefore is safe by definition. Otherwise, we invoke the prior $\hat{a}_t = \hat{\pi}(s_t)$. Using $c_{<t} = c_{<t-1} + \gamma^{t-1} c(s_{t-1}, a_{t-1})$, we obtain

$$c_{<t} + \gamma^t c(s_t, \hat{a}_t) + \gamma^{t+1} \max_{\tilde{f} \in \mathcal{F}_n} \mathbb{E}_\omega \left[ \bar{V}_c^{\hat{\pi}}(\tilde{f}(s_t, \hat{a}_t) + \omega_t) \right] \qquad (14)$$

$$= c_{<t-1} + \gamma^{t-1} c(s_{t-1}, a_{t-1}) + \gamma^t c(s_t, \hat{a}_t) + \gamma^{t+1} \max_{\tilde{f} \in \mathcal{F}_n} \mathbb{E}_\omega \left[ \bar{V}_c^{\hat{\pi}}(\tilde{f}(s_t, \hat{a}_t) + \omega_t) \right] \qquad (15)$$

$$\leq c_{<t-1} + \gamma^{t-1} c(s_{t-1}, a_{t-1}) + \gamma^t \max_{\tilde{f} \in \mathcal{F}_n} \mathbb{E}_\omega \left[ V_c^{\hat{\pi}}(\tilde{f}(s_{t-1}, a_{t-1}) + \omega_{t-1}) \right] < d. \qquad (16)$$

$\square$

Next, we show that adding the cost penalties in $Q_{c,n}^{\hat{\pi}}$ (Equation (5)) upper-bounds the true cost value, using Sukhija et al. (2025, Lemma A.5):

**Lemma 2.** *Suppose Assumptions 1 to 3 hold and $\mathcal{F}_n$ is well-calibrated by Definition 1. Given*

$$\gamma^{t+1} \max_{\tilde{f} \in \mathcal{F}_n} \mathbb{E}_\omega \left[ \bar{V}_c^{\hat{\pi}}(\tilde{f}(s_t, a_t) + \omega_t) \right] = \mathbb{E}_{\hat{\pi}} \left[ \sum_{\tau=t+1}^{\infty} \gamma^\tau c(s_\tau, a_\tau) \right]$$

$$\text{s.t. } s_{\tau+1} = \tilde{f}(s_\tau, a_\tau) + \omega_\tau, \quad a_\tau = \hat{\pi}(s_\tau), \quad s_{t+1} = \tilde{f}(s_t, a_t) + \omega_t,$$

$$\lambda_{pessimism} = \bar{C} \frac{\gamma}{1 - \gamma} \frac{\left(1 + \sqrt{d_{\mathcal{S}}}\right) \beta_N(\delta)}{\sigma},$$

*then for all episodes $n = 1, \ldots, N$ with probability of $1 - \delta$:*

$$\gamma^{t+1} \max_{\tilde{f} \in \mathcal{F}_n} \mathbb{E}_\omega \left[ \bar{V}_c^{\hat{\pi}}(\tilde{f}(s_t, a_t) + \omega_t) \right] \leq \gamma^t \lambda_{pessimism} \| \sigma_n(s_t, a_t) \|$$

$$+ \mathbb{E}_{\hat{\pi}} \left[ \sum_{\tau=t+1}^{\infty} \gamma^\tau \left( c(s_\tau, a_\tau) + \lambda_{pessimism} \| \sigma_n(s_\tau, a_\tau) \| \right) \right],$$

$$\text{s.t. } s_{\tau+1} = \mu_n(s_\tau, a_\tau) + \omega_\tau, \quad a_\tau = \hat{\pi}(s_\tau), \quad s_{t+1} = \mu_n(s_t, a_t) + \omega_t.$$

*Proof.* Let us denote the value following the nominal dynamics $\mu_n$ of $\mathcal{F}_n$ as

$$\tilde{V}_c^{\hat{\pi}}(s) = \mathbb{E}_{\hat{\pi}} \left[ \sum_{t=0}^{\infty} \gamma^t c(s_t, a_t) \mid s_0 = s \right], \tag{17}$$

$$\text{s.t. } s_{t+1} = \mu_n(s_t, a_t) + \omega_t, \quad a_t = \hat{\pi}(s_t),$$

and the upper bound on the cost value following the worst dynamics $\tilde{f} \in \mathcal{F}_n$ by

$$\bar{V}_c^{\hat{\pi}}(s) = \max_{\tilde{f} \in \mathcal{F}_n} V_c^{\hat{\pi}}(s) = \mathbb{E}_{\hat{\pi}} \left[ \sum_{t=0}^{\infty} \gamma^t c(s_t, a_t) \mid s_0 = s \right], \tag{18}$$

$$\text{s.t. } s_{t+1} = \tilde{f}(s_t, a_t) + \omega_t, \quad a_t = \hat{\pi}(s_t).$$

Following Sukhija et al. (2025, Lemma A.5), the difference between these values is bounded by

$$\left| \bar{V}_c^{\hat{\pi}}(s) - \tilde{V}_c^{\hat{\pi}}(s) \right| \leq \lambda_{\text{pessimism}} \mathbb{E}_{\hat{\pi}} \left[ \sum_{t=0}^{\infty} \gamma^t \| \sigma_n(s_t, a_t) \| \mid s_0 = s \right], \tag{19}$$

$$\text{s.t. } s_{t+1} = \mu_n(s_t, a_t) + \omega_t, \quad a_t = \hat{\pi}(s_t).$$

Bringing $\tilde{V}_c^{\hat{\pi}}(s)$ to the other side and using the definition in Equation (17) results in:

$$\bar{V}_c^{\hat{\pi}}(s) \leq \tilde{V}_c^{\hat{\pi}}(s) + \lambda_{\text{pessimism}} \mathbb{E}_{\hat{\pi}} \left[ \sum_{t=0}^{\infty} \gamma^t \| \sigma_n(s_t, a_t) \| \mid s_0 = s \right], \tag{20}$$

$$= \mathbb{E}_{\hat{\pi}} \left[ \sum_{t=0}^{\infty} \gamma^t \left( c(s_t, a_t) + \lambda_{\text{pessimism}} \| \sigma_n(s_t, a_t) \| \right) \mid s_0 = s \right], \tag{21}$$

$$\text{s.t. } s_{t+1} = \mu_n(s_t, a_t) + \omega_t, \quad a_t = \hat{\pi}(s_t). \tag{22}$$

Further, we use this upper bound having $c(s_t, a_t)$ already observed and rewriting

$$\gamma^{t+1} \max_{\tilde{f} \in \mathcal{F}_n} \mathbb{E}_\omega \left[ \bar{V}_c^{\hat{\pi}}(\tilde{f}(s_t, a_t) + \omega_t) \right] = \gamma^t \bar{V}_c^{\hat{\pi}}(s_t) - \gamma^t c(s_t, a_t) \tag{23}$$

$$\overset{(i)}{\leq} \mathbb{E}_{\hat{\pi}} \left[ \sum_{\tau=t}^{\infty} \gamma^\tau \left( c(s_\tau, a_\tau) + \lambda_{\text{pessimism}} \| \sigma_n(s_\tau, a_\tau) \| \right) \right] - \gamma^t c(s_t, a_t) \tag{24}$$

$$\leq \mathbb{E}_{\hat{\pi}} \left[ \sum_{\tau=t+1}^{\infty} \gamma^\tau \left( c(s_\tau, a_\tau) + \lambda_{\text{pessimism}} \| \sigma_n(s_\tau, a_\tau) \| \right) \right] + \gamma^t \lambda_{\text{pessimism}} \| \sigma_n(s_t, a_t) \|, \tag{25}$$

$$\text{s.t. } s_{\tau+1} = \mu_n(s_\tau, a_\tau) + \omega_\tau, \quad a_\tau = \hat{\pi}(s_\tau), \quad s_{t+1} = \mu_n(s_t, a_t) + \omega_t, \tag{26}$$

where step (i) follows from using the upper bound in Equation (21). $\qquad \square$

**Theorem 1** (Safety guarantee). *Suppose Assumptions 1 to 4 hold and $\mathcal{F}_n$ is well-calibrated $\forall n = 1, \ldots, N$ according to Definition 1. If actions are executed for all timesteps $t$ according to*

$$\pi_n(a_t | s_t, c_{<t}, Q_{c,n}^{\hat{\pi}}) := \begin{cases} \pi_n(\cdot | s_t) & \text{if } \Phi(s_t, a_t, c_{<t}, Q_{c,n}^{\hat{\pi}}) < d \\ \hat{\pi}(s_t) & \text{otherwise,} \end{cases}$$

*where $\Phi$ is the discounted sum of the realized accumulated cost $c_{<t} := \sum_{\tau=0}^{t-1} \gamma^\tau c(s_\tau, a_\tau)$ until $t-1$ and the pessimistic cost value $Q_{c,n}^{\hat{\pi}}$ such that $\Phi(s_t, a_t, c_{<t}, Q_{c,n}^{\hat{\pi}}) := c_{<t} + \gamma^t Q_{c,n}^{\hat{\pi}}(s_t, a_t)$ with factor $\lambda_{pessimism} = \bar{C} \frac{\gamma}{1-\gamma} \frac{(1+\sqrt{d_S}) \beta_N(\delta)}{\sigma}$, where $\bar{C} = \max\{C_{max}, k_{max}\}$. Then, Algorithm 1 satisfies the safety constraint with probability $1 - \delta$ on $f$ for every episode $n = 1, \ldots, N$.*

*Proof.* Theorem 1 is proven using Lemmas 1 and 2. Consider the accumulate cost $\bar{\Phi}(s_t, a_t, c_{<t}, \mathcal{F}_n)$ from Lemma 1

$$\bar{\Phi}(s_t, a_t, c_{<t}, \mathcal{F}_n) = c_{<t} + \gamma^t c(s_t, a_t) + \gamma^{t+1} \max_{\tilde{f} \in \mathcal{F}_n} \mathbb{E}_\omega \left[ \bar{V}_c^{\hat{\pi}}(\tilde{f}(s_t, a_t) + \omega_t) \right] \quad (27)$$

$$\overset{\text{(i)}}{\leq} c_{<t} + \gamma^t (c(s_t, a_t) + \lambda_{\text{pessimism}} \|\sigma_n(s_t, a_t)\|)$$

$$+ \mathbb{E}_{\hat{\pi}} \left[ \sum_{\tau=t+1}^{\infty} \gamma^\tau \left( c(s_\tau, a_\tau) + \lambda_{\text{pessimism}} \|\sigma_n(s_\tau, a_\tau)\| \right) \right] \quad (28)$$

$$\overset{\text{(ii)}}{=} c_{<t} + \gamma^t Q_{c,n}^{\hat{\pi}}(s_t, a_t) \overset{\text{(iii)}}{=:} \Phi(s_t, a_t, c_{<t}, Q_{c,n}^{\hat{\pi}}) < d. \quad (29)$$

Step (i) follows from the upper bound on the cost penalties in Lemma 2, and Step (ii) and (iii) follow from the definitions of $Q_{c,n}^{\hat{\pi}}$ and $\Phi(s_t, a_t, c_{<t}, Q_{c,n}^{\hat{\pi}})$ from Equation (5) and lemma 1. $\quad\square$

## B    REGRET ANALYSIS

**Overview of proof.**    Theorem 2 states that the tractable objective in Equation (9) achieves sublinear cumulative regret while satisfying the safety constraint. We start our proof by decomposing the cumulative regret into per-episode regret terms. Using Lemma 3, we lower-bound the performance of the true system $\mathcal{M}_c$ by the performance on the planning MDP $\widetilde{\mathcal{M}}$. This bound allows us to decompose the per-episode regret into two components in Lemma 4 that can be bounded separately: **(i) Safe rollout vs. optimal rollout.** The first regret term captures the performance gap between the optimal policy $\pi_c^*$ and the same policy executed via Algorithm 1 on $\mathcal{M}_c$. The difference arises from invoking the prior policy $\hat{\pi}$ due to uncertainty in the learned dynamics. We express this gap in Lemma 5 in terms of action differences between $\pi_c^*$ and the potentially used safe prior $\hat{\pi}$. By Theorem 1, these action differences can be related in Lemma 6 via the safety criterion $\Phi$ to the accumulated model uncertainty along potentially unsafe trajectories starting with $\pi_c^*(a_t|s_t)$ at step $t$, followed by $\hat{\pi}$. Since it may be unsafe to execute them, we instead express their cumulative uncertainty in terms of the corresponding safe trajectories starting with $\hat{\pi}(s_t)$ at step $t$ (Lemma 7). This is achieved using the difference of Gaussians (Kakade et al., 2020, Lemma C.2). Consequently, we establish in Lemma 8 an upper bound on the first regret term, which depends on the model uncertainty along the safe rollout of $\pi_c^*$ under Algorithm 1. **(ii) Planning model vs. true dynamics.** The second regret term quantifies the value gap between an optimal policy for the planning MDP $\widetilde{\mathcal{M}}$ (with true dynamics $f$) and the policy $\pi_n$ obtained by optimizing the nominal model $\mu_n$ in episode $n$. This gap can be bounded in terms of the model uncertainty evaluated along the optimal policy $\pi^*$ when it is safely executed using Algorithm 1. By Lemma 12, the tractable objective in Equation (9) is an optimistic upper bound on the best achievable performance across all models $\mathcal{F}_n$. Consequently, Lemma 13 provides a bound on the sum of the two per-episode regret components along the safely rolled-out behavioral policy $\pi_n$ using Algorithm 1. Summing these bounds over all episodes yields Theorem 2, which states that the objective in Equation (9) attains sublinear cumulative regret.

To decompose $R(N)$, we first establish a lower bound on the performance of $\bar{\pi}_n$, obtained by safely rolling out $\pi_n$ on $\mathcal{M}_c$ with Algorithm 1 using $\Phi$.

**Planning at the lower performance bound.**    Given the policy $\pi_n$ on $\widetilde{\mathcal{M}}$, we derive in Lemma 3 the performance bound of the safe policy $\bar{\pi}_n$ on $\mathcal{M}_c$ with dynamics $f$ using the pessimistic value function $\underline{V}_r^{\hat{\pi}}(s_0)$ defined in Equation (8).

**Lemma 3.** *Suppose Assumptions 1 to 3 hold and the model $\mathcal{F}_n$ is well-calibrated according to Definition 1. Then the performance $J_r(\bar{\pi}_n, f)$ of the safe policy $\bar{\pi}_n$ induced by Algorithm 1 on $\mathcal{M}_c$ is lower-bounded by the performance $J_{\tilde{r}}(\pi_n, f)$ of the policy $\pi_n$ on $\widetilde{\mathcal{M}}$ following the true dynamics $f$:*

$$J_r(\bar{\pi}_n, f) \geq J_{\tilde{r}}(\pi_n, f).$$

*Proof.* We consider for all timesteps $k$ two cases:

1. *Never fall back to prior* (i.e. $\pi_n$ is always safe). Then, by definition of $\widetilde{\mathcal{M}}$,

$$J_{\tilde{r}}(\pi_n, f) = \mathbb{E}_{\pi_n, s_0} \left[ \sum_{t=0}^{\infty} \gamma^t r(s_t, a_t) \right] = J_r(\bar{\pi}_n, f). \quad (30)$$

This equality follows by definition from $\widetilde{\mathcal{M}}$ and $\mathcal{M}_c$ being identical in case the agent never falls back to the safe prior $\hat{\pi}$ during the rollout and therefore $\pi_n = \bar{\pi}_n$.

2. *Agent falls back to safe prior* (i.e. at $s_t$ action $\hat{a}_t = \hat{\pi}(s_t)$ is executed instead of potentially unsafe $a_t$). For the performance $J_r(\bar{\pi}_n, f)$ of $\bar{\pi}_n$ on $\mathcal{M}_c$ we get by combining the policy values of $\pi_n$ and $\hat{\pi}$

$$J_r(\bar{\pi}_n, f) = \mathbb{E}_{\pi_n, s_0}\Big[\sum_{t=0}^{k-1}\gamma^t r(s_t, a_t)\Big] + \gamma^k V_r^{\hat{\pi}}(s_k), \tag{31}$$

whereas for the performance of $J_{\tilde{r}}(\pi_n, f)$ of $\pi_n$ on $\widetilde{\mathcal{M}}$, we get

$$J_{\tilde{r}}(\pi_n, f) \overset{(i)}{=} \mathbb{E}_{\pi_n, s_0}\Big[\sum_{t=0}^{k-1}\gamma^t \tilde{r}(s_t, a_t) + \gamma^k \tilde{r}(s_k, \hat{a}_k) + \sum_{t=k+1}^{\infty}\gamma^t \tilde{r}(s_\dagger, \cdot)\Big] \tag{32}$$

$$\overset{(ii)}{=} \mathbb{E}_{\pi_n, s_0}\Big[\sum_{t=0}^{k-1}\gamma^t r(s_t, a_t) + \gamma^k \underline{V}_r^{\hat{\pi}}(s_k)\Big]. \tag{33}$$

Step (i) follows from the modified transitions in Equation (6) to the absorbing state $s_\dagger$ as soon as the agent would fall back to the safe prior $\hat{\pi}$, and (ii) follows from the terminal reward given in Equation (7) for transitioning to $s_\dagger$. Since $\underline{V}_r^{\hat{\pi}}$ is defined pessimistically in Equation (8), it holds that $\underline{V}_r^{\hat{\pi}}(s) \le V_r^{\hat{\pi}}(s)$ and thereby follows

$$J_{\tilde{r}}(\pi_n, f) \le \mathbb{E}_{\pi_n, s_0}\Big[\sum_{t=0}^{k-1}\gamma^t r(s_t, a_t)\Big] + \gamma^k V_r^{\hat{\pi}}(s_k) = J_r(\bar{\pi}_n, f), \tag{34}$$

which completes the proof.

$\square$

**Regret decomposition.** Lemma 3 enables us to decompose $R(N)$ into two separate terms. The first term models the regret of executing $\bar{\pi}_n$ on $\mathcal{M}_c$, caused by falling back to the safe prior $\hat{\pi}$ when safely rolling out $\pi_n$ with Algorithm 1. The second component determines the (constraint-free) regret of $\pi_n$ on $\widetilde{\mathcal{M}}$.

**Lemma 4.** *Suppose Assumptions 1 to 4 hold and the model $\mathcal{F}_n$ is well-calibrated according to Definition 1. Let $\pi_c^*$ be the optimal policy on $\mathcal{M}_c$ with $J_r(\pi_c^*, f) = \max_{\pi:J_c(\pi,f)<d} J_r(\pi, f)$ and applying Algorithm 1 to $\pi_c^*$ yields $\bar{\pi}_{c,n}^*$ with $J_r(\bar{\pi}_{c,n}^*, f)$ for the current learned model $\mathcal{F}_n$ in episode $n$. Further, let $\pi^*$ be the optimal policy on $\widetilde{\mathcal{M}}$ that has $J_{\tilde{r}}(\pi^*, f) = \max_\pi J_{\tilde{r}}(\pi, f)$ for $\Phi$ and $Q_{c,n}^{\hat{\pi}}$ in episode $n$. Given $J_{\tilde{r}}(\pi_n, f)$ denotes the performance of the behavioral policy $\pi_n$ on $\widetilde{\mathcal{M}}$, the cumulative regret $R(N) = \sum_{n=1}^N (J_r(\pi_c^*, f) - J_r(\bar{\pi}_n, f))$ of $\bar{\pi}_n$ on $\mathcal{M}_c$ can be decomposed into:*

$$R(N) \le \sum_{n=1}^N \big( \underbrace{J_r(\pi_c^*, f) - J_r(\bar{\pi}_{c,n}^*, f)}_{\Delta_n^1 : P.E.R.\ of\ \bar{\pi}_c^*\ with\ \Phi} \big) + \sum_{n=1}^N \big( \underbrace{J_{\tilde{r}}(\pi^*, f) - J_{\tilde{r}}(\pi_n, f)}_{\Delta_n^2 : P.E.R.\ of\ \pi_n on\ \widetilde{\mathcal{M}}} \big),$$

*with the per-episode regret (P.E.R.) terms $\Delta_n^1$ and $\Delta_n^2$ related to the safe prior and the regret on $\widetilde{\mathcal{M}}$.*

*Proof.* Starting with the cumulative regret definition in Equation (2) for $\mathcal{M}_c$, we upper bound $R(N)$ using

$$R(N) \overset{(i)}{=} \sum_{n=1}^N (J_r(\pi_c^*, f) - J_r(\bar{\pi}_n, f)) \le \sum_{n=1}^N (J_r(\pi_c^*, f) - J_{\tilde{r}}(\pi_n, f)) \tag{35}$$

$$\overset{(ii)}{=} \sum_{n=1}^N (J_r(\pi_c^*, f) - J_r(\bar{\pi}_{c,n}^*, f) + J_r(\bar{\pi}_{c,n}^*, f) - J_{\tilde{r}}(\pi_n, f)), \tag{36}$$

where step (i) follows applying Lemma 3 and (ii) introducing a zero term by adding and subtracting $J_r(\bar{\pi}_{c,n}^*, f)$. Since $\mathcal{M}_c$ and $\widetilde{\mathcal{M}}$ are identical for safe policies (e.g. $\bar{\pi}_{c,n}^*$) by not transitioning to $s_\dagger$ (see Equations (6) and (7)) and $\pi^*$ is the optimal policy on $\widetilde{\mathcal{M}}$ with $\mathcal{F}_n$ in episode $n$

$$J_r(\bar{\pi}_{c,n}^*, f) = J_{\tilde{r}}(\bar{\pi}_{c,n}^*, f) \le J_{\tilde{r}}(\pi^*, f). \tag{37}$$

Thus, we obtain the following upper bound of $R(N)$ by plugging Equation (37) into Equation (36):

$$R(N) \leq \sum_{n=1}^{N} (J_r(\pi_c^*, f) - J_r(\bar{\pi}_{c,n}^*, f)) + \sum_{n=1}^{N} (J_{\tilde{r}}(\pi^*, f) - J_{\tilde{r}}(\pi_n, f)). \tag{38}$$

$\square$

$\Delta_n^1$ **regret bound.** Recall from Lemma 4 that the per episode regret of executing $\pi_c^*$ with $\Phi$ for $\mathcal{F}_n$ in episode $n$ on the planning MDP $\mathcal{M}_c$ is $\Delta_n^1 = J_r(\pi_c^*) - J_r(\bar{\pi}_{c,n}^*)$. We rewrite $\Delta_n^1$ in terms of the distance between the optimal policy $\pi_c^*$ and the safely rolled-out policy $\bar{\pi}_{c,n}^*$ in Lemma 5.

**Lemma 5.** *Suppose Assumptions 1 to 4 hold and the model $\mathcal{F}_n$ is calibrated according to Definition 1. Then the per-episode regret $\Delta_n^1 = J_r(\pi_c^*, f) - J_r(\bar{\pi}_{c,n}^*, f)$ admits the bound*

$$\Delta_n^1 \leq \mathbb{E}_{\bar{\pi}_{c}^*, s_0} \left[ \left( L_r + \frac{\bar{R} L_f}{\sigma} \frac{\gamma}{1-\gamma} \right) \sum_{t=0}^{\infty} \gamma^t \mathbb{E}_{\substack{a_c^* \sim \pi_c^*(\cdot|s_t) \\ \bar{a}_{c,n}^* \sim \bar{\pi}_{c,n}^*(\cdot|s_t)}} \left[ \|a_c^* - \bar{a}_{c,n}^*\| \right] \right],$$

$$s_{t+1} = f(s_t, a_t) + \omega_t, \quad a_t \sim \bar{\pi}_{c,n}^*(\cdot|s_t)$$

*with the Lipschitz constants $L_r, L_f$ for the reward and dynamics, and $\bar{R} = \max\{R_{max}, k_{max}\}$.*

*Proof.* We bound the episodic regret with the advantage formulation (As et al., 2025b, Lemma A.2)

$$\Delta_n^1 = \mathbb{E}_{\bar{\pi}_{c,n}^*, s_0} \left[ \sum_{t=0}^{\infty} \gamma^t \mathbb{E}_{a_c^* \sim \pi_c^*(\cdot|s_t)} \left[ A_r(\bar{\pi}_{c,n}^*, s_t, a_c^*) \right] \right], \tag{39}$$

with $A_{r,t}(\pi, s_t, a_t) = r(s_t, a_t) + \gamma \mathbb{E}_{\omega}[V_r^\pi(s_{t+1}) - V_r^\pi(s_t)]$ and $s_{t+1} = f(s_t, a_t) + \omega_t$.

$$\Delta_n^1 \overset{(i)}{=} \mathbb{E}_{\bar{\pi}_{c,n}^*, s_0} \left[ \sum_{t=0}^{\infty} \gamma^t \mathbb{E}_{a_c^* \sim \pi_c^*(\cdot|s_t)} \left[ \left( r(s_t, a_c^*) \right. \right. \right. \tag{40}$$

$$\left. \left. \left. + \gamma \left( \mathbb{E}_{s_{t+1}|s_t, \pi_c^*, \omega} \left[ V_r^{\bar{\pi}_{c,n}^*}(s_{t+1}) \right] - V_r^{\bar{\pi}_{c,n}^*}(s_t) \right) \right) \right] \right]$$

$$\overset{(ii)}{=} \mathbb{E}_{\bar{\pi}_{c,n}^*, s_0} \left[ \sum_{t=0}^{\infty} \gamma^t \mathbb{E}_{\substack{a_c^* \sim \pi^*(\cdot|s_t) \\ \bar{a}_{c,n}^* \sim \bar{\pi}_{c,n}^*(\cdot|s_t)}} \left[ (r(s_t, a_c^*) - r(s_t, \bar{a}_{c,n}^*)) \right] \right] \tag{41}$$

$$+ \mathbb{E}_{\bar{\pi}_{c,n}^*, s_0} \left[ \sum_{t=0}^{\infty} \gamma^{t+1} \left( \mathbb{E}_{s_{t+1}|s_t, \pi_c^*, \omega} [V_r^{\bar{\pi}_{c,n}^*}(s_{t+1})] - \mathbb{E}_{s_{t+1}|s_t, \bar{\pi}_{c,n}^*, \omega} [V_r^{\bar{\pi}_{c,n}^*}(s_{t+1})] \right) \right]$$

$$\overset{(iii)}{\leq} \mathbb{E}_{\bar{\pi}_{c,n}^*, s_0} \left[ \sum_{t=0}^{\infty} \gamma^t \mathbb{E}_{\substack{a_c^* \sim \pi^*(\cdot|s_t) \\ \bar{a}_{c,n}^* \sim \bar{\pi}_{c,n}^*(\cdot|s_t)}} \left[ \min\{L_r \|a_c^* - \bar{a}_{c,n}^*\|, R_{max}\} \right] \right. \tag{42}$$

$$+ \sum_{t=0}^{\infty} \gamma \sqrt{\max\{\mathbb{E}_{s_{t+1}|s_t, \pi_c^*, \omega_t}[R(s_{t+1})], \mathbb{E}_{s_{t+1}|s_t, \bar{\pi}_{c,n}^*, \omega_t}[R(s_{t+1})]\}}$$

$$\left. \times \gamma^t \mathbb{E}_{\substack{a_c^* \sim \pi^*(\cdot|s_t) \\ \bar{a}_{c,n}^* \sim \bar{\pi}_{c,n}^*(\cdot|s_t)}} \left[ \min \left\{ \frac{L_f \|a_c^* - \bar{a}_{c,n}^*\|}{\sigma}, 1 \right\} \right] \right],$$

where step (i) follows from the advantage definition and (ii) uses the value function to derive the dependence on the reward difference in timestep $t$. We rewrite the advantage in terms of the expected action difference in step (iii) using the expectation difference under two Gaussians (Kakade et al., 2020, lemma C.2) with $R(s) = \left( V_r^{\bar{\pi}_{c,n}^*}(s) \right)^2$. Step (iii) uses the Lipschitz constants $L_r, L_f$ for the reward and the dynamics (Assumption 2). Further, we bound this by:

$$\Delta_n^1 \overset{(i)}{\leq} \mathbb{E}_{\bar{\pi}_{c,n}^*, s_0} \left[ \sum_{t=0}^{\infty} \gamma^t \mathbb{E}_{\substack{a_c^* \sim \pi^*(\cdot|s_t) \\ \bar{a}_{c,n}^* \sim \bar{\pi}_{c,n}^*(\cdot|s_t)}} \left[ \min\{L_r \|a_c^* - \bar{a}_{c,n}^*\|, R_{max}\} \right] \right] \tag{43}$$

$$+ \bar{R}\frac{\gamma}{1-\gamma}\sum_{t=0}^{\infty}\gamma^t \mathbb{E}_{\substack{a_c^* \sim \pi^*(\cdot|s_t) \\ \bar{a}_{c,n}^* \sim \bar{\pi}_{c,n}^*(\cdot|s_t)}}\left[\min\left\{\frac{L_f\|a_c^* - \bar{a}_{c,n}^*\|}{\sigma}, 1\right\}\right]\right]$$

$$\overset{(ii)}{\leq}\mathbb{E}_{\bar{\pi}_{c,n}^*, s_0}\left[\left(L_r + \frac{\bar{R}L_f}{\sigma}\frac{\gamma}{1-\gamma}\right)\sum_{t=0}^{\infty}\gamma^t \mathbb{E}_{\substack{a_c^* \sim \pi^*(\cdot|s_t) \\ \bar{a}_{c,n}^* \sim \bar{\pi}_{c,n}^*(\cdot|s_t)}}\left[\|a_c^* - \bar{a}_{c,n}^*\|\right]\right]. \qquad (44)$$

Step (i) uses $\sqrt{\max\{\mathbb{E}_{\omega_t, s_{t+1}|s_t, \pi_c^*, \omega}[R(s_{t+1})], \mathbb{E}_{\omega_t, s_{t+1}|s_t, \bar{\pi}_{c,n}^*, \omega}[R(s_{t+1})]\}} \leq \bar{R}\frac{1}{1-\gamma}$ with $\bar{R} = \max\{R_{\max}, k_{\max}\}$ due to $r \geq 0$ and in (ii) we drop the $\min$ operator due to non-negativity. $\qquad \square$

Lemma 5 expresses the regret in terms of the difference between actions taken by an optimal policy $\pi_c^*$ and those taken by $\bar{\pi}_{c,n}^*$ in episode $n$ with model $\mathcal{F}_n$. In the following, we use Lemma 5 to relate $\Delta_n^1$ to model uncertainties "collected" along trajectories induced by these policies. Bounding these uncertainties ensures that the learner improves its model of the dynamics after each episode.

**Lemma 6.** *Suppose Assumptions 1 to 4 hold and the model $\mathcal{F}_n$ is well calibrated according to Definition 1. Given $\Phi$ is defined as in Theorem 1, $\Delta_n^1$ depends on the accumulated and discounted uncertainty $\Sigma_t$ along the trajectories of action $a_t$ and the safe $\hat{\pi}$, and the diameter of the bounded action space $D_{\mathcal{A}}$. Given $\nu_n = (1 + \lambda_{pessimism})\lambda_{pessimism}$ with $\lambda_{pessimism}$ being defined as in Theorem 1,*

$$\Delta_n^1 \leq \mathbb{E}_{\bar{\pi}_{c,n}^*, s_0}\left[\left(L_r + \frac{R_{max}L_f}{\sigma}\frac{\gamma}{1-\gamma}\right)\sum_{t=0}^{\infty}\gamma^t D_{\mathcal{A}}\min\left\{\mathbb{E}_{a_t \sim \pi_c^*(\cdot|s_t), \omega_t}\left[\frac{\nu_n \Sigma_t(a_t, f)}{\delta_c}\right], 1\right\}\right],$$

$$\Sigma_t(a_t, f) = \gamma^t\|\sigma_n(s_t, a_t)\| + \sum_{\tau=t+1}^{\infty}\mathbb{E}[\gamma_c^\tau\|\sigma_n(s_\tau, \hat{\pi}(s_\tau))\|], \quad s_{\tau+1} = f(s_\tau, \hat{\pi}(s_\tau)) + \omega_\tau$$

*and $\delta_c = \min_t(\mathbb{E}_{a_t \sim \bar{\pi}_{c,n}^*(\cdot|s_t)}[d - (c_{<t} + \gamma^t c(s_t, a_t) + \gamma^{t+1}\mathbb{E}_{\omega_t}[V_c^{\hat{\pi}}(s_{t+1})])]) \in \mathbb{R}^+$.*

*Proof.* We can upper bound the expected difference between the policies by the action bound $D_{\mathcal{A}}$ times the probability $\mathbb{P}$ of falling back to the safe prior

$$D_{\bar{\pi}_{c,n}^*}(s_t) = \mathbb{E}_{\substack{a_c^* \sim \pi^*(\cdot|s_t) \\ \bar{a}_{c,n}^* \sim \bar{\pi}_{c,n}^*(\cdot|s_t)}}\left[\|a_c^* - \bar{a}_{c,n}^*\|\right] \qquad (45)$$

$$\overset{(i)}{\leq}D_{\mathcal{A}}\mathbb{P}_{a_t \sim \pi_c^*(\cdot|s_t)}\left(c_{<t} + \gamma^t Q_{c,n}^{\hat{\pi}}(s_t, a_t) \geq d\right) \qquad (46)$$

$$\overset{(ii)}{=}D_{\mathcal{A}}\mathbb{P}_{a_t \sim \pi_c^*(\cdot|s_t)}\left(c_{<t} + \gamma^t c(s_t, a_t)\right. \qquad (47)$$

$$\left. + \gamma^{t+1}\left(\mathbb{E}_{\omega_t}[\tilde{V}_c^{\hat{\pi}}(\mu_n(s_t', a_t') + \omega_t)] + \lambda_{pessimism}\Sigma_t(a_t, \mu_n)\right) \geq d\right)$$

with $s_{t+1}' = \mu_n(s_t', a_t') + \omega_t$, $\Sigma_t(a_t, \mu_n) = \|\sigma_n(s_t, a_t)\| + \sum_{\tau=t+1}^{\infty}\mathbb{E}[\gamma^{\tau-t}\|\sigma_n(s_\tau', \hat{\pi}(s_\tau'))\|]$ and $\tilde{V}_c^{\hat{\pi}}(s_0) = \sum_{t=0}^{\infty}\gamma^t c(s_t', a_t')$ following the model dynamics $\mu_n$. Step (i) follows from the switching criterion in Theorem 1 and step (ii) by the definition in Equation (5). Next, we relate the uncertainty along trajectories under the model dynamics $\mu_n$ to those under the true dynamics $f$ by

$$D_{\bar{\pi}_{c,n}^*}(s_t) \overset{(i)}{\leq} D_{\mathcal{A}}\mathbb{P}_{a_t \sim \pi_c^*(\cdot|s_t)}\left(c_{<t} + \gamma^t c(s_t, a_t) + \gamma^{t+1}\left(\mathbb{E}_{\omega_t}[\tilde{V}_c^{\hat{\pi}}(\mu_n(s_t', a_t') + \omega_t)]\right.\right. \qquad (48)$$

$$+ \mathbb{E}_{\omega_t}[V_c^{\hat{\pi}}(f(s_t, a_t) + \omega_t)] - \mathbb{E}_{\omega_t}[V_c^{\hat{\pi}}(f(s_t, a_t) + \omega_t)]$$

$$\left.\left. + \lambda_{pessimism}(\Sigma_t(a_t, \mu_n) + \Sigma_t(a_t, f) - \Sigma_t(a_t, f))\right) \geq d\right)$$

$$\overset{(ii)}{\leq} D_{\mathcal{A}}\mathbb{P}_{a_t \sim \pi_c^*(\cdot|s_t)}\left(c_{<t} + \gamma^t c(s_t, a_t) + \gamma^{t+1}\left(\mathbb{E}_{\omega_t}[V_c^{\hat{\pi}}(f(s_t, a_t) + \omega_t)]\right.\right. \qquad (49)$$

$$\left.\left. + (1 + \lambda_{pessimism})\lambda_{pessimism}\Sigma_t(a_t, f)\right) \geq d\right)$$

$$\overset{(iii)}{\leq} D_{\mathcal{A}}\mathbb{P}_{a_t \sim \pi_c^*(\cdot|s_t)}\left(c_{<t} + \gamma^t c(s_t, a_t) + \gamma^{t+1}\left(\mathbb{E}_{\omega_t}[V_c^{\hat{\pi}}(s_{t+1})] + \nu_n \Sigma_t(a_t, f)\right) \geq d\right)$$

$$(50)$$

We first expand the inequality in (i) with two zero terms. In step (ii), we bound the difference of the value and the uncertainty for the different dynamics using Sukhija et al. (2025, Lemma A.5). In (iii) we define $\nu_n = (1+\lambda_{\text{pessimism}})\lambda_{\text{pessimism}}$ and sample according to the true dynamics $s_{t+1} = f(s_t, a_t) + \omega_t$.

Consider the set of all policies that are safely reachable given $\hat{\pi}$

$$\Pi_{<d}^{\hat{\pi}} := \{\pi \mid \mathbb{E}_\pi[c_{<t} + \gamma^t c(s_t, a_t)] + \gamma^{t+1}\mathbb{E}_{\omega_t}[V_c^{\hat{\pi}}(s_{t+1})] < d, \forall t\}. \tag{51}$$

Since $\pi_c^*$ is strictly feasible we have $\mathbb{E}_{\pi_c^*}[c_{<t} + \gamma^t c(s_t, a_t)] + \gamma^{t+1}\mathbb{E}_{\omega_t}[V_c^{\pi_c^*}(s_{t+1})] < d$; in addition, $V_c^{\hat{\pi}}(s) \le V_c^{\pi_c^*}(s)$ for all initial states (cf. Assumption 4), therefore $\mathbb{E}_{\pi_c^*}[c_{<t} + \gamma^t c(s_t, a_t)] + \gamma^{t+1}\mathbb{E}_{\omega_t}[V_c^{\hat{\pi}}(s_{t+1})] < d$ implying that $\pi_c^* \in \Pi_{<d}^{\hat{\pi}}$. We define the smallest (time-wise) safety gap due to switching from $\pi_c^*$ to $\hat{\pi}$ as

$$\delta_c := \min_t(\mathbb{E}_{a_t \sim \pi_c^*(\cdot|s_t)}[d - (c_{<t} + \gamma^t c(s_t, a_t) + \gamma^{t+1}\mathbb{E}_{\omega_t}[V_c^{\hat{\pi}}(s_{t+1})])]) \in \mathbb{R}_{>0}. \tag{52}$$

We thus rewrite the probability of invoking the safe prior $\hat{\pi}$

$$D_{\bar{\pi}_{c,n}^*}(s_t) = D_{\mathcal{A}}\mathbb{P}_{a_t \sim \pi_c^*(\cdot|s_t)}\left(\nu_n\Sigma_t(a_t, f) \ge d - (c_{<t} + \gamma^t c(s_t, a_t) + \gamma^{t+1}\mathbb{E}_{\omega_t}[V_c^{\hat{\pi}}(s_{t+1})])\right) \tag{53}$$

$$= D_{\mathcal{A}}\mathbb{P}_{a_t \sim \pi_c^*(\cdot|s_t)}\left(\nu_n\Sigma_t(a_t, f) \ge \delta_c\right). \tag{54}$$

Using *Markov's inequality*, we can bound the probability of invoking the prior by

$$D_{\bar{\pi}_{c,n}^*}(s_t) \le D_{\mathcal{A}}\min\left\{\mathbb{E}_{a_t \sim \pi_c^*(\cdot|s_t),\omega_t}\left[\frac{\nu_n\Sigma_t(a_t, f)}{\delta_c}\right], 1\right\} \tag{55}$$

Finally, we insert Equation (55) into Lemma 5 and obtain

$$\Delta_n^1 \le \mathbb{E}_{\bar{\pi}_{c,n}^*,s_0}\left[\left(L_r + \frac{\bar{R}L_f}{\sigma}\frac{\gamma}{1-\gamma}\right)\sum_{t=0}^{\infty}\gamma^t D_{\mathcal{A}}\min\left\{\mathbb{E}_{a_t \sim \pi_c^*(\cdot|s_t),\omega_t}\left[\frac{\nu_n\Sigma_t(a_t, f)}{\delta_c}\right], 1\right\}\right]. \tag{56}$$

$\square$

*Remark* 1. Indeed, at the beginning of learning, $\pi_c^*$ may not lie within the set $\{\pi \mid \mathbb{E}_\pi[c_{<t} + \gamma^t c(s_t, a_t)] + \gamma^{t+1}\mathbb{E}_{\omega_t}[\bar{V}_c^{\hat{\pi}}(s_{t+1})] < d, \forall t\} \subseteq \Pi_{<d}^{\hat{\pi}}$ as the pessimistic cost value $V_c^{\hat{\pi}} \le \bar{V}_c^{\hat{\pi}}$ for all $s \in \mathcal{S}$ with probability $1 - \delta$ by definition (recall Equation (4)). Reducing pessimism such that $\bar{V}_c^{\hat{\pi}}$ converges to $V_c^{\hat{\pi}}$ effectively *expands* this set until it converges to $\Pi_{<d}^{\hat{\pi}}$. Moreover, let us consider the case where the prior policy does *not* satisfy the assumption that $V_c^{\hat{\pi}}(s) \le V_c^{\pi_c^*}(s)$ for all $s \in \mathcal{S}$, i.e., the pessimistic prior policy $\hat{\pi}$ in fact accumulates *more* costs (in expectation) than an optimal policy $\pi_c^*$ when executed on the true dynamics $f$. In this case, $\pi_c^*$ is not guaranteed to lie within $\Pi_{<d}^{\hat{\pi}}$ and SOOPER converges to the optimum within the set of all feasible policies $\Pi_{<d}^{\hat{\pi}}$ that are safely reachable given $\hat{\pi}$.

Lemma 6 establishes an upper bound on the first regret term $\Delta_n^1$, based on the probability of action $a_t \sim \pi_c^*(\cdot|s_t)$ to be unsafe given the model uncertainty. More concretely, even though actions $a_t \sim \pi_c^*(\cdot|s_t)$ in Equation (56) are determined by a *safe* policy $\pi_c^*$, they may be regarded unsafe due to limited knowledge during learning, quantified by epistemic model uncertainty. Since such trajectories would trigger $\hat{\pi}$ when executing Algorithm 1, we must relate the bound in Equation (56) to uncertainty induced along trajectories executed by the prior policy $\hat{\pi}$. Therefore, we relate in Lemma 7 these accumulated uncertainties when executing $\pi_c^*$ freely in $t$, to trajectories induced under Algorithm 1, starting with the safe prior policy at timestep $t$. This relation is visualized in Figure 7.

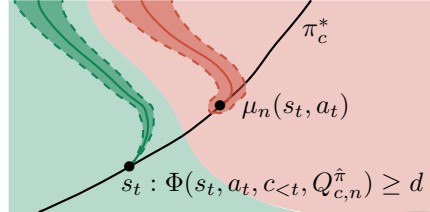

Figure 7: Relating the uncertainty of a safe trajectory (green) to a trajectory that executes $\pi_c^*$ freely at $t$ (i.e. not under Algorithm 1, in red) and therefore may be (possibly wrongly) considered unsafe due to model uncertainties.

**Lemma 7.** *Suppose Assumptions 1 to 4 hold and the model $\mathcal{F}_n$ is well-calibrated according to Definition 1, the accumulated uncertainty along a trajectory induced by executing $\pi_c^*$ freely at $t$, quantified by $\mathbb{E}_{a_t \sim \pi_c^*(\cdot|s_t), \omega_t}[\Sigma_t(a_t, f)]$, can be upper-bounded by the uncertainty along a safe trajectory starting with the safe prior according to Algorithm 1:*

$$\mathbb{E}_{a_t \sim \pi_c^*(\cdot|s_t), \omega_t}[\Sigma_t(a_t, f)] \leq \left(1 + \frac{L_\sigma D_\mathcal{A}}{\sigma}\right) \sqrt{\frac{\gamma^t}{1-\gamma}} \mathbb{E}_{\omega_t}[\Sigma_t(\hat{\pi}(s_t), f)]. \tag{57}$$

*Proof.* Using the definition of $\Sigma_t$ from Lemma 6

$$\Sigma_t(a_t, f) = \gamma^t \|\sigma_n(s_t, a_t)\| + \sum_{\tau=t+1}^{\infty} \mathbb{E}[\gamma_c^\tau \|\sigma_n(s_\tau, \hat{\pi}(s_\tau))\|], \tag{58}$$

we can bound the difference between the accumulated uncertainties of the two trajectories

$$\Delta\Sigma_t = \mathbb{E}_{a_t \sim \pi_c^*(\cdot|s_t)}[\mathbb{E}_{\omega_t}[\Sigma_t(a_t, f)] - \mathbb{E}_{\omega_t}[\Sigma_t(\hat{\pi}(s_t), f)]] \tag{59}$$

$$\overset{(i)}{\leq} \sqrt{\mathbb{E}_{\omega_t}\left[\left(\sum_{\tau=t+1}^{\infty} \mathbb{E}[\gamma^\tau \|\sigma_n(s_\tau, \hat{\pi}(s_\tau))\|]\right)^2\right]} \tag{60}$$

$$\times \min\left\{\frac{\mathbb{E}_{a_t \sim \pi_c^*(\cdot|s_t)}[\gamma^t \|\sigma_n(s_t, a_t) - \sigma_n(s_t, \hat{\pi}(s_t))\|]}{\sigma}, 1\right\}$$

$$\overset{(ii)}{\leq} \frac{L_\sigma D_\mathcal{A}}{\sigma} \sqrt{\mathbb{E}_{\omega_t}[\Sigma_t(\hat{\pi}(s_t), f)]^2}, \tag{61}$$

by using the expectation difference under two Gaussians (Kakade et al., 2020, Lemma C.2) in (i) and bound the uncertainty difference of under the different actions for the first timestep in (ii) using the closed set of actions with diameter $D_\mathcal{A}$ and the Lipschitz constant $L_\sigma$ of the uncertainty. Thus we can further bound this term using $\mathbb{E}_{a_t \sim \pi_c^*(\cdot|s_t), \omega_t}[\Sigma_t] \leq \sqrt{\mathbb{E}_{\omega_t}[\Sigma_t(\hat{\pi}(s_t), f)]^2}$ and obtain

$$\mathbb{E}_{a_t \sim \pi_c^*(\cdot|s_t), \omega_t}[\Sigma_t(a_t, f)] \leq \left(1 + \frac{L_\sigma D_\mathcal{A}}{\sigma}\right) \sqrt{\mathbb{E}_{\omega_t}[\Sigma_t(\hat{\pi}(s_t), f)^2]} \tag{62}$$

$$\overset{(i)}{\leq} \left(1 + \frac{L_\sigma D_\mathcal{A}}{\sigma}\right) \sqrt{\mathbb{E}_{\omega_t}\left[\left(\sum_{\tau=t}^{\infty} \gamma^\tau \|\sigma_n(s_\tau, \hat{\pi}(s_\tau))\|\right)^2\right]} \tag{63}$$

$$\overset{(ii)}{\leq} \left(1 + \frac{L_\sigma D_\mathcal{A}}{\sigma}\right) \sqrt{\mathbb{E}_{\omega_t}\left[\left(\sum_{\tau=t}^{\infty} \gamma^\tau\right)\left(\sum_{\tau=t}^{\infty} \gamma^\tau \|\sigma_n(s_\tau, \hat{\pi}(s_\tau))\|^2\right)\right]} \tag{64}$$

$$\overset{(iii)}{\leq} \left(1 + \frac{L_\sigma D_\mathcal{A}}{\sigma}\right) \sqrt{\frac{\gamma^t}{1-\gamma}\left(\sum_{\tau=t}^{\infty} \mathbb{E}_{\omega_t}[\gamma^\tau \|\sigma_n(s_\tau, \hat{\pi}(s_\tau))\|^2]\right)} \tag{65}$$

$$\overset{(iv)}{\leq} \left(1 + \frac{L_\sigma D_\mathcal{A}}{\sigma}\right) \sqrt{\frac{\gamma^t}{1-\gamma}\left(\sum_{\tau=t}^{\infty} \mathbb{E}_{\omega_t}[\sqrt{\gamma^\tau} \|\sigma_n(s_\tau, \hat{\pi}(s_\tau))\|]\right)^2} \tag{66}$$

$$\leq \left(1 + \frac{L_\sigma D_\mathcal{A}}{\sigma}\right) \sqrt{\frac{\gamma^t}{1-\gamma}} \sum_{\tau=t}^{\infty} \mathbb{E}_{\omega_t}[\sqrt{\gamma^\tau} \|\sigma_n(s_\tau, \hat{\pi}(s_\tau))\|], \tag{67}$$

where we get step (i) by restating the definition of $\Sigma_t(\hat{\pi}(s_t))$ from Lemma 6 and thereafter applying the *Cauchy-Schwarz* inequality in (ii). In step (iii), we factor out the infinite sum of $\gamma^\tau$ and can interchange the square with the summation in (iv) due to the non-negativity of $\|\sigma_n\|$. $\square$

Following from the relation between uncertainty accumulated along trajectories that execute $\pi_c^*$ freely at $t$ and thereafter follow $\hat{\pi}$, and those that pessimistically invoke $\hat{\pi}$ in Lemma 7, we can bound the first regret term when episodes are collected safely using Algorithm 1.

**Lemma 8.** *Suppose Assumptions 1 to 4 hold and the model is calibrated according to Definition 1. Given the Lipschitz constants $L_\sigma$, with bounded actions $D_{\mathcal{A}}$, the first regret term $\Delta_n^1$ is bounded with probability $1 - \delta$ in terms of the safely rolled out policy $\bar{\pi}_{c,n}^*$:*

$$\Delta_n^1 \leq \mathbb{E}_{\bar{\pi}_c^*}\left[\eta_n \sum_{t=0}^{\infty} \mathbb{E}[\sqrt{\gamma^t}\|\sigma_n(s_t, a_t)\|]\right], \quad s_{t+1} = \mu_n(s_t, a_t) + \omega_t, \quad a_t \sim \bar{\pi}_{c,n}^*(\cdot|s_t),$$

*where $\eta_n = \left(L_r + \frac{\bar{R}L_f}{\sigma}\frac{\gamma}{1-\gamma}\right)\frac{1}{(1-\gamma)^{\frac{3}{2}}}D_{\mathcal{A}}\left(1 + \frac{L_\sigma D_{\mathcal{A}}}{\sigma}\right)\frac{(1+\lambda_{pessimism})^2\lambda_{pessimism}}{\delta_c}.$*

*Proof.* Given the derived regret bound in Lemma 6, we bound the sum by the largest summation term during the safe rollout (i.e. executing $\pi_c^*$ under Algorithm 1) using abbreviation $C' = \left(L_r + \frac{\bar{R}L_f}{\sigma}\frac{\gamma}{1-\gamma}\right)$.

$$\Delta_n^1 \leq \mathbb{E}_{\bar{\pi}_{c,n}^*, s_0}\left[C'\sum_{t=0}^{\infty}\gamma^t D_{\mathcal{A}}\min\left\{\mathbb{E}_{a_t\sim\pi_c^*(\cdot|s_t),\omega_t}\left[\frac{\nu_n\Sigma_t(a_t, f)}{\delta_c}\right], 1\right\}\right] \tag{68}$$

$$\leq \mathbb{E}_{\bar{\pi}_{c,n}^*, s_0}\left[C'\frac{1}{1-\gamma}D_{\mathcal{A}}\max_t\left(\min\left\{\mathbb{E}_{a_t\sim\pi_c^*(\cdot|s_t),\omega_t}\left[\frac{\nu_n\Sigma_t(a_t, f)}{\delta_c}\right], 1\right\}\right)\right] \tag{69}$$

$$\leq \mathbb{E}_{\bar{\pi}_{c,n}^*, s_0}\left[C'\frac{1}{1-\gamma}D_{\mathcal{A}}\min\left\{\max_t\left(\mathbb{E}_{a_t\sim\pi_c^*(\cdot|s_t),\omega_t}\left[\frac{\nu_n\Sigma_t(a_t, f)}{\delta_c}\right]\right), 1\right\}\right]. \tag{70}$$

As we invoke the prior for the maximal term we obtain together with Lemma 7

$$\Delta_n^1 \leq \mathbb{E}_{\bar{\pi}_{c,n}^*, s_0}\left[C'\frac{1}{1-\gamma}D_{\mathcal{A}}\min\left\{\max_t\left(\left(1 + \frac{L_\sigma D_{\mathcal{A}}}{\sigma}\right)\sqrt{\frac{\gamma^t}{1-\gamma}}\right.\right.\right. \tag{71}$$

$$\left.\left.\left.\times\frac{\nu_n}{\delta_c}\sum_{\tau=t}^{\infty}\mathbb{E}_{\omega_t}[\sqrt{\gamma^\tau}\|\sigma_n(s_\tau, \hat{\pi}(s_\tau))\|]\right), 1\right\}\right]$$

$$\leq \mathbb{E}_{\bar{\pi}_{c,n}^*, s_0}\left[C'\frac{1}{1-\gamma}D_{\mathcal{A}}\left(1 + \frac{L_\sigma D_{\mathcal{A}}}{\sigma}\right)\sqrt{\frac{1}{1-\gamma}}\frac{\nu_n}{\delta_c}\sum_{t=0}^{\infty}\mathbb{E}_{\omega_t}[\sqrt{\gamma^t}\|\sigma_n(s_t, a_t)\|]\right], \tag{72}$$

$$s_{t+1} = f(s_t, a_t) + \omega_t, \quad a_t \sim \bar{\pi}_{c,n}^*(\cdot|s_t). \tag{73}$$

Since we now have the uncertainty along the entire safe rollout of $\bar{\pi}_{c,n}^*$ following the true dynamics $f$, we can now express the entire rollout in terms of the model dynamics $\mu_n$, using (Sukhija et al., 2025, Lemma A.5):

$$\Delta_n^1 \overset{\text{(i)}}{\leq} \mathbb{E}_{\bar{\pi}_{c,n}^*, s_0}\left[C'\frac{1}{1-\gamma}D_{\mathcal{A}}\left(1 + \frac{L_\sigma D_{\mathcal{A}}}{\sigma}\right)\sqrt{\frac{1}{1-\gamma}}\frac{\nu_n}{\delta_c}\left(\sum_{t=0}^{\infty}\mathbb{E}_{\omega_t}[\sqrt{\gamma^t}\|\sigma_n(s_t, a_t)\|]\right.\right. \tag{74}$$

$$\left.\left.+ \sum_{t=0}^{\infty}\mathbb{E}_{\omega_t}[\sqrt{\gamma^t}\|\sigma_n(s_t', a_t')\|] - \sum_{t=0}^{\infty}\mathbb{E}_{\omega_t}[\sqrt{\gamma^t}\|\sigma_n(s_t', a_t')\|]\right)\right]$$

$$\overset{\text{(ii)}}{\leq} \mathbb{E}_{\bar{\pi}_{c,n}^*, s_0}\left[C'\frac{1}{1-\gamma}D_{\mathcal{A}}\left(1 + \frac{L_\sigma D_{\mathcal{A}}}{\sigma}\right)\sqrt{\frac{1}{1-\gamma}}\frac{\nu_n}{\delta_c}\right. \tag{75}$$

$$\left.\times (1 + \lambda_{\text{pessimism}})\sum_{t=0}^{\infty}\mathbb{E}_{\omega_t}[\sqrt{\gamma^t}\|\sigma_n(s_t', a_t')\|]\right]$$

$$s_{t+1} = f(s_t, a_t) + \omega_t, \quad s_{t+1}' = \mu_n(s_t', a_t') + \omega_t. \tag{76}$$

In step (i), we introduce a zero term by adding and subtracting the accumulated uncertainties along the model dynamics. Next, we bound in (ii) the difference of the uncertainties along the two different dynamics with (Sukhija et al., 2025, Lemma A.5). Thus, the proof is complete by plugging back in $C'$ and $\nu_n$. $\qquad\square$

**$\Delta_n^2$ regret bound.** Let us now consider the second term $\Delta_n^2$ that determines the regret of the policy $\pi_n$ on $\widetilde{\mathcal{M}}$, given $\mathcal{F}_n$. Therefore, we first bound the value difference between the pessimistic value estimate $\underline{V}_r^{\hat{\pi}}$ and the true value of executing $\hat{\pi}$ on $f$. Leveraging this bound, we can upper-bound the value difference of executing $\pi_n$ on $\mathcal{M}_c$ and $\widetilde{\mathcal{M}}$ in Lemma 10. Finally, we bound the second regret term $\Delta_n^2$ in Lemma 11.

**Lemma 9.** *Suppose Assumptions 1 to 3 hold and the model $\mathcal{F}_n$ is well-calibrated according to Definition 1. Given the pessimistic cost value $\bar{V}_c^{\hat{\pi}}$ from Equation (4), we get with probability $1 - \delta$*

$$\gamma^k |V_r^{\hat{\pi}}(s_k) - \underline{V}_r^{\hat{\pi}}(s_k)| \le \lambda_n \sum_{t=k}^{\infty} \mathbb{E}\left[\gamma^t \|\sigma_n(\tilde{s}_t, \hat{\pi}(\tilde{s}_t))\|\right], \quad \lambda_n = \frac{\bar{R}\gamma}{1-\gamma} \frac{2(1 + \sqrt{d_x})\beta_{n-1}(\delta)}{\sigma},$$

*where $\bar{R} = \max\{R_{max}, k_{max}\}$. The trajectory can either be along the worst-case dynamics $\tilde{s}_{t+1} = \mu_n(\tilde{s}_t, \hat{\pi}(\tilde{s}_t)) + \alpha(1 + \sqrt{d_x})\beta_{n-1}(\delta)\sigma_n(\tilde{s}_t, \hat{\pi}(\tilde{s}_t)) + \omega_t$, with $\alpha \in [-1, 1]$, initial state $\tilde{s}_k = s_k$, or the true dynamics $\tilde{s}_{t+1} = f(\tilde{s}_t, \hat{\pi}(\tilde{s}_t)) + \omega_t$, yielding the same bound.*

*Proof.*

$$\gamma^k |V_r^{\hat{\pi}}(s_k) - \underline{V}_r^{\hat{\pi}}(s_k)| = \left|\mathbb{E}\left[\sum_{t=k}^{\infty} \gamma^t (V_r^{\hat{\pi}}(s_{t+1}) - V_r^{\hat{\pi}}(\tilde{s}_{t+1}))\right]\right| \tag{77}$$

$$\le \sum_{t=k}^{\infty} \gamma^t \mathbb{E}[|\mathbb{E}_{\omega_t}(\underline{V}_r^{\hat{\pi}}(s_{t+1}) - \underline{V}_r^{\hat{\pi}}(\tilde{s}_{t+1}))|], \tag{78}$$

where $\tilde{s}_{t+1}$ is in each step the worst next state given $\mathcal{F}_n$, leading to the worst value (see Equation (4)). Thus with $\tilde{s}_{t+1} = \mu_n(\tilde{s}_t, \hat{\pi}(\tilde{s}_t)) + \alpha(1 + \sqrt{d_x})\beta_{n-1}(\delta)\sigma_n(\tilde{s}_t, \hat{\pi}(\tilde{s}_t)) + \omega_t$, where $\alpha \in [-1, 1]$ and $\hat{R}(s) = (V_r^{\hat{\pi}})^2(s)$ it holds

$$\gamma^k |V_r^{\hat{\pi}}(s_k) - \underline{V}_r^{\hat{\pi}}(s_k)| \overset{(i)}{\le} \sum_{t=k}^{\infty} \gamma \mathbb{E}\left[\sqrt{\max\{\mathbb{E}_{\omega_t}[\hat{R}(s_{t+1})], \mathbb{E}_{\omega_t}[\hat{R}(\tilde{s}_{t+1})]\}}\right. \tag{79}$$

$$\left. \times \gamma^t \min\left\{\frac{\|f(s_t, \hat{\pi}(s_t)) - \mu_n(\tilde{s}_t, \hat{\pi}(\tilde{s}_t)) + \alpha(1 + \sqrt{d_x})\beta_{n-1}(\delta)\sigma_n(\tilde{s}_t, \hat{\pi}(\tilde{s}_t))\|}{\sigma}, 1\right\}\right]$$

$$\overset{(ii)}{\le} \frac{\bar{R}\gamma}{1-\gamma} \frac{2(1 + \sqrt{d_x})\beta_{n-1}(\delta)}{\sigma} \sum_{t=k}^{\infty} \mathbb{E}\left[\gamma^t \|\sigma_n(s_t, \hat{\pi}(s_t))\|\right]. \tag{80}$$

We rewrite the value difference dependent on the worst-case dynamics $\mu_n + \alpha(1 + \sqrt{d_x})\beta_{n-1}(\delta)\sigma_n$ and the true dynamics $f$ in step (i) using Kakade et al. (2020, Lemma C.2), which can be expressed in terms of the model uncertainty in (ii) (Sukhija et al., 2023a, Corollary 3). $\square$

**Lemma 10.** *Suppose Assumptions 1 to 3 hold and the model $\mathcal{F}_n$ is calibrated according to Definition 1. We consider the following definitions*

$$J_{\tilde{r}}(\pi_n, f) = \mathbb{E}_{\pi, s_0}\left[\sum_{t=0}^{\infty} \gamma^t \tilde{r}(s_t, a_t)\right],$$

$$a_t \sim \pi_n(\cdot|s_t), \quad s_{t+1} = \begin{cases} f(s_t, a_t) + \omega_t & \Phi(s_t, a_t, c_{<t}, Q_{c,n}^{\hat{\pi}}) \ge d, s_t \ne s_\dagger, \\ s_\dagger & otherwise, \end{cases}$$

$$J_{\tilde{r}}(\pi_n, \mu_n) = \mathbb{E}_{\pi, s_0}\left[\sum_{t=0}^{\infty} \gamma^t \tilde{r}(\tilde{s}_t, \tilde{a}_t)\right],$$

$$\tilde{a}_t \sim \pi_n(\tilde{s}_t), \quad \tilde{s}_{t+1} = \begin{cases} \mu_n(\tilde{s}_t, \tilde{a}_t) + \omega_t & \Phi(\tilde{s}_t, a_t, c_{<t}, Q_{c,n}^{\hat{\pi}}) \ge d, \tilde{s}_t \ne s_\dagger, \\ s_\dagger & otherwise, \end{cases}$$

$$\lambda_n = \frac{\bar{R}\gamma}{1-\gamma} \frac{2(1 + \sqrt{d_x})\beta_{n-1}(\delta)}{\sigma}.$$

*Then we have for all $n \geq 0$, with probability $1 - \delta$*

$$|J_{\tilde{r}}(\pi_n, \mu_n) - J_{\tilde{r}}(\pi_n, f)| \leq \lambda_n \sum_{t=0}^{\infty} \mathbb{E}_{\bar{\pi}_n, s_0} \left[ \gamma^t \|\sigma_{n-1}(\tilde{s}_t, \tilde{a}_t)\| \right]$$

$$|J_{\tilde{r}}(\pi_n, \mu_n) - J_{\tilde{r}}(\pi_n, f)| \leq \lambda_n \sum_{t=0}^{\infty} \mathbb{E}_{\bar{\pi}_n, s_0} \left[ \gamma^t \|\sigma_{n-1}(s_t, a_t)\| \right].$$

*Proof.* We prove the first inequality $|J_{\tilde{r}}(\pi_n, \mu_n) - J_{\tilde{r}}(\pi_n, f)| \leq \lambda_n \sum_{t=0}^{\infty} \mathbb{E}_{\pi_n, s_0} \left[ \gamma^t \|\sigma_{n-1}(\tilde{s}_t, \tilde{a}_t)\| \right]$ and the second one holds for a analogous argument. We extend the argument of Sukhija et al. (2025, Lemma A.5) to account for the termination state $s_\dagger$.

$$|J_{\tilde{r}}(\pi_n, \mu_n) - J_{\tilde{r}}(\pi_n, f)| = \left| \mathbb{E}_{\pi_n, s_0, \mu_n} \left[ \sum_{t=0}^{\infty} \gamma^{t+1} (V_{r,f}^{\pi_n}(\tilde{s}_{t+1}) - V_{r,f}^{\pi_n}(s_{t+1})) \right] \right|, \tag{81}$$

where $V_{r,f}^{\pi}$ is the value following $\pi$ under dynamics $f$, and we define the states $\tilde{s}_{t+1}$ and $s_{t+1}$ as

$$s_{t+1} = \begin{cases} f(\tilde{s}_t, \tilde{a}_t) + \omega_t & \Phi(\tilde{s}_t, \tilde{a}_t, c_{<t}, Q_{c,n}^{\hat{\pi}}) \geq d, \tilde{s}_t \neq s_\dagger, \\ s_\dagger & \text{otherwise,} \end{cases}, \quad \tilde{a}_t \sim \pi_n(\cdot|\tilde{s}_t), \tag{82}$$

$$\tilde{s}_{t+1} = \begin{cases} \mu_n(\tilde{s}_t, \tilde{a}_t) + \omega_t & \Phi(\tilde{s}_t, \tilde{a}_t, c_{<t}, Q_{c,n}^{\hat{\pi}}) \geq d, \tilde{s}_t \neq s_\dagger, \\ s_\dagger & \text{otherwise.} \end{cases}, \quad \tilde{a}_t \sim \pi_n(\cdot|\tilde{s}_t). \tag{83}$$

Let $\frac{(1+\sqrt{d_s})\beta_n(\delta)}{\sigma} R(s) = \left( V_{r,f}^{\pi}(s) \right)^2$ and with $R(s) \leq \lambda_n$, if $\pi_n$ does not invoke the safe prior $\hat{\pi}$, we obtain the following bound following Sukhija et al. (2025, Lemma A.5)

$$|J_{\tilde{r}}(\pi_n, \mu_n) - J_{\tilde{r}}(\pi_n, f)| \leq \sum_{t=0}^{\infty} \gamma \mathbb{E}_{\pi_n, s_0} \left[ \sqrt{\max\{\mathbb{E}_{\omega_t}[R(\tilde{s}_{t+1})], \mathbb{E}_{\omega_t}[R(s_{t+1})]\}} \right. \tag{84}$$

$$\left. \times \gamma^t \min \left\{ \frac{\|s_{t+1} - \tilde{s}_{t+1}\|}{\sigma}, 1 \right\} \right].$$

$$\leq \sum_{t=0}^{\infty} \gamma \mathbb{E}_{\pi_n, s_0} \left[ \sqrt{\max\{\mathbb{E}_{\omega_t}[R(\tilde{s}_{t+1})], \mathbb{E}_{\omega_t}[R(s_{t+1})]\}} \right. \tag{85}$$

$$\left. \times \gamma^t \min \left\{ \frac{\|f(\tilde{s}_t, \tilde{a}_t) - \mu_n(\tilde{s}_t, \tilde{a}_t)\|}{\sigma}, 1 \right\} \right]$$

$$\leq \frac{\bar{R}\gamma}{1-\gamma} \frac{(1+\sqrt{d_x})\beta_{n-1}(\delta)}{\sigma} \sum_{t=0}^{\infty} \mathbb{E}_{\pi_n, s_0} \left[ \gamma^t \|\sigma_n(\tilde{s}_t, \tilde{a}_t)\| \right]. \tag{86}$$

In case the agent invokes $\hat{\pi}$ at some timestep $t = k$, we formulate the regret as

$$|J_{\tilde{r}}(\pi_n, \mu_n) - J_{\tilde{r}}(\pi_n, f)| \overset{(i)}{\leq} \sum_{t=0}^{k-1} \gamma \mathbb{E}_{\pi_n, s_0} \left[ \sqrt{\max\{\mathbb{E}_{\omega_t}[R(\tilde{s}_{t+1})], \mathbb{E}_{\omega_t}[R(s_{t+1})]\}} \right. \tag{87}$$

$$\left. \times \gamma^t \min \left\{ \frac{\|s_{t+1} - \tilde{s}_{t+1}\|}{\sigma}, 1 \right\} \right] + \gamma^k |V_r^{\hat{\pi}}(\tilde{s}_k) - \underline{V}_r^{\hat{\pi}}(\tilde{s}_k)|$$

$$+ \sum_{t=k+1}^{\infty} \sqrt{\mathbb{E}_{\omega_t}[R(s_\dagger)]} \gamma^t \min \left\{ \frac{\|s_\dagger - s_\dagger\|}{\sigma}, 1 \right\}$$

$$\overset{(ii)}{=} \sum_{t=0}^{k-1} \gamma \mathbb{E}_{\pi_n, s_0} \left[ \sqrt{\max\{\mathbb{E}_{\omega_t}[R(\tilde{s}_{t+1})], \mathbb{E}_{\omega_t}[R(s_{t+1})]\}} \right. \tag{88}$$

$$\left. \times \gamma^t \min \left\{ \frac{\|s_{t+1} - \tilde{s}_{t+1}\|}{\sigma}, 1 \right\} \right] + \gamma^k |V_r^{\hat{\pi}}(\tilde{s}_k) - \underline{V}_r^{\hat{\pi}}(\tilde{s}_k)|,$$

$$\overset{(iii)}{\leq} \sum_{t=0}^{k-1} \gamma \mathbb{E}_{\pi_n, s_0} \left[ \sqrt{\max\{\mathbb{E}_{\omega_t}[R(\tilde{s}_{t+1})], \mathbb{E}_{\omega_t}[R(s_{t+1})]\}} \right. \tag{89}$$

$$\times \gamma^t \min \left\{ \frac{\|f(\tilde{s}_t, \pi(\tilde{s}_t)) - \mu_n(\tilde{s}_t, \pi(\tilde{s}_t))\|}{\sigma}, 1 \right\} \Bigg]$$

$$+ \frac{\bar{R}\gamma}{1-\gamma} \frac{2(1+\sqrt{d_x})\beta_{n-1}(\delta)}{\sigma} \sum_{t=k}^{\infty} \mathbb{E}_{\hat{\pi}, \tilde{s}_k} \left[ \gamma^t \|\sigma_n(\tilde{s}_t, \hat{\pi}(\tilde{s}_t))\| \right]$$

$$\overset{(iv)}{\leq} \frac{\bar{R}\gamma}{1-\gamma} \frac{(1+\sqrt{d_x})\beta_{n-1}(\delta)}{\sigma} \sum_{t=0}^{k-1} \mathbb{E}_{\pi_n, s_0} [\gamma^t |\sigma_n(\tilde{s}_t, \tilde{a}_t)\|] \tag{90}$$

$$+ \frac{\bar{R}\gamma}{1-\gamma} \frac{2(1+\sqrt{d_x})\beta_{n-1}(\delta)}{\sigma} \sum_{t=k}^{\infty} \mathbb{E}_{\hat{\pi}, \tilde{s}_k} \left[ \gamma^t \|\sigma_n(\tilde{s}_t, \hat{\pi}(\tilde{s}_t))\| \right]$$

$$\overset{(v)}{\leq} \frac{\bar{R}\gamma}{1-\gamma} \frac{2(1+\sqrt{d_x})\beta_{n-1}(\delta)}{\sigma} \sum_{t=0}^{\infty} \mathbb{E}_{\bar{\pi}_n, s_0} \left[ \gamma^t \|\sigma_n(\tilde{s}_t, \tilde{a}_t)\| \right]. \tag{91}$$

In step (i), we split the infinite sum along the trajectory, when the policy prior $\hat{\pi}$ is invoked. We therefore follow the trajectory along the model dynamics in Equation (83). Further, we eliminate the difference term for the terminal state $s_\dagger$ in (ii). In step (iii), we use Lemma 9 to bound the value difference between the true value $V_r^{\hat{\pi}}$ along $f$ and the pessimistic value $\underline{V}_r^{\hat{\pi}}$. Using Sukhija et al. (2023a, Corollary 3) we bound the difference between the next states in (iv). Finally, in step (v), we formulate the bound over $\bar{\pi}_n$, allowing us to upper bound both terms with the uncertainty using the safely rolled out policy $\bar{\pi}_n$ and abbreviate $\lambda_n = \frac{\bar{R}\gamma}{1-\gamma} \frac{2(1+\sqrt{d_x})\beta_{n-1}(\delta)}{\sigma}$. $\qquad\square$

**Lemma 11.** *Suppose Assumptions 1 to 3 hold and the model is well-calibrated according to definition 1. Given the optimal policy $\pi^*$, then we obtain the following per-episode regret on $\widetilde{\mathcal{M}}$ in episode $n$, $\Delta_n^2 = J_{\tilde{r}}(\pi^*, f) - J_{\tilde{r}}(\pi, f)$ with $\lambda_n = \frac{\bar{R}\gamma}{1-\gamma} \frac{2(1+\sqrt{d_x})\beta_{n-1}(\delta)}{\sigma}$ with probability $(1 - \delta)$*

$$\Delta_n^2 = J_{\tilde{r}}(\pi^*, f) - J_{\tilde{r}}(\pi_n, f) \leq J_{\tilde{r}}(\pi^*, \mu_n) + \lambda_n \sum_{t=0}^{\infty} \mathbb{E}_{\bar{\pi}^*, s_0} \left[ \gamma^t \|\sigma_n(s_t, a_t)\| \right] - J_{\tilde{r}}(\pi_n, f),$$

*where $s_{t+1} = \mu_n(s_t, a_t) + \omega_t$*

*Proof.* We can show this by applying Lemma 10 to an optimal policy $\pi^*$ on $\widetilde{\mathcal{M}}$. $\qquad\square$

**Intrinsic rewards for exploration and expansion.** Next, we show with Lemmas 12 and 13 that the policy $\pi_n$ in Equation (9) achieves both optimism and expansion by upper-bounding the sum of both regret terms $\Delta_n = \Delta_n^1 + \Delta_n^2$.

**Lemma 12.** *Given Assumptions 1 to 4 hold and the model $\mathcal{F}_n$ is well-calibrated according to Definition 1, the policy $\pi_n$ as defined in Equation (9)*

$$\pi_n = \arg\max_{\pi} \mathbb{E}_{\pi} \left[ \sum_{t=0}^{\infty} \left( \gamma^t \tilde{r}(s_t, a_t) + (\gamma^t \lambda_{explore} + \sqrt{\gamma^t} \lambda_{expand}) \|\sigma_n(s_t, a_t)\| \right) \right],$$

$$s_{t+1} = \mu_n(s_t, a_t) + \omega_t, \quad a_t \sim \pi(\cdot|s_t), \quad s_0 \sim \rho_0(\cdot),$$

*with $\lambda_{explore} = 3\lambda_n$ (Lemma 11) and $\lambda_{expand} = 3\eta_n$ (Lemma 8), satisfies with probability $1 - \delta$:*

$$\Delta_n = \Delta_n^1 + \Delta_n^2 \leq \eta_n \sum_{t=0}^{\infty} \mathbb{E}_{\pi_c^*, s_0} [\sqrt{\gamma^t} \|\sigma_n(s_t, a_t)\|] + J_{\tilde{r}}(\pi^*, \mu_n)$$

$$+ \lambda_n \sum_{t=0}^{\infty} \mathbb{E}_{\bar{\pi}^*, s_0} \left[ \gamma^t \|\sigma_n(s_t, a_t)\| \right] - J_{\tilde{r}}(\pi_n, f)$$

$$\leq \sum_{t=0}^{\infty} \mathbb{E}_{\pi_n, s_0} [(\gamma^t \lambda_{explore} + \sqrt{\gamma^t} \lambda_{expand}) |\sigma_n(s_t, a_t)\|] + J_{\tilde{r}}(\pi_n, \mu_n) - J_{\tilde{r}}(\pi_n, f).$$

*Proof.* Recall that by construction $\pi_n$ maximizes

$$J(\pi) = \mathbb{E}_\pi \left[ \sum_{t=0}^\infty \gamma^t \, \tilde{r}(s_t, a_t)) + \sum_{t=0}^\infty \left( \sqrt{\gamma^t} 3\eta_n + \gamma^t 3\lambda_n \right) \|\sigma_n(s_t, a_t)) \| \right]. \tag{92}$$

Hence

$$J(\pi_n) \geq J(\pi) \quad \forall \pi. \tag{93}$$

Apply Equation (93) first to the policy $\bar{\pi}_{c,n}^*$ (i.e. the policy induced by using $\pi_c^*$ and then invoking the safe prior policy $\hat{\pi}$). By the linearity of expectations and non-negativity of all terms,

$$J(\pi_n) \geq J_{\tilde{r}}(\bar{\pi}_{c,n}^*, \mu_n) + \mathbb{E}_{\bar{\pi}_{c,n}^*, s_0} \left[ 3\eta_n \sum_{t=0}^\infty \sqrt{\gamma^t} \|\sigma_n(s_t, a_t)\| + 3\lambda_n \sum_{t=0}^\infty \gamma^t \|\sigma_n(s_t, a_t)\| \right]. \tag{94}$$

Next apply Equation (93) to the optimal policy $\pi^*$ on $\mathcal{F}_n$, yielding

$$J(\pi_n) \geq J_{\tilde{r}}(\pi^*, \mu_n) + \mathbb{E}_{\pi^*, s_0} \left[ 3\eta_n \sum_{t=0}^\infty \sqrt{\gamma^t} \|\sigma_n(s_t, a_t)\| + 3\lambda_n \sum_{t=0}^\infty \gamma^t \|\sigma_n(s_t, a_t)) \| \right]. \tag{95}$$

Adding Equation (94) and Equation (95) results in

$$2J(\pi_n) \geq J_{\tilde{r}}(\bar{\pi}_{c,n}^*, \mu_n) + \mathbb{E}_{\bar{\pi}_{c,n}^*, s_0} \left[ 3\eta_n \sum_{t=0}^\infty \sqrt{\gamma^t} \|\sigma_n(s_t, a_t)\| + 3\lambda_n \sum_{t=0}^\infty \gamma^t \|\sigma_n(s_t, a_t)\| \right] \tag{96}$$

$$+ J_{\tilde{r}}(\pi^*, \mu_n) + \mathbb{E}_{\pi^*, s_0} \left[ 3\eta_n \sum_{t=0}^\infty \sqrt{\gamma^t} \|\sigma_n(s_t, a_t)\| + 3\lambda_n \sum_{t=0}^\infty \gamma^t \|\sigma_n(s_t, a_t)) \| \right].$$

Let us derive a bound for $J_{\tilde{r}}(\bar{\pi}_{c,n}^*, \mu_n)$, that we can relate to the optimal policy $\pi^*$ for $\widetilde{\mathcal{M}}$.

$$J_{\tilde{r}}(\pi^*, \mu_n) \overset{(i)}{\leq} J_{\tilde{r}}(\pi^*, f) + \lambda_n \mathbb{E}_{\pi^*, s_0} \left[ \sum_{t=0}^\infty \gamma^t \|\sigma_n(s_t, a_t)\| \right] \tag{97}$$

$$\overset{(ii)}{\leq} J_r(\bar{\pi}^*, f) + \lambda_n \mathbb{E}_{\bar{\pi}^*, s_0} \left[ \sum_{t=0}^\infty \gamma^t \|\sigma_n(s_t, a_t)\| \right] \tag{98}$$

$$\overset{(iii)}{\leq} J_r(\pi_c^*, f) + \lambda_n \mathbb{E}_{\bar{\pi}^*, s_0} \left[ \sum_{t=0}^\infty \gamma^t \|\sigma_n(s_t, a_t)\| \right] \tag{99}$$

$$\overset{(iv)}{\leq} J_r(\bar{\pi}_{c,n}^*, f) + \lambda_n \mathbb{E}_{\bar{\pi}^*, s_0} \left[ \sum_{t=0}^\infty \gamma^t \|\sigma_n(s_t, a_t)\| \right] \tag{100}$$

$$+ \eta_n \mathbb{E}_{\bar{\pi}_{c,n}^*, s_0} \left[ \sum_{t=0}^\infty \sqrt{\gamma^t} \|\sigma_n(s_t, a_t)\| \right].$$

Step (i) follows from Lemma 10 and (ii) is derived from the definition of $\widetilde{\mathcal{M}}$ in Equations (6) and (7). Next, step (iii) uses the optimality of $\pi_c^*$ and finally step (iv) uses the upper bound of the first regret term $\Delta_n^1$ in Lemma 8. Hence, we can lower bound parts of the term by $J_{\tilde{r}}(\pi^*, \mu_n)$. Therefore, $2J(\pi_n)$ in Equation (96) is lower bounded by

$$2J(\pi_n) \geq 2J_{\tilde{r}}(\pi^*, \mu_n) + 2\eta_n \mathbb{E}_{\bar{\pi}_{c,n}^*, s_0} \left[ \sum_{t=0}^\infty \sqrt{\gamma^t} \|\sigma_n(s_t, a_t)\| \right] \tag{101}$$

$$+ 3\lambda_n \mathbb{E}_{\bar{\pi}_{c,n}^*, s_0} \left[ \sum_{t=0}^\infty \gamma^t \|\sigma_n(s_t, a_t)\| \right] + 3\eta_n \mathbb{E}_{\bar{\pi}^*, s_0} \left[ \sum_{t=0}^\infty \sqrt{\gamma^t} \|\sigma_n(s_t, a_t)\| \right]$$

$$+ \mathbb{E}_{\bar{\pi}^*, s_0} \left[ 2\lambda_n \sum_{t=0}^\infty \gamma^t \|\sigma_n(s_t, a_t)\| \right].$$

Since all terms are non-negative, we can drop the terms $3(\lambda_n + \eta_n)\mathbb{E}_{\bar{\pi}_{c,n}^*,s_0}[\sum_{t=0}^{\infty} \gamma^t \|\sigma_n(s_t, a_t)\|]$ and obtain by dividing both sides by two:

$$2J(\pi_n) \geq 2J_{\tilde{r}}(\pi^*, \mu_n) + 2\lambda_n \mathbb{E}_{\bar{\pi}^*,s_0}\left[\sum_{t=0}^{\infty} \gamma^t \|\sigma_n(s_t, a_t)\|\right] \tag{102}$$

$$+ \mathbb{E}_{\bar{\pi}_{c,n}^*,s_0}\left[2\eta_n \sum_{t=0}^{\infty} \sqrt{\gamma^t}\|\sigma_n(s_t, a_t)\|\right]$$

$$J(\pi_n) \geq J_{\tilde{r}}(\pi^*, \mu_n) + \lambda_n \mathbb{E}_{\bar{\pi}^*,s_0}\left[\sum_{t=0}^{\infty} \gamma^t \|\sigma_n(s_t, a_t)\|\right] \tag{103}$$

$$+ \mathbb{E}_{\bar{\pi}_{c,n}^*,s_0}\left[\eta_n \sum_{t=0}^{\infty} \sqrt{\gamma^t}\|\sigma_n(s_t, a_t)\|\right].$$

By definition of $J(\pi_n)$ and Equation (103), subtracting $J_{\tilde{r}}(\pi_n, f)$ completes the proof. $\square$

In Lemma 12 we show that the objective upper bounds the performance and the per-episode regret on the *model dynamics*. Next, we show in Lemma 13 that we can also bound the regret on the *true dynamics $f$* using $\bar{\pi}_n$.

**Lemma 13.** *Given Assumptions 1 to 4 hold and the model $\mathcal{F}_n$ is well-calibrated according to Definition 1, the per-episode regret in $n$, $\Delta_n = \Delta_n^1 + \Delta_n^2$, is upper-bounded with probability of at least $1 - \delta$ for all $n > 0$ by*

$$\Delta_n \leq (3\lambda_n^2 + 4\lambda_n + 3\eta_n^2 + 3\eta_n)) \sum_{t=0}^{\infty} \mathbb{E}_{\bar{\pi}_n,s_0}[\sqrt{\gamma^t}\|\sigma_n(s_t, a_t)\|],$$

*for $s_t = f(s_t, a_t) + \omega_t$ following the true dynamics with $a_t \sim \bar{\pi}_n(\cdot|s_t, c_{<t}, Q_{c,n}^{\hat{\pi}})$.*

*Proof.* Using the derivation in Lemma 12, we can upper bound the regret in terms of $\pi_n$ by

$$\Delta_n \overset{(i)}{\leq} \sum_{t=0}^{\infty} \mathbb{E}_{\pi_n,s_0}[(\gamma^t \lambda_{\text{explore}} + \sqrt{\gamma^t}\lambda_{\text{expand}})\|\sigma_n(s_t', a_t')\|] + J_{\tilde{r}}(\pi_n, \mu_n) - J_{\tilde{r}}(\pi_n, f) \tag{104}$$

$$\overset{(ii)}{\leq} 3(\eta_n + \lambda_n)\mathbb{E}_{\bar{\pi}_n,s_0}\left[\sum_{t=0}^{\infty} \sqrt{\gamma^t}\|\sigma_n(s_t', a_t')\|\right] + J_{\tilde{r}}(\pi_n, \mu_n) - J_{\tilde{r}}(\pi_n, f), \tag{105}$$

where $s_t' = \mu_n(s_t', a_t') + \omega_t$ follows the true dynamics with $a_t' \sim \pi_n(\cdot|s_t')$ and we insert Lemma 12 in (i). Step (ii) follows from $\gamma \in (0, 1)$ and the uncertainty along the safely executed policy $\bar{\pi}_n$ being larger than for $\pi_n$, which terminates in $s_\dagger$. We next use Lemma 10 to bound $J_{\tilde{r}}(\pi_n, \mu_n) - J_{\tilde{r}}(\pi_n, f)$ and obtain

$$\Delta_n \leq 3(\eta_n + \lambda_n)\mathbb{E}_{\bar{\pi}_n,s_0}\left[\sum_{t=0}^{\infty} \sqrt{\gamma^t}\|\sigma_n(s_t', a_t')\|\right] + \lambda_n \mathbb{E}_{\bar{\pi}_n,s_0}\left[\sum_{t=0}^{\infty} \sqrt{\gamma^t}\|\sigma_n(s_t, a_t)\|\right] \tag{106}$$

$$\overset{(i)}{=} 3\eta_n \mathbb{E}_{\bar{\pi}_n,s_0}\left[\sum_{t=0}^{\infty} \sqrt{\gamma^t}\|\sigma_n(s_t, a_t)\|\right] + 3\eta_n\left(\mathbb{E}_{\bar{\pi}_n,s_0}\left[\sum_{t=0}^{\infty} \sqrt{\gamma^t}\|\sigma_n(s_t', a_t')\|\right]\right. \tag{107}$$

$$\left. - \mathbb{E}_{\bar{\pi}_n,s_0}\left[\sum_{t=0}^{\infty} \sqrt{\gamma^t}\|\sigma_n(s_t, a_t)\|\right]\right) + 3\lambda_n\left(\mathbb{E}_{\bar{\pi}_n,s_0}\left[\sum_{t=0}^{\infty} \sqrt{\gamma^t}\|\sigma_n(s_t', a_t')\|\right]\right.$$

$$\left. - \mathbb{E}_{\bar{\pi}_n,s_0}\left[\sum_{t=0}^{\infty} \sqrt{\gamma^t}\|\sigma_n(s_t, a_t)\|\right]\right) + 4\lambda_n \mathbb{E}_{\bar{\pi}_n,s_0}\left[\sum_{t=0}^{\infty} \sqrt{\gamma^t}\|\sigma_n(s_t, a_t)\|\right],$$

where step (i) introduces zero terms by adding and subtracting accumulated uncertainties. We further bound the per episode regret using non-negativity and Lemma 10

$$\Delta_n \leq (3\lambda_n^2 + 4\lambda_n + 3\eta_n^2 + 3\eta_n)\mathbb{E}_{\bar{\pi}_n,s_0}\left[\sum_{t=0}^{\infty} \sqrt{\gamma^t}\|\sigma_n(s_t, a_t)\|\right]. \tag{108}$$

$\square$

In Lemma 13, we derived an upper bound on the per episode regret $\Delta_n$, based on the policy $\bar{\pi}_n$ executed on the true dynamics $s_{t+1} = f(s_t, a_t) + \omega_t$. Using Chowdhury & Gopalan (2017), we obtain a sublinear cumulative regret for our safe exploration algorithm SOOPER.

**Theorem 2** (Sublinear cumulative regret). *Suppose Assumptions 1 to 4 hold and the model $\mathcal{F}_n$ is well calibrated according to Definition 1. Then, Algorithm 2 guarantees with probability $1 - \delta$*

$$R(N) = \sum_{n=1}^{N} \left( J_r(\pi_c^*, f) - J_r(\bar{\pi}_n, f) \right) \leq \mathcal{O}\left( \Gamma_{N\log(N)}^{7/2} \sqrt{N} \right),$$

$$\text{and } J_c(\bar{\pi}_n, f) \leq d, \ \forall n \in \{1, \ldots, N\},$$

*that is, the cumulative regret grows sub-linearly in $N$, dependent on the maximal information gain $\Gamma_{N\log(N)}$, while satisfying the constraint throughout learning by Theorem 1.*

*Proof.* Given the per episode regret bound in Lemma 13, we sum over all episodes $n = 1, \ldots, N$

$$R(N) = \sum_{n=1}^{N} \Delta_n^1 + \Delta_n^2 \leq \sum_{n=1}^{N} (3\lambda_n^2 + 4\lambda_n + 3\eta_n^2 + 3\eta_n) \sum_{t=0}^{\infty} \mathbb{E}_{\bar{\pi}_n, s_0} \left[ \sqrt{\gamma^t} \|\sigma_n(s_t, a_t)\| \right]. \quad (109)$$

As by definition Chowdhury & Gopalan (2017), for the well-calibrated model $\mathcal{F}_n$, $\beta_n \leq \beta_N$ for all $n = 1, \ldots, N$, we rewrite

$$R(N) \leq (4\lambda_N + 3(\lambda_N^2 + \eta_N^2 + \eta_N)) \sum_{n=1}^{N} \sum_{t=0}^{\infty} \mathbb{E}_{\bar{\pi}_n, s_0} \left[ \sqrt{\gamma^t} \|\sigma_n(s_t, a_t)\| \right] \quad (110)$$

$$\overset{(i)}{\leq} ((4\lambda_N + 3(\lambda_N^2 + \eta_N^2 + \eta_N)) \sqrt{N} \sqrt{\sum_{n=1}^{N} \mathbb{E}_{s_0} \left[ \left( \sum_{t=0}^{\infty} \mathbb{E}_{\bar{\pi}_n} \left[ \sqrt{\gamma^t} \|\sigma_n(s_t, a_t)\| \right] \right)^2 \right]} \quad (111)$$

$$\overset{(ii)}{\leq} (4\lambda_N + 3(\lambda_N^2 + \eta_N^2 + \eta_N)) \sqrt{\frac{N}{1 - \sqrt{\gamma}}} \sqrt{\sum_{n=1}^{N} \mathbb{E}_{s_0} \left[ \sum_{t=0}^{\infty} \mathbb{E}_{\bar{\pi}_n} \left[ \sqrt{\gamma^t} \|\sigma_n(s_t, a_t)\|^2 \right] \right]}. \quad (112)$$

We use *Cauchy-Schwarz* in step (i) for the outer sum over episodes $n = 1, \ldots, N$ and in (ii) for the inner sum over timesteps $t$. Further, we can bound the regret term for episodes $n = 1, \ldots, N$, using the truncated horizon $T_n = -\frac{\log(n)}{\log(\sqrt{\gamma})}$, by decomposing the term into

$$\sum_{n=1}^{N} \mathbb{E}_{s_0} \left[ \sum_{t=0}^{\infty} \mathbb{E}_{\bar{\pi}_n} \left[ \sqrt{\gamma^t} \|\sigma_n(s_t, a_t)\|^2 \right] \right] \quad (113)$$

$$= \sum_{n=1}^{N} \left( \sum_{t=0}^{T_n - 1} \mathbb{E}_{\bar{\pi}_n} \left[ \sqrt{\gamma^t} \|\sigma_n(s_t, a_t)\|^2 \right] + \sum_{t=T_n}^{\infty} \mathbb{E}_{\bar{\pi}_n} \left[ \sqrt{\gamma^t} \|\sigma_n(s_t, a_t)\|^2 \right] \right) \quad (114)$$

$$\leq \sum_{n=1}^{N} \sum_{t=0}^{T_n - 1} \mathbb{E}_{\bar{\pi}_n} \left[ \sqrt{\gamma^t} \|\sigma_n(s_t, a_t)\|^2 \right] + \sum_{t=1}^{N} \gamma^{\frac{T_n}{2}} \frac{k_{max}^2}{1 - \gamma} \quad (115)$$

$$\overset{(i)}{\leq} \sum_{n=1}^{N} \sum_{t=0}^{T_n - 1} \mathbb{E}_{\bar{\pi}_n} \left[ \sqrt{\gamma^t} \|\sigma_n(s_t, a_t)\|^2 \right] + \sum_{n=1}^{N} \frac{1}{n} \frac{k_{max}^2}{1 - \gamma} \quad (116)$$

$$\overset{(ii)}{\leq} \sum_{n=1}^{N} \sum_{t=0}^{T_n - 1} \mathbb{E}_{\bar{\pi}_n} \left[ \sqrt{\gamma^t} \|\sigma_n(s_t, a_t)\|^2 \right] + \frac{k_{max}^2}{1 - \gamma} (\log(N) + 1). \quad (117)$$

Step (i) follows from $\frac{T_n}{2} = -\frac{\log(n)}{\log(\gamma)}$, with $\frac{\log(n)}{\log(\gamma)} = \log_\gamma(n)$ and hence $\gamma^{-\log_\gamma(n)} = n^{-1}$. In (ii), we use the fact that $\frac{1}{n}$ is non-decreasing in $n$ and therefore due to Lehman et al. (2018, Theorem 3.9.1) we bound $\sum_{n=1}^{N} \frac{1}{n} \leq \log(N) + 1$. Further, we bound the first term

$\sum_{n=1}^{N} \sum_{t=0}^{T_n-1} \mathbb{E}_{\tilde{\pi}_n} \left[ \sqrt{\gamma^t} \|\sigma_n(s_t, a_t)\|^2 \right]$ as in Sukhija et al. (2025, theorem 5.7):

$$\leq (4\lambda_N + 3(\lambda_N^2 + \eta_N^2 + \eta_N)) \sqrt{\frac{R_\gamma N \Gamma_{N \log(N)}}{1 - \sqrt{\gamma}} + \frac{R_\gamma \sigma_{max}^2 N(\log(N) + 1)}{1 - \gamma}}, \tag{118}$$

where $R_\gamma = \frac{s_{max}}{\log(1 + s_{max})}$, with $s_{max} = \frac{\sigma^{-2} d_x \sigma_{max}^2}{1 - \sqrt{\gamma}}$. As $T_n$ is non-decreasing, we fix $T_N$ and since $\lambda_N \propto \frac{\beta_N}{1 - \gamma}, \eta_n \propto \frac{\beta_N^3}{1 - \gamma}$, we obtain from the definition of $\beta_N$ from Chowdhury & Gopalan (2017) that

$$R_N \leq \mathcal{O}\left( \Gamma_{N \log(N)}^{7/2} \sqrt{N} \right). \tag{119}$$

$\square$

## C  THEORETICAL EXTENSION TO WORLD MODELS

For our theoretical analysis in Appendices A and B, we assume, for simplicity in presentation, that the reward and cost functions are known. All arguments can be extended to the more general setting of unknown costs and rewards, which are learned as part of a "world model", in addition to the unknown dynamics $f$. In practice, we consider this general setting in our experiments (Section 5). Below we briefly illustrate how one can extend the analysis above to this more general case.

First, one needs to impose an additional RKHS assumption on the cost and reward functions, analogous to Assumption 3 for the unknown dynamics $f$. Accordingly, we jointly learn with the unknown dynamics $f$ a calibrated model for the reward and cost function with nominal predictions $\mu_n^r, \mu_n^c$ and epistemic uncertainty $\sigma_n^r, \sigma_n^c$. Hence, it holds $\forall (s, a) \in \mathcal{S} \times \mathcal{A} : r(s, a) \in \{r' \mid |r' - \mu_n^r| \leq \beta_n \sigma_n^r\}$ and $\forall (s, a) \in \mathcal{S} \times \mathcal{A} : c(s, a) \in \{c' \mid |c' - \mu_n^c| \leq \beta_n \sigma_n^c\}$.

Given only access to the calibrated model predictions, we modify the reward and cost structure on the planning MDP $\widetilde{\mathcal{M}}$ as follows:

$$\tilde{r}(s_t, a_t) := \begin{cases} \underline{V}_r^{\hat{\pi}}(s_t) & \text{if } \Phi(a_t, s_t, c_{<t}, Q_{c,n}^{\hat{\pi}}) \geq d, \\ 0 & \text{if } s_t = s_\dagger, \\ \mu_n^r(s_t, a_t) + \beta_n \sigma_n^r(s_t, a_t) & \text{otherwise}, \end{cases} \tag{120}$$

$$\tilde{c}(s_t, a_t) := \mu_n^c(s_t, a_t) + \beta_n \sigma_n^c(s_t, a_t). \tag{121}$$

By utilizing the above cost and reward, we demonstrate safe exploration and achieve sublinear cumulative regret, even in the presence of unknown cost and reward functions.

**Safety guarantee for unknown costs.** Under the modified cost $\tilde{c}$ in Equation (121), the update of the pessimistic Q-value is given by:

$$Q_{\tilde{c},n}^{\hat{\pi}}(s, a) := \mathbb{E}_{\hat{\pi}} \left[ \sum_{t=0}^{\infty} \gamma^t (\tilde{c}(s_t, a_t) + \lambda_{\text{pessimism}} \|\sigma_n(s_t, a_t)\|) \mid s_0 = s, a_0 = a \right]. \tag{122}$$

For this modified update using $\tilde{c}$, it is possible to show that Lemma 2 holds using uncertain predictions. One can derive it by bounding the difference between the value $\tilde{V}_c^{\hat{\pi}}(s)$ following the model dynamics with the true cost function, and the value $\tilde{V}_{\tilde{c}}^{\hat{\pi}}(s)$ following the model dynamics with the mean predictions

$$|\tilde{V}_c^{\hat{\pi}}(s) - \tilde{V}_{\tilde{c}}^{\hat{\pi}}(s)| = \left| \mathbb{E}_{\hat{\pi}} \left[ \sum_{t=0}^{\infty} \gamma^t (c(s_t, a_t) - \mu_n^c(s_t, a_t)) \right] \mid s_0 = s \right| \tag{123}$$

$$\leq \mathbb{E}_{\hat{\pi}} \left[ \sum_{t=0}^{\infty} \gamma^t |c(s_t, a_t) - \mu_n^c(s_t, a_t)| \mid s_0 = s \right] \tag{124}$$

$$\leq \mathbb{E}_{\hat{\pi}} \left[ \sum_{t=0}^{\infty} \gamma^t \beta_n \sigma_n^c(s_t, a_t) \mid s_0 = s \right], \tag{125}$$

$$s_{t+1} = \mu_n(s_t, a_t) + \omega_t, \quad s_0 = s. \tag{126}$$

Hence, by bringing the actually observable value $\tilde{V}_{\tilde{c}}^{\hat{\pi}}(s)$ to the right side, it follows that

$$\tilde{V}_c^{\hat{\pi}}(s) \leq \tilde{V}_{\tilde{c}}^{\hat{\pi}}(s) + \mathbb{E}_{\hat{\pi}}\left[\sum_{t=0}^{\infty}\gamma^t\beta_n\sigma_n^c(s_t,a_t) \mid s_0 = s\right] \tag{127}$$

$$= \mathbb{E}_{\hat{\pi}}\left[\sum_{t=0}^{\infty}\gamma^t(\mu_n^c(s_t,a_t) + \beta_n\sigma_n^c(s_t,a_t)) \mid s_0 = s\right] \tag{128}$$

$$= \mathbb{E}_{\hat{\pi}}\left[\sum_{t=0}^{\infty}\gamma^t\tilde{c}(s_t,a_t) \mid s_0 = s\right], \tag{129}$$

$$s_{t+1} = \mu_n(s_t,a_t) + \omega_t, \quad a_t = \hat{\pi}(s_t). \tag{130}$$

Using the derived upper bound and plugging it into Equation (20) yields

$$\bar{V}_c^{\hat{\pi}}(s) \leq \mathbb{E}_{\hat{\pi}}\left[\sum_{t=0}^{\infty}\gamma^t\left(\tilde{c}(s_t,a_t) + \lambda_{\text{pessimism}}\|\sigma_n(s_t,a_t)\|\right) \mid s_0 = s\right], \tag{131}$$

$$s_{t+1} = \mu_n(s_t,a_t) + \omega_t, \quad a_t = \hat{\pi}(s_t). \tag{132}$$

Hence, the proof of Theorem 1 holds using $Q_{\tilde{c},n}^{\hat{\pi}}(s,a)$ from Equation (122) in Equation (29). We can also upper-bound this in a more compact form by assuming the same kernel $k$ for the cost and the dynamics prediction, so that we have $\beta_n\sigma_n^c(s_t,a_t) + \lambda_{\text{pessimism}}\|\sigma_n(s_t,a_t)\| \leq (\lambda_{\text{pessimism}} + \beta_n)\|\sigma_n^{f,c}(s_t,a_t)\|$. This follows from the non-negativity of $\sigma$ and with $\sigma_n^{f,c}$ being defined as the concatenation of the prediction uncertainties for the dynamics $\sigma_n$ and the cost $\sigma_n^c$.

**Regret bound for unknown reward and cost function.** Similar to the modified cost update in Equation (122), we use $\tilde{r}$ in Equation (120), and optimize for

$$\pi_n = \arg\max_{\pi} \mathbb{E}_{\tilde{p},\pi,s_0}\left[\sum_{t=0}^{\infty}\left(\gamma^t\tilde{r}(s_t,a_t) + (\gamma^t\lambda_{\text{explore}} + \sqrt{\gamma^t}\lambda_{\text{expand}})\|\sigma_n(s_t,a_t)\|\right)\right], \tag{133}$$

$$s_{t+1} = \mu_n(s_t,a_t) + \omega_t, \quad a_t \sim \pi(\cdot|s_t). \tag{134}$$

For the derivation of the regret bound, the first regret term is affected by enlarging the pessimistic additive cost with $\lambda_{\text{pessimism}} + \beta_n$. This contributes an additive $\beta_n$ term, which does not change the proportionality of $\eta_n \propto \frac{\beta_n^3}{1-\gamma}$. For the second regret term, we obtain an additional term that quantifies the uncertainty in the reward prediction, similar to the derivation in Equation (129). Hence, we obtain the additive term in Lemma 11 while following the mean dynamics and using the model's mean predictions $\mu_n^r$.

$$\Delta_n^2 = J_{\tilde{r}}(\pi^*, f) - J_{\tilde{r}}(\pi_n, f) \tag{135}$$

$$\leq J_{\tilde{r}}(\pi^*, \mu_n) + \lambda_n\sum_{t=0}^{\infty}\mathbb{E}_{\bar{\pi}^*,s_0}\left[\gamma^t\|\sigma_n(s_t,a_t)\|\right] - J_{\tilde{r}}(\pi_n, f) \tag{136}$$

$$\leq \sum_{t=0}^{\infty}\mathbb{E}_{\bar{\pi}^*}\left[\gamma^t(\mu_n^r(s_t,a_t) + \beta_n\sigma_n^r(s_t,a_t)) \mid s_0 = s\right] \tag{137}$$

$$+ \lambda_n\sum_{t=0}^{\infty}\mathbb{E}_{\bar{\pi}^*,s_0}\left[\gamma^t\|\sigma_n(s_t,a_t)\|\right] - J_{\tilde{r}}(\pi_n, f). \tag{138}$$

By definition of the modified objective in Equation (133), we still upper bound in Lemma 12 the regret with the behavioral policy $\pi_n$, so that the following proof steps still hold. Further, we can again assume that the dynamics and the reward are part of the same kernel, resulting in the bound $\beta_n\sigma_n^r(s_t,a_t) + \lambda_n\|\sigma_n(s_t,a_t)\| \leq (\lambda_n + \beta_n)\|\sigma_n^{f,r}(s_t,a_t)\|$, where $\sigma_n^{f,r}$ is defined as the concatenation of the prediction uncertainties of the dynamics $\sigma_n$ and the reward $\sigma_n^r$. Thereby, also for the second regret term, the proportionality remains $\lambda_n \propto \frac{\beta_n}{1-\gamma}$. Consequently, the regret bound remains unchanged $R_N \leq \mathcal{O}\left(\Gamma_{N\log(N)}^{7/2}\sqrt{N}\right)$ for unknown reward and cost functions.

# D    ADDITIONAL EXPERIMENTS

We conduct further empirical analysis of SOOPER, focusing on the following key aspects: **(i)** demonstrating the role of prior policies in accelerating learning; **(ii)** leveraging terminations for safe policy improvement; **(iii)** ablating the role of Algorithm 1; **(iv)** robustness of SOOPER to prior policies with varying pessimism levels; **(v)** the necessity of acting pessimistically for safety, and **(vi)** the influence of optimism on the agent's regret. The goal of these ablations is to provide additional empirical evidence to support the different design choices of SOOPER.

**Safe transfer under task misspecification.**    We demonstrate the effectiveness of policy priors compared to "backup" policies that are commonly considered in previous works. For this, we reduce the size of the goal in the PointGoal1 task from SafetyGym (see Figure 18) from 0.3 to 0.15. This makes learning harder, as additional exploration is required to find the goal. We compare the following methods: **(i)** SOOPER that uses a policy trained on the original 0.3 goal task as a safe prior; **(ii)** SAILR that uses a hand-crafted "backup" policy that brakes the robot upon triggering, following SAILR's "advantage-based filtering". The policy $\pi_n$ in this case is initialized with the same policy prior that SOOPER uses; **(iii)** SOOPER that uses the policy prior only within Algorithm 1, but learns its policy $\pi_n$ completely from scratch; **(iv)** finally, a baseline that uses the hand-crafted braking backup policy, falling back to it as done in in SAILR, while training the policy $\pi_n$ from scratch. Figure 8 demonstrates two results: **(a)** as expected, both baselines that learn from scratch fail to converge within the training budget of 200 episodes. **(b)** On the other hand, when finetuning the prior policy, using a hand-crafted braking "backup" policy indeed satisfies the constraint, however at the cost of slower learning compared to SOOPER. This result illustrates the importance of using policy priors not only to maintain safety, but also to guide and accelerate learning.

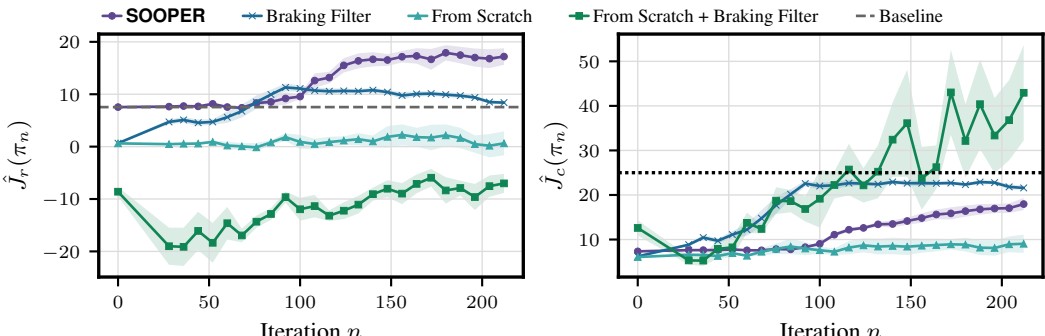

Figure 8: Learning curves of the objective and constraint for PointGoal1 goal size transfer. SOOPER performs significantly better using the prior policy trained on a larger goal size, significantly helping in exploration when learning to reach a smaller goal.

**Learning from terminations.**    We study the effect of learning from terminations on the agent's ability to reduce its dependency on the policy prior. To this end, we ablate the use of terminations in Equations (6) and (7) and compare SOOPER with a variant that does not terminate in $\widetilde{\mathcal{M}}$. In Figure 9, we observe a decrease in the incurred costs for SOOPER with terminations, while SOOPER without terminations still encounters high costs for both CartpoleSwingup and RaceCar. If the optimal unconstrained policy violates the safety budget, the agent trained without terminations fails to achieve the optimal safe performance. This failure occurs because, in the absence of termination signals, the agent receives no feedback about triggering the prior policy during online rollouts.

**Robustness to policy priors.**    Different degrees of sim-to-real gap or lack of data coverage in the offline setting require increasing levels of conservatism. We demonstrate that SOOPER can effectively learn from policy priors with varying levels of conservatism and initial performance. For this, we train three prior policies on PointGoal2 for different sim-to-real gaps with sufficient pessimism levels and show that SOOPER converges to optimal performance while satisfying the constraints using all three policy priors in Figure 10. These results highlight SOOPER's ability to safely explore in highly uncertain environments, given conservative policy priors.

**Robustness to noise.**    Our problem formulation in Equation (1) focuses on *additive Gaussian* noise. While this modeling assumption is key to our theoretical development of SOOPER—in particular

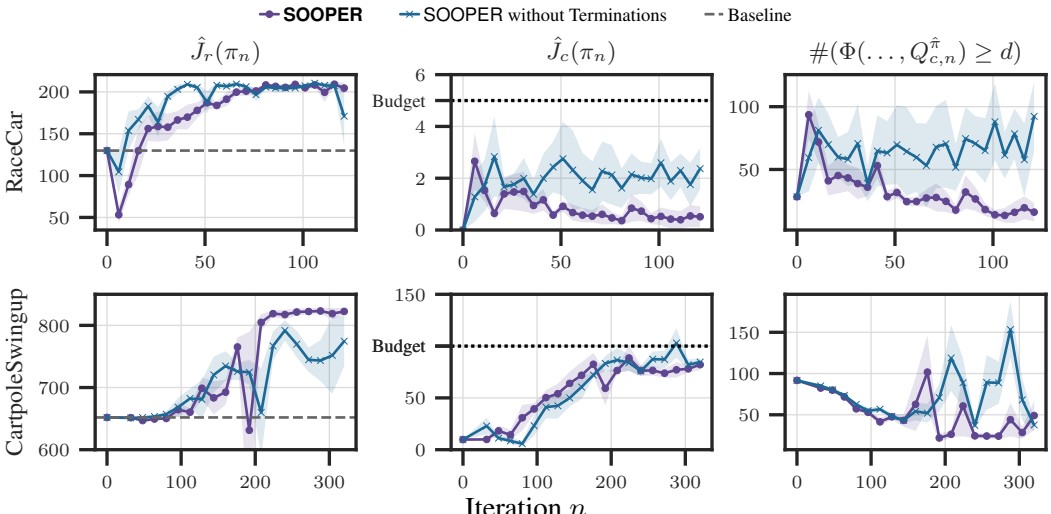

Figure 9: Learning curves of the objective, constraint, and number of times the safe prior is triggered during online rollouts. The number of times SOOPER uses the prior shrinks over training.

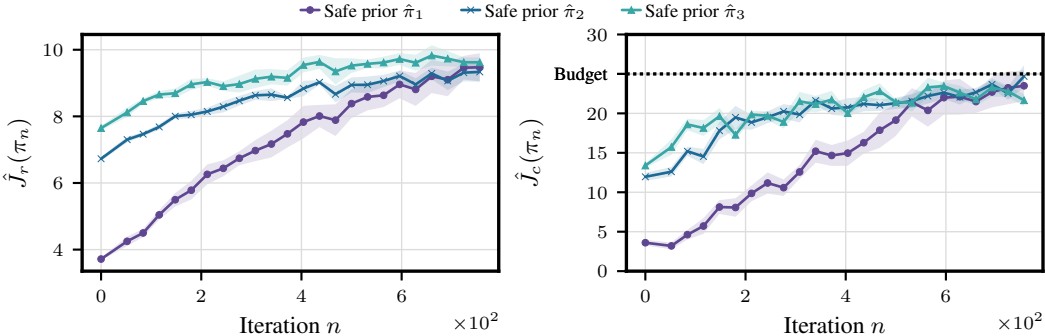

Figure 10: Learning curves of the objective and the constraint, starting from different safe policy priors with various levels of conservatism, differing in initial performances.

for the proof of Lemma 7—we now demonstrate that SOOPER remains safe under substantial disturbances that deviate from the additive Gaussian assumption. In particular, we consider noise of the form $\tilde{a}_t = a_t + \eta_t$, $\eta_t \sim \text{Unif}(-\eta, \eta)$. Notably, this form of noise enters non-linearly to the dynamics $f$. To study this setting, we sweep over the interval of the disturbance distribution with $\eta \in \{0, 0.05, 0.1, 0.15, 0.2, 0.25\}$ for all tasks considered in Figure 3. We use the same prior policies of the experiment in Figure 3; these policies are trained in simulation using $\eta = 0$. Note that for all tasks in our experiments $\mathcal{A} = [-1, 1]^{d_\mathcal{A}}$, hence $\eta = 0.25$ is a non-negligible noise. We present our results in Figure 11 where we report the performance at the end of training and the largest average accumulated cost measured during training for all tasks. As shown, SOOPER maintains safety without suffering performance drops, even under significant noise disturbances. Crucially, these results show that even though $\pi_0$ is obtained under $\eta = 0$, SOOPER can still satisfy the constraint, as long is $\pi_0$ is conservative enough. These results are in line with Theorem 1: safety can be guaranteed for a very general class of noise structures as long as $\pi_0$ satisfies the constraint.

**Pessimism for safe exploration.** Given the previous ablation on conservatism of safe policy priors, we now study the scale of pessimism used *during* learning. As shown for CartpoleSwingup in Figure 12, using increased $\lambda_{\text{pessimism}}$, the agent consistently satisfies the constraints without significant sacrifice in performance. Moreover, SOOPER demonstrates robustness to increased $\lambda_{\text{pessimism}}$, and showcases higher performance for lower chosen pessimism values. Crucially, this behavior is expected as it reflects an inherent tradeoff between pessimism and exploration.

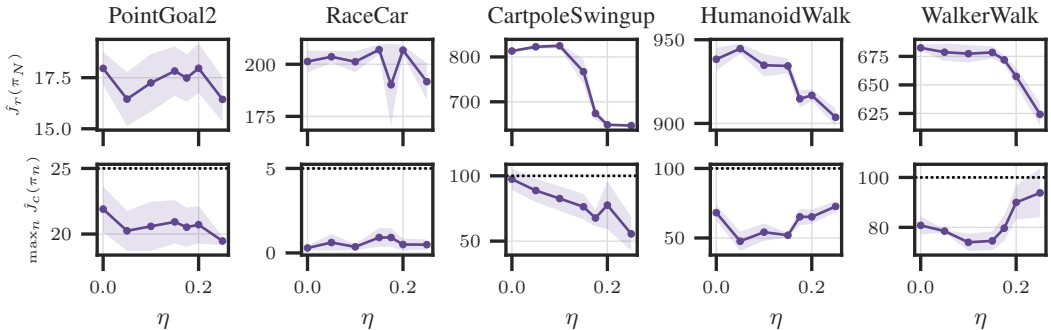

Figure 11: SOOPER achieves good performance and satisfies the constraint under non-Gaussian, non additive noise.

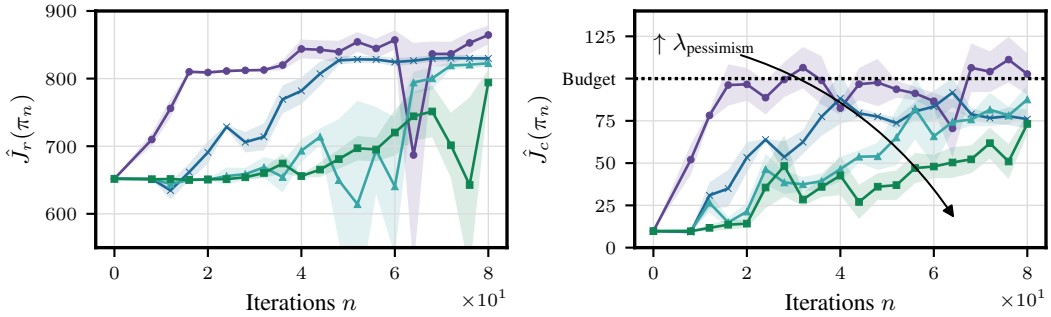

Figure 12: Learning curves of the objective and the constraint for SOOPER with increasing $\lambda_{\text{pessimism}}$. Pessimism improves robustness of constraint satisfaction.

**Optimistic planning for sublinear regret.** We now study the influence of SOOPER's intrinsic reward on its performance, safety during learning and on the cumulative regret. Specifically, we study how varying $\lambda_{\text{expand}}$ together with $\lambda_{\text{explore}}$ improves SOOPER's performance in the CartpoleSwingup task. Intuitively, high values of $\lambda_{\text{expand}} + \lambda_{\text{explore}}$ may lead to excessive exploration, while small values of $\lambda_{\text{expand}} + \lambda_{\text{explore}}$ may not explore enough. As visible in Figure 13, empirically, the cumulative regret $\frac{1}{N} R(N)$ exhibits a single optimum, since for small optimism scales the agent under-explores and fails to expand the safe region, and for large optimism scale the agent over-explores, which reduces short-term greediness and lowers cumulative reward over the finite horizon. As $N$ increases, short-term suboptimality due to over-exploration decreases and the cumulative reward becomes less sensitive to the optimism scale, consistent with the sublinear-regret guarantee for sub-linear growth in $N$.

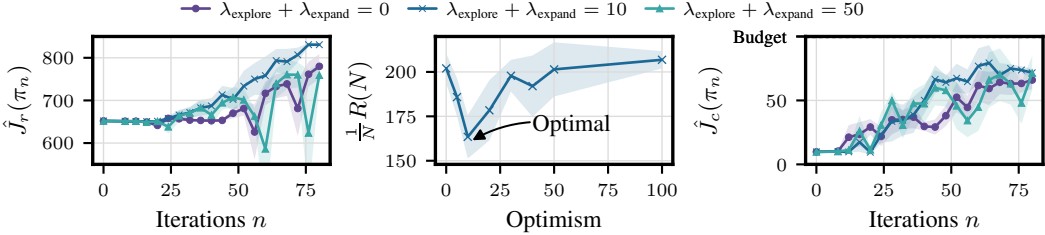

Figure 13: Learning curves of the objective and the constraint, as well as cumulative regret, for different optimism scales. A good value for the optimism value can be identified, where both under- and over-exploration are avoided and regret is minimized.

**Safe offline-to-online on real hardware.** We repeat our hardware experiment in Section 5 on the race car however now use SOOPER using an offline-trained policy. We collect 25K real-world transitions, corresponding to roughly ten minutes of real-time data. To collect this data, we use the

simulation-trained policy used in our sim-to-real experiment as baseline. This dataset is then used to train a pessimistic policy using MOPO (Yu et al., 2020) with a primal-dual solver for the constraint. We introduce pessimism by penalizing the reward and cost with model uncertainty (see Yu et al., 2020). We repeat our hardware experiment with five random seeds, reporting the mean and standard error of the accumulated rewards and costs of each iteration $n$ in Figure 14. As shown, SOOPER leverages the offline-trained policy prior to learn near-optimal policy while satisfying the constraint throughout learning.

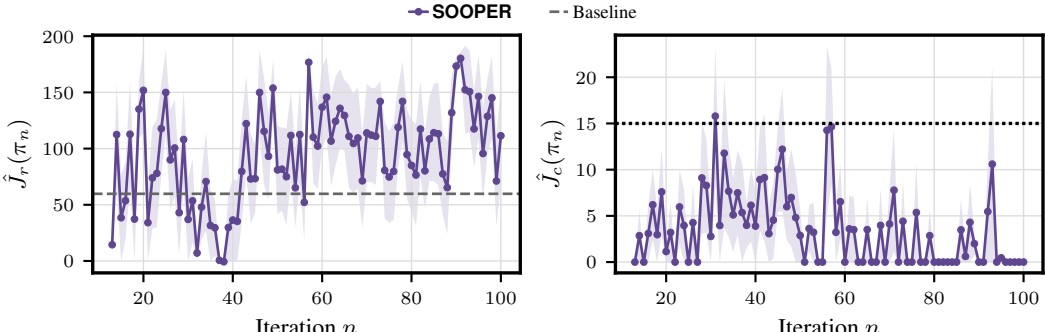

Figure 14: Performance and safety on the real race car given a prior policy that is trained via a fixed offline data. SOOPER improves the performance while satisfying the constraint during learning.

**Learning directly from costs.** While we demonstrate that SOOPER can learn safety from an unconstrained MDP with termination, our method can be extended to learning directly from costs via model-based rollouts with any off-the-shelf CMDP solver. While this variant enjoys the safety guarantee of Theorem 1 using Algorithm 1, it is not guaranteed to learn an optimal policy. We instantiate this variant with CRPO and refer to this baseline as SafeCRPO. The evaluations across various tasks show that SOOPER's Algorithm 1 can be paired with alternative CMDP solvers and ensures safe learning. However, SafeCRPO consistently learns slower and achieves lower performance than SOOPER. This demonstrates the advantage of SOOPER using terminations paired with Algorithm 1 to learn safely and guarantee convergence.

**Runtime.** Even though SOOPER builds on MBPO (Janner et al., 2019b) and therefore enjoys a relatively simple implementation, its use of Algorithm 1 and post-hoc relabeling of transitions with terminations deviate from the standard "RL toolbox". Here, we study the effect of these components on the runtime of SOOPER compared to other model-based baselines. Specifically, we record training throughput, namely, the total number of environment steps divided by the overall wall-clock time for the runs of the experiments in Figure 3. In Figure 16 we aggregate the number of steps per second across all tasks (using five seeds) and report the mean and standard deviation. As shown, even though SOOPER has additional algorithmic components, it does not suffer from significant runtime overhead.

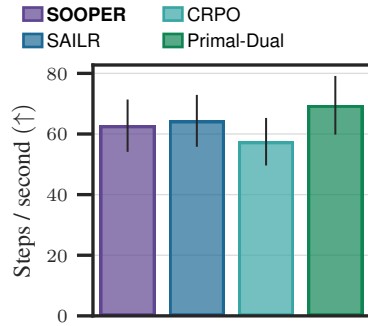

Figure 16: The number of environment steps per second of SOOPER compared to other baselines.

**Additional learning curves.** Figure 17 complements Figure 3 in Section 5 and shows the learning curves on all sim-to-sim tasks. SOOPER consistently achieves near-optimal performance without violating the constraint during learning. In addition to the baselines presented in Section 5, we include another CMDP solver that is based on Log Barriers (Usmanova et al., 2024). This solver is theoretically guaranteed to maintain feasibility during learning by penalizing iterates that approach the boundary of the feasible set. For all of our sim-to-sim experiments, we train the safe policy priors following SPiDR (As et al., 2025a). SPiDR leverages domain randomization to introduce the conservatism required for constraint satisfaction when deploying RL policies under distribution shifts. In particular, to obtain pessimistic policies, SPiDR penalizes the cost according to the disagreement in next-state predictions of the randomized dynamics.

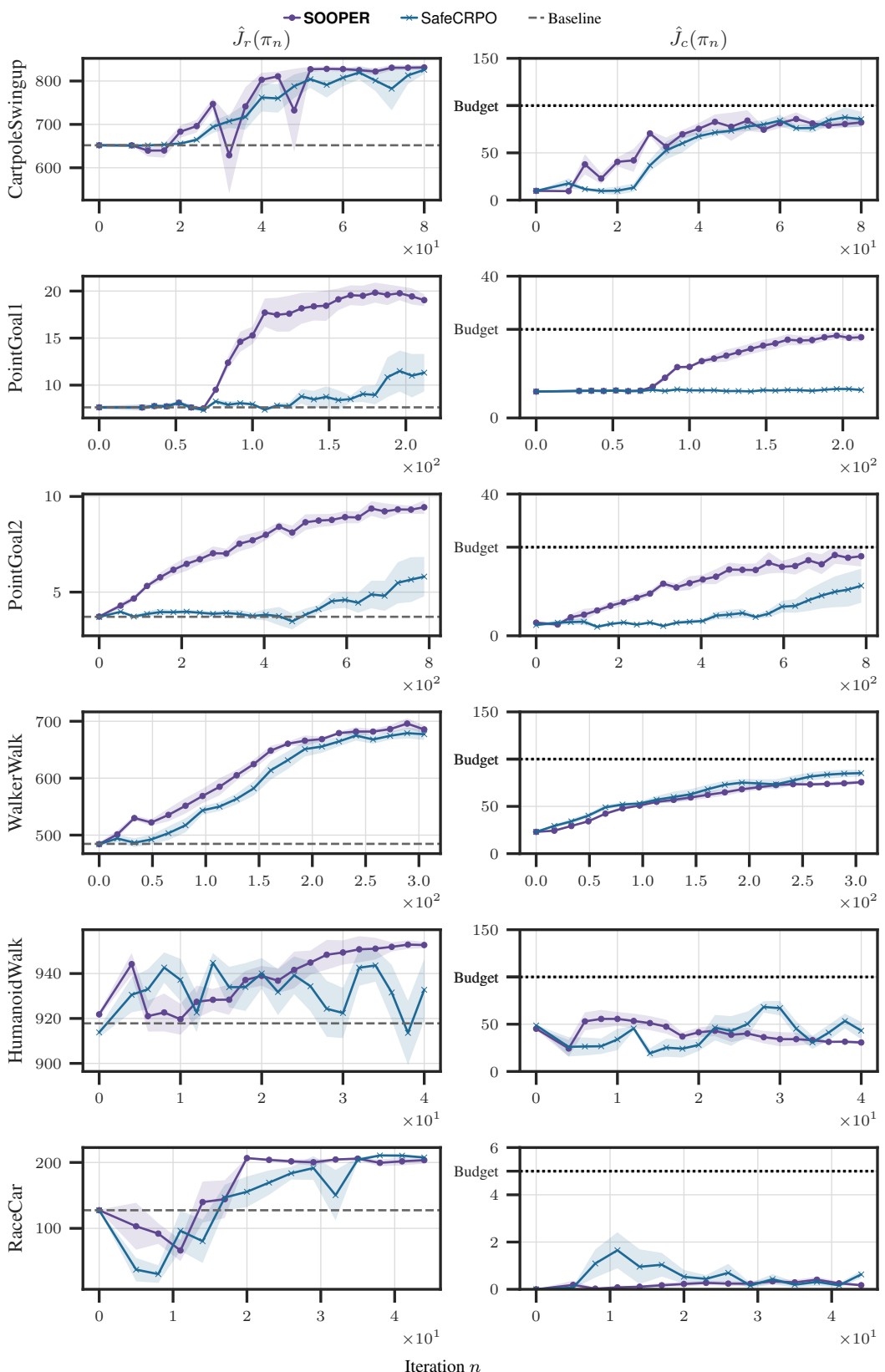

Figure 15: Learning curves of the objective and the constraint for SOOPER and SafeCRPO across all environments. SOOPER consistently outperforms and shows asymptotic convergence.

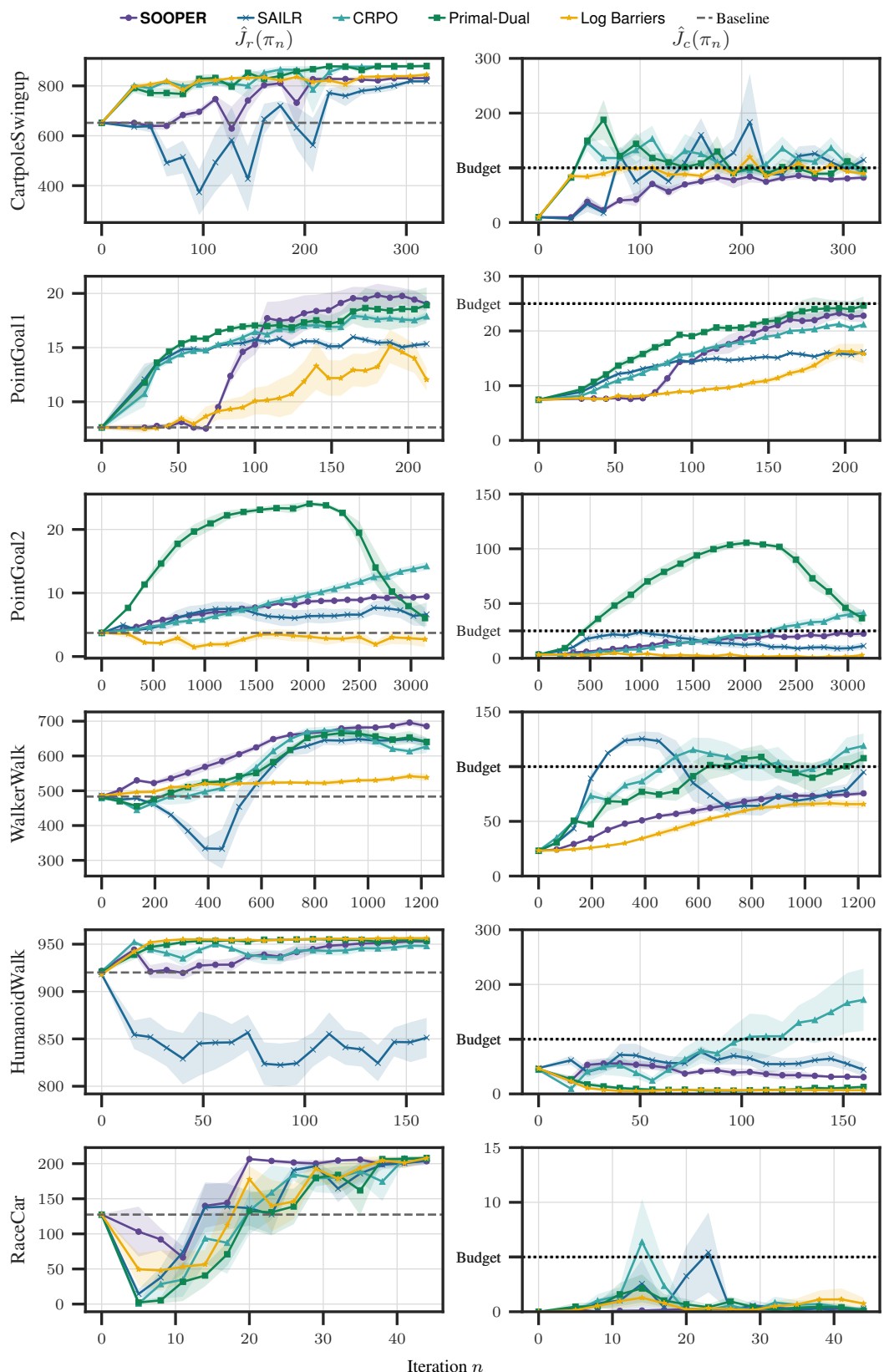

Figure 17: Learning curves of the objective and the constraint for SOOPER and comparison baselines. SOOPER maintains safety throughout training in all tasks while outperforming the baselines in terms of performance.

## E SafetyGym

**PointGoal environments.** In these tasks, the agent must navigate to a target location while avoiding hazards which include free-moving vases and designated hazard zones. The environment is depicted in Figure 18. The initial positions of the agent, goal, vases, and hazards are randomized at the beginning of each episode. The reward function is defined as the change in Euclidean distance to the goal between successive steps

$$r_t(s_t, a_t) \coloneqq d_{t-1} - d_t + \mathbf{1}[d_t \leq \epsilon],$$

where $d_t = \|\mathbf{x}_t - \mathbf{x}_{\text{goal}}\|_2$ is the Euclidean distance from the robot to the goal. The term $\mathbf{1}[d_t \leq \epsilon]$ is an indicator that gives a reward bonus when the agent reaches the goal, i.e. when it is within $\epsilon = 0.3$ of the center of the goal. The goal position is resampled to another free position in the environment once reached. PointGoal1 and PointGoal2 differ in the amount of obstacles located in the environment. A cost of 1 is incurred when the agent collides with a vase $v$, when one of the vases crosses a linear velocity threshold (after collision) or when the agent is inside a hazard zone $h$:

$$c_t(s_t, a_t) \coloneqq \mathbf{1}[\exists v \in V : \text{collides}(\mathbf{x}_t, \mathbf{x}_v)] + \mathbf{1}[\exists v \in V : \mathbf{x}_v' \geq \gamma] + \mathbf{1}[\exists h \in H : d_t \leq \rho],$$

where $\gamma = 5e^{-2}$ and $\rho = 0.2$. Please see the implementation of Ray et al. (2019) for more specific details.

**Simulation gap.** In our sim-to-sim experiments (in Figure 3), before training, we uniformly sample the gear parameters of the actuators that control the linear and angular velocity of the robot. The precise ranges are given in Table 1. Note that the $z$-joint is a hinge joint that allows the agent to rotate around the $z$-axis and the $x$-joint is a slide joint that allows translation in the $xy$-plane.

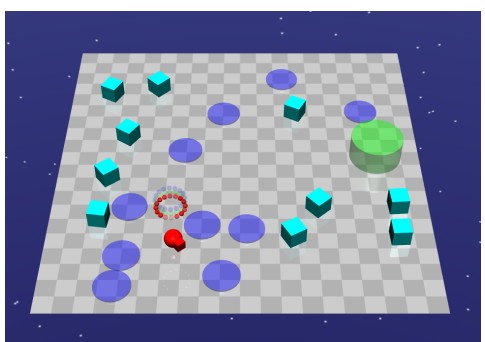

Table 1: Domain randomization parameters and ranges used during training.

| Parameter | Value |
|-----------|-------------|
| Gear (x)  | [−0.2, 0.2] |
| Gear (z)  | [−0.1, 0.1] |

Figure 18: Visualization of a random initialization of the PointGoal2 environment. The red pointmass is the agent, the green transparent cylinder is the goal, the cyan boxes are vases and the blue circles are hazard zones.

## F RWRL Benchmark

We use the RWRL benchmark suite (Dulac-Arnold et al., 2021), which adds safety constraints and distribution shifts to the tasks from the DeepMind Control benchmark suite (Tassa et al., 2018).

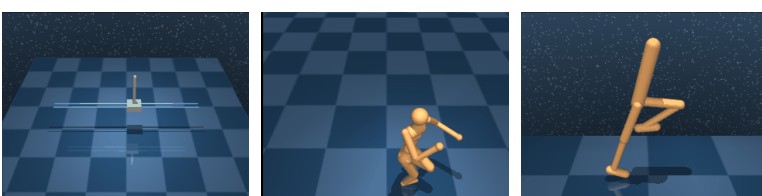

Figure 19: RWRL tasks.

**Constraints.** We use the joint position limits constraint for HumanoidWalk, joint velocity limits for WalkerWalk, and slider position limits for CartpoleSwingup. These are the standard constraints proposed by Dulac-Arnold et al. (2021).

**Simulation gap.** In our sim-to-sim experiments from Figure 3, we follow a similar experimental setup as Queeney & Benosman (2023), introducing variability in the physical properties of the system during training to simulate distribution shifts. In Table 2 we provide the specific parameters we perturb in each task.

Table 2: Additive domain randomization parameters and ranges used during training of the policy prior across tasks from RWLR.

| Parameter | Value |
|---|---|
| **CartpoleSwingup** | |
| Pole Length | [-0.1, 0.1] |
| Gear | [-1.0, 1.0] |
| Knee Gear | [-40., 40.0] |
| **HumanoidWalk** | |
| Friction | [-0.05, 0.05] |
| Hip Gear (x) | [-20.0, 20.0] |
| Hip Gear (y) | [-20.0, 20.0] |
| Hip Gear (z) | [-60.0, 60.0] |
| **WalkerWalk** | |
| Torso Length | [-0.1, 0.1] |
| Gear | [-5.0, 5.0] |

## G    RACECAR ENVIRONMENT

In this environment the agent is asked to navigate to a goal position while avoiding static obstacles. This environment is implemented both in our sim-to-sim experiments, as well as on the real-world race car experiments.

**Reward and cost.** The reward at timestep $t$ is given by

$$r_t(s_t, a_t) := d_{t-1} - d_t + \mathbf{1}[d_t \leq \epsilon] - \lambda_c \|a_t\|_2 - \lambda_l \|a_t - a_{t-1}\|_2^2,$$

where $d_t$ is the Euclidean distance to the goal, and $a_t \in \mathbb{R}^2$ denotes the action applied at time $t$ (consisting of steering and throttle). The term $\mathbf{1}[d_t \leq \epsilon]$ is an indicator function that gives a reward bonus when the agent is within $\epsilon = 0.3$ of the goal. The penalties $\lambda_c$ and $\lambda_l$ weight the control effort (magnitude of the action) and the change in action between consecutive timesteps, respectively. The cost function at time $t$ is defined as

$$c_t(s_t, a_t) := \sum_{i=1}^{3} \mathbf{1}\left[\|x_t - p_i\| < \rho_i\right] E_t^k + \mathbf{1}[x_t \notin \mathcal{V}],$$

where $x_t \in \mathbb{R}^2$ is the car's position, $p_i$ and $\rho_i$ are the position and radius of the $i$-th obstacle and $E_t^k$ is the kinetic energy of the car at time $t$, simulating a plastic collision between the car and obstacles. This choice of cost function allows us to penalize more severely collisions in which the car smashes with high velocity into obstacles, as opposed to softly touching them. The second term penalizes the agent for leaving the valid area $\mathcal{V}$, which corresponds to a bounded rectangular arena.

**Simulation gap.** In the previous sim-to-sim environments, we model the sim-to-sim gap for the experiments in Figure 3 by introducing an auxiliary dynamics parameter (e.g., pendulum length) that is not observed during training. In contrast, in the RaceCar environment, the car dynamics in the training environments are governed by a semi-kinematic bicycle model that does not account for interactions between the tire and the ground. On the other hand, in evaluation, we use the dynamical bicycle model of Kabzan et al. (2020). We refer the reader to Kabzan et al. (2020) for the detailed equations of motion.

