# OpenReview forum: "Safe Exploration via Policy Priors"
_ICLR.cc/2026/Conference — ICLR 2026 Poster_

### Official Review · Reviewer_XArr · 2025-10-20

**Soundness:** 3
**Presentation:** 3
**Contribution:** 3
**Rating:** 8
**Confidence:** 5

**Summary:**

This paper introduces SOOPER, a model-based reinforcement learning algorithm designed for safe exploration for constrained Markov decision processes. The method leverages conservative policy priors to ensure safety while optimistically exploring to improve performance. Conservative policy priors are derived from offline data or simulations. Theoretical guarantees are established for both safety and sublinear cumulative regret. The experiments demonstrate strong empirical performance on several benchmarks and real-world robotic systems.

**Strengths:**

1. This paper is easy to follow and the motivations behind the proposed algorithm are clearly presented. The paper provides a well-motivated discussion of the need for safe exploration in reinforcement learning, emphasizing the trade-off between conservatism and exploration.

1. The relevant literature has been well-covered and I do not find missing references.

1. The proposed algorithm is technically sound.

1. Theoretical results are rigorous, providing both safety guarantees and a sublinear cumulative regret bound. As far as I know, the theoretical results in this paper is novel.

1. The experimental section is comprehensive, including evaluations on RWRL, SafetyGym, and a real hardware setup. The proposed method consistently outperforms or matches strong baselines such as SAILR, CRPO, and Primal-Dual.

1. The implementation based on MBPO demonstrates that the proposed algorithm can scale to high-dimensional continuous control tasks. MBPO is known as a highly scalable algorithm, which is a good technique for making SOOPER practical.

**Weaknesses:**

1. The theoretical analysis relies on several strong assumptions (e.g., Gaussian noise, Lipschitz continuity, bounded RKHS norms, and the existence of a pessimistic policy prior that satisfies safety for all plausible dynamics). While such assumptions are sometimes seen in other safe RL literature, they may limit the practical applicability of the theoretical results.

1. Although the paper claims that SOOPER can be implemented on top of standard model-based methods, the overall system involves several non-trivial components (uncertainty estimation, pessimistic value computation, termination logic). A short discussion on the computational overhead or design trade-offs would be helpful.

**Questions:**

1. We consider the assumption of treating state transitions in the form of equation (1) and Assumptions 1–3 to be very strong. Could you share the authors' views on the strength of this set of assumptions and any attempts to relax them in future research?

1. To what extent do the assumptions made in the paper hold in the experimental setting? While the paper provides a sound theoretical framework, it is somewhat unclear whether all these assumptions are necessary in the experimental setting. It would strengthen the paper to discuss or empirically examine how SOOPER behaves when some of these assumptions do not hold. Since the implementation builds on MBPO to learn the dynamics model, the method should, in practice, already be capable of handling moderately stochastic transitions. Confirming this empirically would provide valuable evidence for the robustness of the proposed approach, by evaluating SOOPER under relaxed or partially violated assumptions. I do not intend to criticize the fact that some assumptions may not hold; rather, if the proposed method still functions effectively as a kind of meta-algorithm even when certain assumptions are violated, that would further enhance the value of this paper.

---

> ### Author Response · Authors · 2025-11-19
>
> Thank you for thoroughly reviewing our paper and for your positive evaluation of our work!
>
> > the overall system involves several non-trivial components (uncertainty estimation, pessimistic value computation, termination logic). A short discussion on the computational overhead or design trade-offs would be helpful.
>
> Thank you for raising this question.
> We agree that SOOPER's implementation involves components that are not yet standard in the deep RL toolbox. We clarify below the design trade-offs and computational overhead of these components:
> - Terminations vs. CMDP solver: terminations allow us to use MBPO "off-the-shelf" instead of implementing a CMDP solver on top of it. Moreover, such CMDP solvers typically involve additional hyper-parameter tuning that SOOPER avoids. We demonstrate this design choice in Figures 9 and 15 in Appendix D where we show that it improves safety and performance.
> - Algorithm 1: this is the main component where SOOPER deviates from MBPO. It plays a critical role in maintaining constraint satisfaction, due to its layered safety approach that follows the "Swiss cheese model" [4]. This importance is demonstrated in Figure 15, showing that even when combined with a standard CMDP solver, it substantially improves safety in practice. Computationally, this component involves an additional forward pass through the action-value cost function and the prior policy. In our hardware experiments, we show that despite this, it runs in real-time at 60Hz. We highlight that our implementation uses standard tools such as Python and JAX.
> Based on your comment, we added an additional experiment that measures the total runtime to Appendix D where we show that SOOPER's runtime aggregated across all simulated tasks is on par with the rest of the baselines.
> Overall, we believe that the contribution of these components to SOOPER's performance and safety in practice outweighs this additional complexity. Please let us know if you find the discussion above valuable; we are happy to include it in our paper.
>
> >   I consider the assumption of treating state transitions in the form of equation (1) and Assumptions 1–3 to be very strong. Could you share the authors' views on the strength of this set of assumptions and any attempts to relax them in future research?
>
> Regarding assumption 1, we would like to emphasize that both conditions (additivity and Gaussian noise) are only required to prove the optimality of SOOPER. The safety guarantee remains satisfied for a broad class of nonlinear dynamics $f(s_t,a_t,\omega_t)$.
> This assumption is important since it enables us to analyze the performance under different dynamics in a continuous nonlinear setting, using the difference of Gaussians lemma in [1]. While this may appear restrictive, we note that many works on RL theory develop such "simulation lemmas" in tabular settings. Extending these ideas to the infinite-horizon, continuous and nonlinear setting---as we do in this paper---is non-trivial; see [2] for further discussion.
> Developing tighter simulation lemmas under weaker assumptions is an important direction and would be of independent interest beyond this paper.
>
> Furthermore, our analysis assumes that the unknown dynamics lie in the RKHS, allowing us to give an explicit non-asymptotic sample complexity bound for nonlinear dynamics. This assumption strictly generalizes linear MDPs [3], a well-established framework that already enjoys rigorous sample complexity guarantees.
>
> > handling moderately stochastic transitions. Confirming this empirically would provide valuable evidence for the robustness of the proposed approach, by evaluating SOOPER under relaxed or partially violated assumptions.
>
> Thank you for this great suggestion. We agree that an additional experiment demonstrating SOOPER's performance when the noise distribution assumption breaks would add value to our paper.
> Based on that, we have added an experiment that injects uniform noise directly into the system dynamics. In particular, for all five tasks considered in Figure 3, we add $\tilde{a}_t = a_t + \eta_t, \; \eta_t \sim \text{Unif}[-\eta, \eta]$ and sweep over the noise magnitude $\eta \in \\{0, 0.05, 0.1, 0.15, 0.2, 0.25\\}$. Note that for all tasks in our experiments $\mathcal{A} = [-1, 1]^{d_a}$, hence $\eta = 0.25$ is non-negligible. Our experiments---added to Appendix D---show that SOOPER maintains safety and achieves good performance on all tasks even under violated assumptions.

---

> ### Author Response · Authors · 2025-11-19
>
> ---
> [1] Kakade, Sham, Akshay Krishnamurthy, Kendall Lowrey, Motoya Ohnishi, and Wen Sun. "Information theoretic regret bounds for online nonlinear control." Advances in Neural Information Processing Systems 33 (2020): 15312-15325.
>
> [2] Ferns, Norm, Prakash Panangaden, and Doina Precup. "Bisimulation metrics for continuous Markov decision processes." SIAM Journal on Computing 40, no. 6 (2011): 1662-1714.
>
> [3] Chowdhury, Sayak Ray, and Aditya Gopalan. "Online learning in kernelized markov decision processes." The 22nd International Conference on Artificial Intelligence and Statistics. PMLR, 2019.
>
> [4] Reason, James. "The contribution of latent human failures to the breakdown of complex systems." Philosophical Transactions of the Royal Society of London. B, Biological Sciences 327, no. 1241 (1990): 475-484.

---

### Official Review · Reviewer_zYbx · 2025-10-25

**Soundness:** 3
**Presentation:** 3
**Contribution:** 3
**Rating:** 6
**Confidence:** 4

**Summary:**

This paper investigates the use of conservative policy prior to enhance safe exploration in reinforcement learning (RL). The proposed method, SOOPER, performs optimistic exploration when safety can be guaranteed with high probability, and switches to the conservative policy prior when exploration becomes potentially unsafe. The authors provide theoretical guarantees for both high-probability constraint satisfaction and optimality. Extensive experiments demonstrate that SOOPER outperforms state-of-the-art baselines across various benchmark tasks.

**Strengths:**

1. While the combination of offline training and online exploration is not entirely new, the paper demonstrates novelty through its theoretical development, particularly in providing safety and optimality guarantees.
2. The empirical evaluations are comprehensive and effectively support the theoretical findings, illustrating the method’s applicability to practical scenarios.
3. The paper is well written and easy to follow, even in the theoretical sections, which are presented with clarity and coherence.

**Weaknesses:**

1. In the Introduction, the authors state that their theoretical results hold under regularity assumptions, and in the “Optimality” subsection of Related Works, they claim to relax some assumptions from prior studies. However, in Section 3 (Problem Setting), Assumption 1 regarding Gaussian noise appears rather restrictive. Moreover, the assumption that the transition dynamics follow \( s_{t+1} = f(s_t, a_t) + \omega_t \) is quite strong; a more general formulation such as \( s_{t+1} = f(s_t, a_t, \omega_t) \) might be preferable. As far as I understand, these assumptions are stronger than those used in [1].
2. The proposed simulated exploration strategy is model-based, which may limit its applicability to environments with large or complex state spaces. It would be valuable for the authors to discuss these limitations and possible extensions to model-free settings.

[1] Akifumi Wachi, and et al, Safe exploration in reinforcement learning: a generalized formulation and algorithms, in NeurIPS 2023.

**Questions:**

1. The proposed method relies heavily on the set of plausible models. In general, as the amount of data increases, the model can be refined and improved. How does such model improvement affect the theoretical guarantees presented in this paper? I did not find a clear discussion on this point.
2. Could the authors clarify why CRPO fails in the safe online learning experiment on real hardware?

---

> ### Author Response · Authors · 2025-11-18
>
> Thank you for thoroughly reviewing our paper and for helping us improve it. We appreciate your acknowledgment of the theoretical novelty and empirical evaluations of SOOPER. Please find our detailed response to the concerns raised below.
>
> > Assumption 1 regarding Gaussian noise appears rather restrictive. Moreover, the assumption that the transition dynamics follow $s_{t+1} = f(s_t, a_t) + \omega_t$ is quite strong.
>
> We would like to point out that both assumptions (additivity and Gaussian noise) are required only for the optimality guarantee (specifically, Equation 59).
> The safety guarantees hold for a broad class of nonlinear dynamics $f(s_t,a_t, \omega_t)$, as you correctly mentioned.
> Our analysis requires additive Gaussian noise in the optimality proof in Lemma 7, where we use the Difference of Gaussians lemma of Kakade et al. [1]. Note that one could use other simulation lemmas like in [2, 3]. While these lemmas require arguably less stringent assumptions on the noise, they are much more technically intricate and would not fundamentally change the theoretical insight we present in this work. We agree that modeling the noise directly through the dynamics is a more general formulation. At the same time, the additive Gaussian noise assumption is widely used in the control theory and robotics literature [4].
> To show the practical applicability of SOOPER beyond additive noise,
> we have added an experiment in this revision that injects uniform noise directly into the nonlinear system dynamics.
> In particular, we add $\tilde{a}_t = a_t + \eta_t$ and sweep over the noise magnitude $\eta$ for the tasks in Figure 3. Our experiments—which we add to Appendix D based on your suggestion—show that SOOPER maintains safety and achieves good performance on all tasks even under non-additive noise.
>
> > The proposed simulated exploration strategy is model-based, which may limit its applicability to environments with large or complex state spaces. It would be valuable for the authors to discuss these limitations and possible extensions to model-free settings.
>
> We acknowledge that learning complex dynamics at scale is still an active research area e.g. [5, 6]. Despite that, in our experiments, we in fact demonstrate one possible approach for scaling SOOPER to challenging observations such as images (see Figure 4). Specifically, we show that learning the dynamics on top of latent feature representations yields good performance and that uncertainty estimates are sufficiently accurate to maintain safety. Please note that a similar architecture for scaling (unsafe) model-based RL to such large-scale problems has been extensively studied in [7].
>
> Regarding the extension to model-free algorithms, in the unconstrained setting, [8, 9, 10] achieve optimality guarantees using RKHS theory directly on the value functions. This requires the value function to lie within an RKHS of an appropriately chosen kernel. We believe this is a promising direction for future work and are happy to discuss that in the appendix, based on your recommendation.
>
> > In general, as the amount of data increases, the model can be refined and improved. How does such model improvement affect the theoretical guarantees presented in this paper?
>
> In Line 4 of Algorithm 2, we update the model $\mathcal{F}$ using all the transitions experienced so far. By updating $\mathcal{F}_n$ with the observed data, the uncertainty $\sigma_n$ shrinks with increasing number of observations. To formally show that, we use the information-gain bounds in [11, 12, 13]. Finally, as described in lines 351-354, the reduction of uncertainty leads SOOPER to depend less on the (suboptimal) prior policy. We have clarified this point in the main text at the end of the proof sketch paragraph, based on the discussion above. Thank you for bringing this to our attention!
>
> > Could the authors clarify why CRPO fails in the safe online learning experiment on real hardware?
>
> Thank you for this question. We believe that the reason CRPO fails to satisfy the constraint on hardware is mainly because it does not leverage the prior safe policy throughout learning. In contrast, Algorithm 1 of SOOPER allows it to robustly satisfy the constraint, even if a policy $\pi_n$ in intermediate episodes fails to do so. This problem becomes more significant under distribution shifts, e.g. from a simulator to a real system. SOOPER enjoys improved robustness because it follows a “Swiss cheese model” [14] for safety: not only learning a policy that satisfies the constraints through terminations, but also online via Algorithm 1. We are happy to elaborate on that in the paper if you believe it would improve it.
>
> We hope our response addresses all of your concerns, and would appreciate it if you would consider increasing your score. We are happy to answer any other open questions. Thanks again for your high-quality review!

---

> > ### Author Response · Authors · 2025-11-18
> > **References**
> >
> > ---
> > [1] Kakade, Sham, Akshay Krishnamurthy, Kendall Lowrey, Motoya Ohnishi, and Wen Sun. "Information theoretic regret bounds for online nonlinear control." Advances in Neural Information Processing Systems 33 (2020): 15312-15325.
> >
> > [2] Ferns, Norm, Prakash Panangaden, and Doina Precup. "Bisimulation metrics for continuous Markov decision processes." SIAM Journal on Computing 40, no. 6 (2011): 1662-1714.
> >
> > [3] Asadi, Kavosh, Dipendra Misra, and Michael Littman. "Lipschitz continuity in model-based reinforcement learning." In International conference on machine learning, pp. 264-273. PMLR, 2018.
> >
> > [4] Todorov, Emanuel, and Weiwei Li. "A generalized iterative LQG method for locally-optimal feedback control of constrained nonlinear stochastic systems." In Proceedings of the 2005, American Control Conference, 2005., pp. 300-306. IEEE, 2005.
> >
> >
> > [5] Ali, A., Bai, J., Bala, M., Balaji, Y., Blakeman, A., Cai, T., Cao, J., Cao, T., Cha, E., Chao, Y.W. and Chattopadhyay, P., 2025. World Simulation with Video Foundation Models for Physical AI. arXiv preprint arXiv:2511.00062.
> >
> > [6] Hafner, Danijar, Wilson Yan, and Timothy Lillicrap. "Training agents inside of scalable world models." arXiv preprint arXiv:2509.24527 (2025).
> >
> > [7] Hansen, Nicklas, Hao Su, and Xiaolong Wang. "Td-mpc2: Scalable, robust world models for continuous control." arXiv preprint arXiv:2310.16828 (2023).
> >
> > [8] Yang, Zhuoran, et al. "On function approximation in reinforcement learning: Optimism in the face of large state spaces." arXiv preprint arXiv:2011.04622 (2020).
> >
> > [9] Chowdhury, Sayak Ray, and Rafael Oliveira. "Value function approximations via kernel embeddings for no-regret reinforcement learning." Asian Conference on Machine Learning. PMLR, 2023.
> >
> > [10] Yeh, Sing-Yuan, et al. "Sample complexity of kernel-based q-learning." International Conference on Artificial Intelligence and Statistics. PMLR, 2023.
> >
> > [11] Sayak Ray Chowdhury and Aditya Gopalan. "On kernelized multi-armed bandits." International Conference on Machine Learning, 2017.
> >
> > [12] Srinivas, Niranjan, Andreas Krause, Sham M. Kakade, and Matthias Seeger. "Gaussian process optimization in the bandit setting: No regret and experimental design." arXiv preprint arXiv:0912.3995 (2009).
> >
> > [13] Curi, Sebastian, Felix Berkenkamp, and Andreas Krause. "Efficient model-based reinforcement learning through optimistic policy search and planning." Advances in Neural Information Processing Systems 33 (2020): 14156-14170.
> >
> > [14] Reason, James. "The contribution of latent human failures to the breakdown of complex systems." Philosophical Transactions of the Royal Society of London. B, Biological Sciences 327, no. 1241 (1990): 475-484.

---

> > ### Comment · Reviewer_zYbx · 2025-11-26
> >
> > Thanks for your response. I think your response has addressed my concerns.

---

> ### Author Response · Authors · 2025-11-26
>
> Many thanks for your thorough review and for re-evaluating your score to match the other reviewers---we truly appreciate it!

---

### Official Review · Reviewer_vK36 · 2025-11-04

**Soundness:** 3
**Presentation:** 3
**Contribution:** 3
**Rating:** 8
**Confidence:** 4

**Summary:**

The paper proposes SOOPER (Safe Online Optimism for Pessimistic Expansion in RL), a model-based algorithm for safe exploration in continuous CMDPs. It leverages pessimistic policy priors, which are safe but suboptimal policies learned offline or in simulation, to ensure constraint satisfaction, while using probabilistic world models for optimistic exploration.

The authors prove that SOOPER satisfies safety constraints at all times (Theorem 1) and achieves sublinear cumulative regret (Theorem 2), unlike prior methods that only guarantee asymptotic safety or simple-regret bounds. The analysis couples online cost-tracking with a "planning MDP with pessimistic termination reward," allowing standard RL methods to be used while maintaining guarantees. Empirical validation on SafetyGym, RWRL, and real robots shows consistent safety and performance gains.

**Strengths:**

+ Strong analytical results by combining always-safe learning with sublinear cumulative regret.
+ The pessimistic-termination MDP approach converts a constrained problem into a standard RL one.
+ Unification of optimism, pessimism, and expansion through a single intrinsic-reward objective.
+ Empirically validated on diverse continuous-control tasks and real hardware, not only simulators.

**Weaknesses:**

- The proof seems to implicitly assume Lipschitz continuity of the uncertainty estimate $\sigma_n,$ which is not formally stated.  I think this assumption is required for several bounds.
- Assumption 4 requires the prior policy to be at least as safe as the optimal one for all states, which seems very strong.

**Questions:**

How sensitive is the safety guarantee to imperfect calibration of $\sigma_n$	​
Could Assumption 4 be relaxed to hold in expectation over $\rho_0$
How would the analysis change if the policy prior were stochastic or state-dependent only approximately safe?

---

> ### Author Response · Authors · 2025-11-18
>
> We would like to thank you for carefully reviewing our paper and for acknowledging the strengths of it. Please find below our answers to your questions.
>
> > Assumption 4 requires the prior policy to be at least as safe as the optimal one for all states.
> How would the analysis change if the policy prior were state-dependent only approximately safe?
>
> Thank you for this question. A precise answer to this question goes rather deeply into the theoretical analysis. We summarize the main implications below:
>
> - As long as the prior policy satisfies the constraint—irrespective of whether its cost-value is lower than that of an optimal policy—SOOPER is guaranteed to be safe. This is demonstrated in Lemma 1 in Appendix A.
> - When the cost-value is not smaller than that of an optimal policy,
>     global optimality may not be achieved; instead, optimality can only be guaranteed within a set of policies connected with $\pi_0$, similar to the optimality notion used in prior works (e.g. Actsafe [1]).
>     Please note that we elaborate on this part of the assumption in Remark 1 (L. 1167) in Appendix B. In particular, we detail how the result changes when this assumption is not satisfied.
>
> To summarize, the “strict” part of this assumption is required to guarantee _global optimality_—a fundamentally difficult problem. On the other hand, guaranteeing safety only requires the prior policy $\pi_0$ to satisfy the constraint; without this assumption, the problem is ill-posed because in the very first iteration $\pi_0$ may violate the constraint. While we introduce these two requirements together in Assumption 4 to keep the presentation concise, we are happy to revise the paper to make the above distinction clearer.
> Finally, in Appendix D, Figure 10, we empirically study the effect of this assumption and show the robustness of SOOPER to different levels of conservatism of the prior policy $\pi_0$.
>
> > How would the analysis change if the policy prior were stochastic?
>
> Thank you for raising this important question.
> It is indeed possible to run SOOPER with a stochastic prior policy. Specifically, Theorem 1 still holds in this setting. The analysis would change in Lemma 7 of the optimality proof. This is because a deterministic policy allows us to easily use Kakade's “Difference of Gaussians” simulation lemma [2] to achieve a relatively tight bound after Equation (59).
> In principle, under different assumptions on the policy, one could explore other (more technically involved) simulation lemmas developed for continuous domains [3, 4] to derive such bounds.
>
> > The proof seems to implicitly assume Lipschitz continuity of the uncertainty estimate which is not formally stated.
>
> Thank you for pointing this out. This assumption was implicit under our existing kernel regularity conditions (Assumption 3), but it was not stated explicitly in the previous revision. We have now added it in Assumption 2 based on your suggestion. As shown in Lemma 13 of [5], the Lipschitz constant follows directly from the continuity of the kernel over a bounded domain.
>
> We greatly appreciate your feedback, and we are happy to address any further questions or comments.
>
> ---
> [1] As, Yarden, Bhavya Sukhija, Lenart Treven, Carmelo Sferrazza, Stelian Coros, and Andreas Krause. "Actsafe: Active exploration with safety constraints for reinforcement learning." arXiv preprint arXiv:2410.09486 (2024).
>
> [2] Kakade, Sham, Akshay Krishnamurthy, Kendall Lowrey, Motoya Ohnishi, and Wen Sun. "Information theoretic regret bounds for online nonlinear control." Advances in Neural Information Processing Systems 33 (2020): 15312-15325.
>
> [3] Ferns, Norm, Prakash Panangaden, and Doina Precup. "Bisimulation metrics for continuous Markov decision processes." SIAM Journal on Computing 40, no. 6 (2011): 1662-1714.
>
> [4] Asadi, Kavosh, Dipendra Misra, and Michael Littman. "Lipschitz continuity in model-based reinforcement learning." In International conference on machine learning, pp. 264-273. PMLR, 2018.
>
> [5] Curi, Sebastian, Felix Berkenkamp, and Andreas Krause. "Efficient model-based reinforcement learning through optimistic policy search and planning." Advances in Neural Information Processing Systems 33 (2020): 14156-14170.

---

> > ### Comment · Reviewer_vK36 · 2025-11-26
> >
> > Thank you for your response in addressing my comments.

---

> > > ### Author Response · Authors · 2025-11-26
> > >
> > > Thank you again for your thorough and positive evaluation of our work!

---

### Meta-Review · Area_Chair_A4ej · 2026-01-08

**Summary:**

Reviewer vK36 was broadly positive (rating 8/10) but raised theory-facing concerns: the analysis appears to rely on an unstated Lipschitz continuity condition (needed for several bounds), and Assumption 4 was flagged as potentially very strong, requiring the prior policy to be at least as safe as the optimal policy for all states. Reviewer also asked about sensitivity to imperfect calibration / stochasticity of the prior policy.

Reviewer zYbx (rating 6/10) found the paper strong overall but expressed concern that the theoretical assumptions, notably around Gaussian/additive noise and related modeling conditions, may be stronger than prior work, and asked for clarification about the set of plausible models, the limitations of a model-based exploration strategy, and an explanation for why CRPO fails in a real-hardware experiment.

Reviewer XArr was also positive (rating 8/10) but echoed that the theory relies on strong assumptions that may limit applicability, and requested clearer discussion of computational overhead / design trade-offs for implementing SOOPER at scale as claimed.

**Reviewer Concerns:**

Most of the concerns are addressed by the thorough rebuttal.

Reviewer vK36: The rebuttal explicitly acknowledges and clarifies that the Lipschitz/continuity requirement was implicit and indicates it can be derived from kernel continuity over a bounded domain; it also distinguishes what part of Assumption 4 is needed for regret analysis and argues safety holds as long as the prior satisfies the constraint, with discussion of the stochastic-prior case and pointers to related techniques for continuous domains.

Reviewer zYbx: The rebuttal responds that the Gaussian/additivity assumptions are not as restrictive as they might appear (and notes empirical robustness under non-additive noise), adds clarification about model learning and algorithmic details (including the model update line), discusses a path to include model-free extension discussion in an appendix, and provides an explanation for the CRPO hardware result.

Reviewer XArr: The rebuttal provides a clearer breakdown of system components and trade-offs (including why terminations help bypass CMDP solvers, and discussion around implementation choices), and agrees that additional experiments under violated assumptions would be valuable—while also stating they observe good performance even when assumptions are relaxed/violated.

**Reviewer Scores:**

Reviewers vK36 and XArr are very likely to keep their high score (8/10). Reviewer zYbx  is very likely to increase their score from 6 to 8, based on the response of this reviewer to the rebuttal.

---

### Decision · Program_Chairs · 2026-01-26

Accept (Poster)